# AGENTIC DESIGN OF COMPOSITIONAL MACHINES

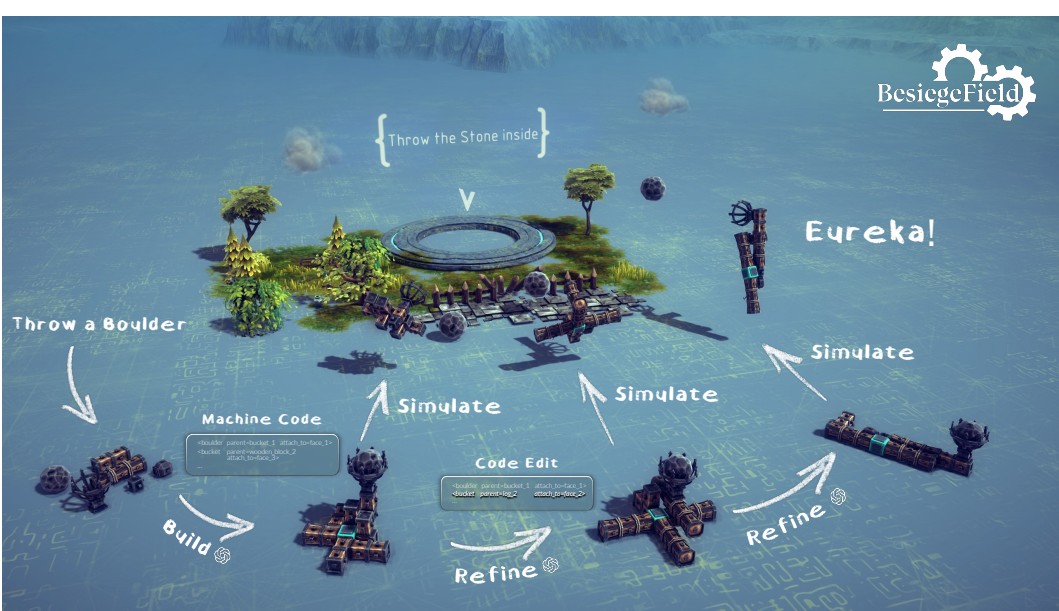

Figure 1: The task of compositional machine design is illustrated in our **BesiegeField** environment. The figure shows a high-level sketch of the agentic workflow (w/ Gemini Pro 2.5), along with the resulting machines and their simulated performance. The design objective is to create a machine that throws boulders long distances.

## ABSTRACT

The design of complex machines stands as both a marker of human intelligence and a foundation of engineering practice. Given recent advances in large language models (LLMs), we ask whether they, too, can learn to create. We approach this question through the lens of compositional machine design: a task in which machines are assembled from standardized components to meet functional demands like locomotion or manipulation in a simulated physical environment. With this simplification, machine design is expressed as writing XML-like code that explicitly specifies pairwise part connections. To support this investigation, we introduce **BesiegeField**, a testbed built on the machine-building game Besiege with part-based construction and physical simulation. Using **BesiegeField**, we benchmark state-of-the-art LLMs with agentic workflows and identify key capabilities required for success, including spatial reasoning, strategic assembly, and instruction-following. As current open-source models fall short, we explore reinforcement learning (RL) as a path to improvement: we curate a cold-start dataset, conduct RL finetuning experiments, and highlight open challenges at the intersection of language, machine design, and physical reasoning.

*"Man is a tool-making animal."*

— Benjamin Franklin

## 1 INTRODUCTION

The history of human progress is, at its core, the history of machines, just as the ancient Greeks built the Antikythera mechanism to predict eclipses and Leonardo da Vinci envisioned machines to fly. Today, as large language models (LLMs) begin to approximate—and in some domains, surpass—human cognitive abilities, a natural question arises:

> *Can computational models, like humans, conceive and create complex machines to achieve purposeful goals?*

At the heart of this question lie two tightly coupled concepts: *compositionality*, how parts are put together into assemblies, and *functionality*, the tasks these assemblies perform as they interact with external forces or inputs. While foundation models are already capable of synthesizing 3D shapes and building mechanical parts with computer-aided design (CAD) models, it is the complex compositional structures, in which very different parts and components are orchestrated to smoothly move together, that realize a vast array of demands. Just as a clock emerges from the composition of simple and standardized mechanical elements such as gears and flywheels, these same elements, when combined differently, can give rise to entirely different machines, such as a sewing machine. On the other hand, the same functionality may be realized by different part compositions, just as both cars and bicycles can transport a person from place to place. Put it concisely: *composition is shaped by functionality, and functionality is realized through composition*. Since such compositional machines can be expressed programmatically, with types, placements and articulations of parts represented in structured code that LLMs can generate and manipulate, we formalize the above question as:

> *Can LLMs, given standardized mechanical parts and a reward function for the desired functionality, discover diverse spatial part compositions that maximize the reward and complete the task?*

The question is not only about the pursuit of intelligence but also about the practice of engineering. Modern design pipelines are often long and costly, especially in large-scale projects where each iteration demands substantial resources. These projects accumulate vast collections of documents and blueprints, making it difficult to trace, retrieve, or reuse past design efforts. Much essential know-how is passed informally across teams and generations, and in many cases, never fully recorded and since forgotten. An automated machine design system could directly address these challenges.

Rather than merely mimicking patterns from historical designs, such a system should be agentic: capable of exploring the exponentially large design space, leveraging prior knowledge to create novel designs for new demands and constraints, and improving them through feedback. To investigate this concretely, we introduce **BesiegeField**, an interactive environment built on the machine-design game of Besiege[1]. The environment allows for construction of simple mechanical machines with standardized and semantic parts such as gears and wheels, and supports customized physical scenarios in which LLM agents can test constructed machines and evaluate their dynamics and interactions.

Building on **BesiegeField**, we benchmark state-of-the-art LLMs with different agent designs and strategies for selecting and placing basic mechanical elements to build machines for representative functional demands, a task we term *compositional machine design*. Through these experiments, we empirically identify key capabilities required for this task: accurate spatial reasoning, high-level knowledge of design strategies, and instruction-following in spatial domains. Since only a few proprietary LLMs achieve satisfactory results, we further investigate how reinforcement learning (RL) can improve the performance of open-source LLMs. To this end, we curate a small machine design dataset to cold-start RL finetuning, perform exploratory RL experiments, and highlight key challenges that chart directions for future research. In summary, our contributions are listed below:

- We introduce and formalize the task of *compositional machine design*, where machines are assembled from standardized parts to achieve functional goals.

- We present **BesiegeField**, an interactive environment that enables LLM agents to construct, simulate, and evaluate compositional machines in customized physical scenarios.

- We systematically benchmark state-of-the-art LLMs and different agentic workflow designs on representative machine-design tasks.

- We explore RL finetuning of LLMs on this task, for which we curate a cold-start dataset, conduct experiments, and highlight the key challenges.

## 2 COMPOSITIONAL MACHINE DESIGN

Full machine design involves many coupled elements: geometry, statics and dynamics, demand analysis, failure modes, safety, and even legal constraints (Beitz et al., 1996; Wong et al., 2025). To

---

[1]https://en.wikipedia.org/wiki/Besiege_(video_game)

isolate a tractable subproblem, we focus on the structural composition of machines: how standardized parts are spatially arranged and mechanically linked to produce functional behavior. We refer to this task, introduced in the previous section, as *compositional machine design*. It captures two essential components: (i) the static geometry of a machine as a part-based assembly, and (ii) its compatibility with functional demands, typically assessed through physical simulation. This abstraction omits considerations such as manufacturing constraints, material properties, or domain-specific regulations, but retains the core spatial and behavioral reasoning challenges relevant to design.

This special task of compositional machine design mirrors challenges found in other exploration domains. For example, automatic theorem proving involves a compositional and exponentially large action space, while electronic design automation (EDA) for chip layouts requires spatial reasoning to place components of varying shapes under spatial constraints (albeit in a more regular and grid-constrained fashion than mechanical parts in machines). A unique challenge in machine design, however, is its dependence on diverse long-horizon behaviors, both autonomous and non-autonomous, within an environment. Specifically, a machine may behave differently when operated in different ways (*e.g.*, a bicycle when pedaled versus when braking) or under different external conditions (*e.g.*, driving a car in sunny versus rainy weather). Similarly, many sophisticated machines cannot function without appropriate control policies, as exemplified by aircraft that rely on fly-by-wire systems to stabilize their inherently unstable aerodynamic configurations (which would otherwise be unflyable by a human pilot alone). A key open problem is therefore how to account for the interplay among physics, control policy, and compositional structure in machine design.

It is worth noting that, unlike in math theorem proving where one valid proof often suffices (even though multiple proofs may still be valued), design domains typically require generating a diverse set of candidate solutions. This diversity is essential to (i) differentiate products, (ii) adapt to unpredictable market demands, and (iii) account for uncertainty in real-world testing and deployment. Consequently, the task places greater emphasis on diversity, and a model for compositional machine design should function more like a generative model than a simple reward maximizer.

## 3 BESIEGEFIELD: PLAYGROUND FOR COMPOSITIONAL MACHINE DESIGN

Studying the full problem of compositional machine design is challenging, as it involves the coupling of many interacting factors. We therefore focus on a minimalist, component-level setting in which machines are constructed primarily from cuboid primitives with clear functional semantics, together with a small set of specialized exceptions, and operate under a shared control policy in an environment governed by rigid-body and elastic mechanics. This abstraction allows us to properly benchmark the capabilities of existing LLMs and to assess the upper bounds, potential, and challenges of agentic systems and RL algorithms.

To this end, we create **BesiegeField**, an interactive environment adapted from the machine-building game Besiege, in which players design medieval machines to complete tasks such as destroying castles. Powered by the built-in physics engine, **BesiegeField** supports physical simulation of mechanical systems such as vehicles and catapults in user-customized environments with terrains, obstacles, external forces (*e.g.*, wind and gravity), and co-existing agents. The environment provides nearly 80 types of building blocks , including passive ones like drills and logs, and powered ones like powered cogs and wheels. Machines are constructed by sequentially attaching new parts to vacant and attachable faces of existing blocks, starting from a root block and thus forming a "construction tree" (indeed a directly acyclic graph (DAG), in the sense of operation orders; one block can has two parents in the DAG; the actual structures may contain loops). Powered blocks can receive control commands, allowing machines to be operated precisely. During simulation, complete state information (e.g., the position and velocity of each block in the constructed machine) can be recorded for model feedback. Finally, the environment supports custom modifications and can be extended with additional block types and richer physics (*e.g.*, simple fluid simulation). Further details are explained in Appendix B.

**BesiegeField** is unique in balancing real-world geometry and physics, part-level semantics, and simple compositional rules. Block-stacking environments like LEGO (Fan et al., 2022) and Minecraft (Fan et al., 2022; Pun et al., 2025) allow intuitive combinatorial assembly but do not natively provide realistic physical simulation and rely on generic blocks with limited semantic meaning. CAD modeling (Li et al., 2025) captures fine-grained geometry and interactions, but its complexity makes rules cumbersome and sequences prohibitively long. By contrast, **BesiegeField** uses

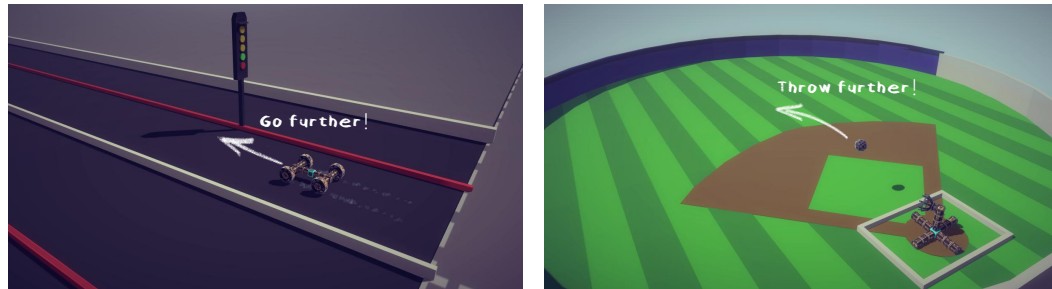

Figure 2: Demonstration of the machine design tasks in our experiments. (Left: *car*; Right: *catapult*).

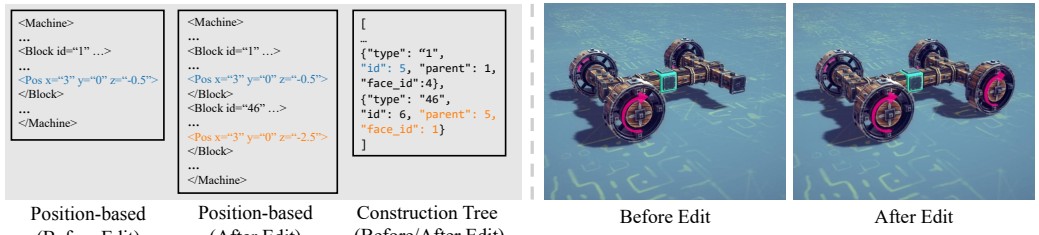

| Position-based (Before Edit) | Position-based (After Edit) | Construction Tree (Before/After Edit) | Before Edit | After Edit |

Figure 3: Demonstration of the default position-based representation and our construction tree representation. Parent block info is in blue and child info is in red.

semantically meaningful parts with cuboid-like construction rules-supporting realistic physics while remaining abstract enough for tractable composition. This calibrated balance enables the study of compositional creativity and geometric reasoning at a level of difficulty that both differentiates algorithms and permits rapid experimentation. Moreover, unlike prior environments, **BesiegeField** supports machine destruction, adding durability and failure analysis to the design space.

## 4 BENCHMARKING LLMs FOR COMPOSITIONAL MACHINE DESIGN

### 4.1 BENCHMARK SETTINGS

**Representative target machines and tasks.** To benchmark and characterize the performance of different LLMs for agentic compositional machine design, we consider two conceptually simple yet representative target machines to build: *car* and *catapult* as shown in Fig. 2. While success in both requires understanding part semantics and structural syntax, *car* building primarily tests static relational reasoning, such as enforcing correct part orientations, symmetry, and stability; in contrast, *catapult* building challenges models with dynamic relational reasoning, where parts must coordinate over time to produce causal mechanical effects. Moreover, the two tasks are simple enough to be constructed with only a few blocks so that they fit within the LLM's context window, yet complex enough to require explicit reasoning about construction strategies and causal dependencies. We evaluate the performance of *cars* and *catapults* by their moving distance and their throwing distance (*i.e.*, the moving distance of the stone), respectively, towards a fixed and given direction. During each simulation, the generated machine will be placed at a designated position, and the active parts will be powered after a few seconds. As there can be reward hacking issues, for *catapults* experiments we surround the designated machine placement position with moderate-height walls. More details about the target machines, rewards, and environments can be found in Appendix B.

**Machine representations.** In **BesiegeField**, the default position-based representation records all blocks with global 3D positions and uses a built-in algorithm to recover connections. Such a representation, however, does not well encode machine structures. Instead, we propose a parsimonious representation aligned with the game's building logic, based on pairwise relative attachment relationships (*i.e.*, how one part is rotated and attached to another). Details are in Appendix D.1.2.

**Performance metrics.** We evaluate our agentic systems using the following quantitative metrics: 1) *file validity rate*, the proportion of generated JSON files that can be successfully parsed into machine construction trees; 2) *spatial validity rate*, the proportion of generated machines that are free from self-collisions; 3) *machine validity rate*, the proportion of machines that satisfy both file and spatial validity; 4) *mean and maximum simulation scores*, the average and highest rewards achieved by generated machines in the environment.

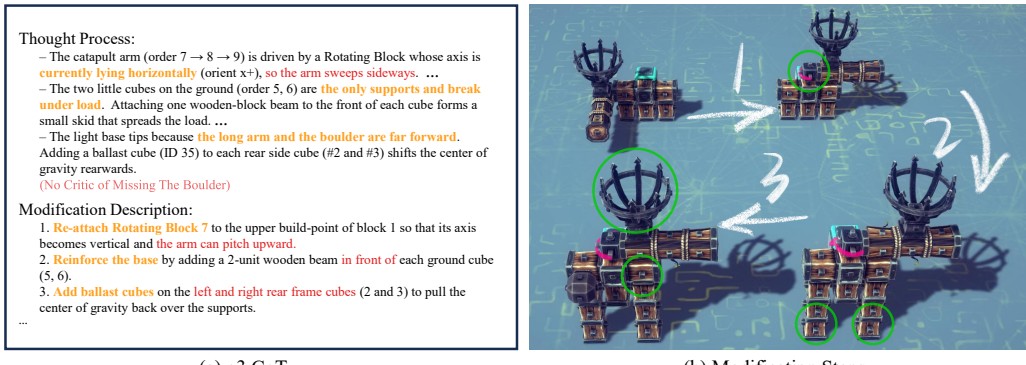

I need to analyze the provided machine data to identify and correct the flaws.
...

**Analysis of Defects:**
– **Severe Structural Imbalance.** The machine has a tiny support base (1, 2, 3, 4, 5). The long arm with the heavy Boulder 12 will create a large torque, causing the machine to tip over backward (towards z-). ...
– **Incorrect Pivot Orientation.** The block 9 spins the arm horizontally. ...Let's check the current setup: ...Rotation around the x-axis causes a swing in the y-z plane... However, the Log 10 is attached to the 'Up' face of the pivot. This means the arm starts pointing straight up, which is ineffective for a throw. ...
– **Improper Container Setup.** After moving the arm, I must also move the Container 11 to an 'Up' attachable face on the Log 10 to make it face upward y+ again. The Log 10 has several 'Up' attachable faces; face_id 8 is a suitable choice.
...

**Consolidated Modification Plan:**
1. **Remove Old Base:** Remove blocks [3], [4], and [6] first, then remove their parents [1], [2], and [5].
2. **Reposition Arm And Container:** Move Log [10] from Rotating Block [9]'s face_id: 3 (Up) to face_id: 2 (Right). Move Container [11] from Log [10]'s face_id: 0 (Front) to face_id: 8 (Up). And Add a Ballast [35] to Rotating Block [9] at face_id: 1 (Left).
3. **Build New Base:** Add three Log [63] blocks to the Starting Block [0] at face_id: 1 (Back), face_id: 2 (Left), and face_id: 3 (Right).
...

(a) Gemini 2.5 Pro CoT         (b) Modification Steps

Figure 4: Example CoT of inspector agents (w/ Gemini 2.5 Pro). Blue text highlights the moderate capability of LLMs in spatial reasoning and imagined physical simulation.

**Thought Process:**
– The catapult arm (order $7 \rightarrow 8 \rightarrow 9$) is driven by a Rotating Block whose axis is currently lying horizontally (orient x+), so the arm sweeps sideways. ...
– The two little cubes on the ground (order 5, 6) are the only supports and break under load. Attaching one wooden-block beam to the front of each cube forms a small skid that spreads the load. ...
– The light base tips because the long arm and the boulder are far forward. Adding a ballast cube (ID 35) to each rear side cube (#2 and #3) shifts the center of gravity rearwards.
(No Critic of Missing The Boulder)

**Modification Description:**
1. **Re-attach Rotating Block 7** to the upper build-point of block 1 so that its axis becomes vertical and the arm can pitch upward.
2. **Reinforce the base** by adding a 2-unit wooden beam in front of each ground cube (5, 6).
3. **Add ballast cubes** on the left and right rear frame cubes (2 and 3) to pull the center of gravity back over the supports.
...

(a) o3 CoT         (b) Modification Steps

Figure 5: Example CoT of inspector agents (w/ OpenAI o3). Red text highlights reasoning errors.

**Environment feedback.** For the simple target machines *car* and *catapult*, we consider environment feedback within a time window of 5 seconds that is long enough to characterize their designated functionalities. Specifically, for *car* we consider maximum speed and driving distance; for *catapult*, we consider boulder throwing distance and maximum height. We also record the machines' global orientation and broken parts information (if any). Details are elaborated in Appendix D.3.

## 4.2 AGENTIC WORKFLOW DESIGN

**Single-agent setting.** We first benchmark if a single LLM agent alone is capable of completing the task. Specifically, one LLM agent is provided with the environment description, the available machine components, the assembly syntax, and the functional requirements (*e.g.*, moving an object forward). The agent generates a chain-of-thought (CoT; Wei et al. (2022)) to reason about what is needed and why, and then derives an abstract plan (e.g., connecting a lever to a container with a boulder). This plan is later translated into the construction tree representation.

**Iterative editing.** Compositional machine design demands both low-level spatial reasoning and high-level conceptual planning, making it unlikely for a single agent to produce a satisfactory machine on the first attempt. We therefore develop an iterative editing workflow involving four main agents:

- *Designer*: Generates an initial machine plan given the environment description, available components, assembly syntax, and functional requirements.
- *Inspector*: Evaluates the machine from a high-level, structural perspective and identifies conceptual issues.
- *Refiner*: Acts as a critic that assesses the design against requirements and constraints, and proposes candidate revisions.

Figure 7: Machines produced by agentic systems with different LLMs (Top: *car*; Bottom: *catapult*).

- *Environment querier*: Executes machine simulations and summarizes environment feedback. It consistently reports global information (e.g., orientation over the trajectory) and selectively provides detailed block-level feedback (e.g., position, speed) to support further refinement.

In summary, the designer produces an initial draft, which is then improved through a sequence of "refiner–querier" loops. Because the initial draft is typically flawed, the *inspector* is invoked *once* to produce high-level critiques, and *only the first refiner* receives this feedback along with the initial draft. Subsequent refinement stages do not use the inspector; instead, each refiner generates several candidate revisions, which are evaluated via Monte Carlo Tree Search (MCTS; Coulom (2006)). The best design identified in this search is selected as the output.

**Hierarchical construction.** Inspired by typical human design processes as well as recent designs of agentic systems (Xiao et al., 2025; Teng et al., 2025; Zhang et al., 2025), we introduce a meta-designer agent that first analyzes the requirements and constraints, and then constructs a high-level blueprint of the major functional blocks (*e.g.*, the suspension system) and their interconnections. With this decomposition in hand, we adopt a beam-search–style strategy to construct the machine block by block:

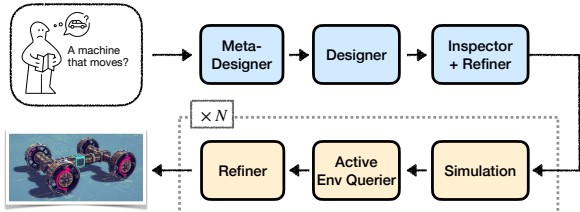

Figure 6: Our agentic machine design workflow.

1. We start with the first functional block and dispatch the task to 8 parallel designer agents. Some proposed designs may fail, so fewer than 8 outputs may be valid. We then duplicate the valid outputs to obtain 8 candidates (the initial states of 8 trajectories).

2. Each of these 8 candidates is sent to an independent designer agent, which constructs the second functional block. As before, spatial conflicts may arise and only a subset of designs will be valid. We again duplicate the valid ones to maintain 8 candidates.

3. This iterative process continues block by block until the entire machine is assembled.

Empirically, the meta-designer tends to decompose a machine into three to four functional blocks.

### 4.3 KEY EMPIRICAL OBSERVATIONS

**General observations.** We find compositional machine design to be a challenging task for LLMs (Fig. 7 and Table 1), though not intractable: Gemini 2.5 Pro can consistently construct visually sensible machines with non-trivial performance. We find no evidence that reasoning models outperform non-reasoning ones, suggesting the main bottleneck lies in LLMs' limited 3D understanding and/or in-context learning. We also find that LLMs, especially reasoning models, still exhibit some spatial and physical reasoning as exemplified by the CoT from Gemini Pro 2.5 (Fig. 4), much like a world model in text space.

**Failure patterns.** We identified common failure patterns in LLM-generated machines (Fig. 20): 1) *incorrect part orientations*; 2) *incorrect part placements*, where parts attach to wrong parents; 3) *instruction-following failures*, where elements of the high-level blueprint are not strictly observed; 4) *flawed high-level reasoning*, where LLMs fail to recognize correct physics or essential components.

**Effect of environment feedback.** It is unsurprising that with the more environment feedback the agents receive, the better performance of generated machines improve in general (Table 15).

| Models | Single-agent | | | Iterative Editing | | | Hierarchical Design | | |
|---|---|---|---|---|---|---|---|---|---|
| | Mean | Max | Std | Mean | Max | Std | Mean | Max | Std |
| *"Catapult" Task* | | | | | | | | | |
| Gemini 2.5 Pro | 2.30 | 9.0 | 3.86 | 4.67 | **21.95** | 8.68 | **9.83** | **18.19** | 8.35 |
| OpenAI o3 | 2.87 | 5.22 | 1.96 | **9.14** | 14.01 | 3.71 | 2.00 | 11.11 | 3.98 |
| Qwen3-Coder-480B-A35B | 1.75 | 9.24 | 3.17 | 5.10 | 12.02 | 5.54 | 3.90 | 6.52 | 2.54 |
| Doubao Seed 1.6 | 3.18 | 8.2 | 2.99 | 4.82 | 9.10 | 3.41 | 1.73 | 4.76 | 2.39 |
| Claude Opus 4 | 1.19 | 4.82 | 2.21 | 1.18 | 4.91 | 2.18 | 2.27 | 9.32 | 4.22 |
| DeepSeek-V3 | **3.50** | 4.86 | 2.17 | 3.07 | 5.24 | 2.55 | 2.41 | 4.93 | 2.58 |
| Kimi K2 | 2.57 | **9.05** | 3.72 | 2.82 | 11.39 | 5.23 | 5.39 | 12.02 | 5.16 |
| Llama 4 Scout 17B 16E | 3.18 | 5.64 | 1.95 | 1.28 | 5.94 | 2.41 | 3.59 | 11.83 | 4.15 |
| Human | 153.90 | 977.1 | 291.83 | - | - | - | - | - | - |
| *"Car" Task* | | | | | | | | | |
| Gemini 2.5 Pro | **33.96** | **40.85** | 6.73 | **34.34** | **41.66** | 13.96 | **29.96** | **41.52** | 7.78 |
| OpenAI o3 | 15.28 | 32.08 | 8.97 | 14.34 | 35.08 | 11.79 | 28.39 | 36.18 | 11.01 |
| Qwen3-Coder-480B-A35B | 8.87 | 11.50 | 4.46 | 15.24 | 28.95 | 13.12 | 12.59 | 34.05 | 10.78 |
| Doubao Seed 1.6 | 3.51 | 9.40 | 4.85 | 8.11 | 10.04 | 3.58 | 18.75 | 26.02 | 4.38 |
| Claude Opus 4 | 9.83 | 12.98 | 1.28 | 8.07 | 28.04 | 12.48 | 14.56 | 38.67 | 20.69 |
| DeepSeek-V3 | 9.06 | 10.53 | 3.68 | 8.23 | 18.84 | 7.12 | 17.92 | 31.94 | 12.85 |
| Kimi K2 | 1.75 | 8.09 | 2.80 | 14.36 | 28.34 | 9.47 | 1.94 | 14.99 | 5.48 |
| Llama 4 Scout 17B 16E | 0.02 | 0.03 | 0.01 | 3.04 | 12.76 | 5.23 | 1.55 | 2.00 | 0.32 |
| Human | 177.23 | 386.58 | 103.29 | - | - | - | - | - | - |

Table 1: Quantitative results of agentic systems with different LLMs.

**Effect of edit history.** We find that edit histories are generally helpful in decreasing the number of failure attempts in creating valid machines (Table 6), which underscores the importance of longer context window of base models for efficient exploration.

**Hierarchical design.** We observe the mean performance improves with hierarchical design only when the abstract-level reasoning on blueprints is reliable, as shown by the performance of Gemini 2.5 Pro. In the meantime, consistent with the intuition that hierarchical design is more structured and principled, it generally yields lower variance in obtained scores.

**Effect of CoT reasoning.** As shown in Fig. 20, LLMs often fail to faithfully translate high-level machine design plans in their CoT into semantically and geometrically consistent machine construction trees. To better assess the impact of CoT reasoning on high-level design, we feed the CoT generated by Gemini 2.5 Pro (the best-performing model) to other LLMs, prompting them to directly output construction trees. The resulting machines generally show improved performance (Fig. 35) and highlight the critical role of high-level semantic reasoning in machine design.

**CoT-machine correspondence.** Though the CoT often provides a reasonably high-level blueprint, agents may still generate machines that deviate from the intended structure (Fig. 20). We hypothesize that this misalignment is a key reason many LLMs struggle to build better machines.

**Machine representation.** We experiment with a coordinate-only representation derived from the default position-based (Appendix D.1) and our construction tree representation. Results show that the coordinate-only representation performs significantly worse (Table 7), implying that explicit structural information is necessary for LLM understanding.

**3D information.** We observe that (Table 5) the performance generally improves when we also feed parsed 3D information into the context of LLMs, which implies that LLMs are less capable of understanding relative spatial relationship (*e.g.*, construction trees).

## 5 TOWARDS MACHINE DESIGN THROUGH REINFORCEMENT LEARNING

Although agentic systems show promise in compositional machine design, simply scaling system size is unlikely to be economical, as errors compound rapidly. Like humans who internalize experience, LLM agents should consolidate new knowledge into weights. We thus explore reinforcement learning with verifiable rewards (RLVR) in **BesiegeField** to develop machine-design capabilities.

### 5.1 EXPERIMENTAL SETTINGS

**Cold-start finetuning and dataset curation.** Following recent RLVR practices (Lambert et al., 2025; Yue et al., 2025; Zhu et al., 2025a), we curated a small dataset to cold-start LLMs by aligning their reasoning process with expert CoT. Specifically, we collected textual descriptions of machine functionalities from Besiege player communities and prompted Gemini 2.5 Pro to generate corresponding machines. After filtering out invalid generations, we obtained 9,984 valid machine-CoT

| Models | Catapult | | | Car | | |
|---|---|---|---|---|---|---|
| | Validity Ratio | Mean Score | Max Score | Validity Ratio | Mean Score | Max Score |
| Qwen2.5-14B-Instruct | 11/50 | 0.06 | 2.41 | 46/50 | 4.97 | 19.10 |
| Qwen2.5-14B-Instruct + Cold-Start | 9/50 | 0.11 | 5.54 | 40/50 | 4.67 | 20.23 |
| Qwen2.5-14B-Instruct + RL | **12**/50 | 0.13 | 5.92 | 41/50 | 3.72 | 24.08 |
| Qwen2.5-14B-Instruct + Cold-Start + RL | 11/50 | **0.14** | **7.14** | **42**/50 | **5.05** | **45.72** |

Table 2: Results of RLVR post-training in **BesiegeField**. We use Qwen2.5-14B as the backbond LLM.

pairs. We then used this dataset to perform supervised finetuning on Qwen-2.5-14B-Instruct for 12 epochs. Additional training details are provided in Appendix F.2.

**Reward design.** We use the reward $R = \texttt{is\_valid} \times \texttt{performance}$ where `is_valid` indicates whether constraints are satisfied (Appendix D.2). For *car*, `performance` is the maximum travel distance; for *catapult*, it is the product of the boulder's maximum height and distance, penalizing solutions that are extreme in only one dimension.

**RL finetuning settings.** We finetune agents specialized in building a single type of machine (either *car* or *catapult*), making our setup closely aligned with one-shot RLVR (Wang et al., 2025) where a single prompt is used throughout the RL process. We adopt group relative policy optimization (GRPO; Shao et al. (2024)) with LoRA parametrization (Hu et al., 2022) (rank 64) and mixed-precision training to finetune the cold-started model. We evaluate both the standard GRPO advantage estimator and the pass@k variant (Tang et al., 2025b). In the latter case, due to the implementation of the RLVR framework verl (Sheng et al., 2025), the number of rollouts is set equal to $k$. Each experiment is run for 400 iterations on 8 A100 GPUs with per-GPU batch size of 1 and gradient accumulation of 8. We apply KL regularization with strength 0.001 to encourage the model to remain close to its initialization.

## 5.2 Main Results and Observations

**General results.** As shown in Fig. 29, RL finetuning can generally improve the mean performance, mostly by increasing the percentage that machines are valid (including file validity, machine validity and satisfaction of minimum performance threshold). In the meantime, we also find that the maximum reward increases in our best setting. Similar to observations in many other RLVR settings, the entropy of the output distribution quickly drops even with regularization.

**Pass@k advantage vs. Pass@1 advantage.** Since we eventually care about the best performing designs, especially given the low validity rate, our default setting adopts Pass@k advantage estimator. Indeed, Pass@k finetuning is more likely to discovery promising machine designs (Fig. 27).

**Evolution of generated machines during finetuning.** In Fig. 8, we qualitatively examine how models refine their designs over the course of finetuning. We observe that models typically make detail-level adjustments, such as shifting part positions, while keeping the same high-level design strategy rather than exploring alternative strategies. Although these strategies are often reasonable, the models struggle to find precise configurations that enable smooth coordination among parts. This precision is especially critical for sophisticated mechanisms like catapults to function properly.

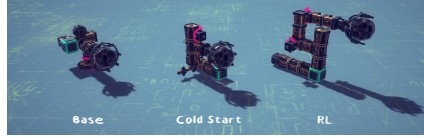

Figure 8: Designs at RL finetuning stages.

**Cold-start.** Not surprisingly, we find that cold-start alone does not enable models to produce satisfactory designs, and that finetuning on the cold-start model is better than on the base model (Table 2).

## 6 Discussions and Intriguing Insights

**Capabilities for compositional machine design.** Although tasks such as visual understanding and generation also depend on spatial, physical, and semantic reasoning, compositional machine design introduces unique requirements for LLM capabilities. Without precise spatial placement of machine parts, a design may fail to function correctly; a gear train, for example, will not transmit rotation if the gears are misaligned. Since the design process is typically hierarchical, successful LLMs must be able to accurately translate high-level blueprints into detailed geometric designs. In addition, machine design spans both concept-level reasoning and detailed specification. This dual demand often leads to large design documents and calls for a form of "visual reasoning" expressed through

text, similar to what has been studied in LLMs applied to scalable vector graphics (SVG) and CAD models (Qiu et al., 2025b; Alrashedy et al., 2025). Multimodal reasoning is also important because effective machine design typically relies on integrating textual descriptions with visual or schematic representations. In this work, however, we focus only on pure LLM-based reasoning to isolate and analyze its capabilities for compositional machine design.

**Challenges in agentic machine design systems.** The task of machine design faces similar challenges found in agentic systems in domains such as legal services and other knowledge-intensive fields. A key difficulty is the highly varied requirements and domain knowledge of different customers. To address this, LLMs need to acquire task-specific knowledge through in-context learning or finetuning. In addition, the complexity of design tasks often requires multiple agents to coordinate, and such pipelines can suffer error accumulation when the base LLM lacks sufficient capability.

**Exploration in machine design space.** Different from tasks such as theorem proving, the goal of compositional machine design is to discover structures that more effectively achieve desired functionalities. Rather than reusing existing solutions, a practical design agent should be able to propose novel strategies, structural layouts, and part specifications as machine complexity increases. Meeting this requirement calls for RL finetuning methods that prevent models from collapsing into a narrow set of strategies and structures, which recent methods aim to alleviate (Zhu et al., 2025b; Chen et al., 2025c; Cui et al., 2025; Cheng et al., 2025; Liu et al., 2025b). This demand is closely related to continual RL (Schwarz et al., 2018), since finetuned LLMs must avoid catastrophic forgetting, maintain its reasoning ability, and consolidate learned strategies, which is particularly important because large-scale machine design datasets are rare and commercially infeasible to collect.

## 7 RELATED WORK AND CONCLUDING REMARKS

**3D graphics codes for generative modeling.** There is a long history in 3D asset generation and engineering design of representing the construction of a target instance as a program or sequence of operations in a domain-specific language (Ritchie et al., 2023; Sun et al., 2025; Deng et al., 2022), which we refer to here as 3D graphics codes (Qiu et al., 2025b; Chen et al., 2025a). Unlike geometric representations such as point clouds or meshes, these codes describe objects at a higher semantic level, capturing part composition, design constraints, and user operations in modeling software. Similar to programming languages, 3D graphics codes are inherently discrete and are typically generated with autoregressive models trained from scratch (Yuan et al., 2024) or with LLMs finetuned on curated datasets (Kulits et al., 2025; Chen et al., 2025b). Much of the existing work centers on CAD scripts for individual parts (Wu et al., 2023; Alrashedy et al., 2025; Li et al., 2025) or Blender macros for single assets (Huang et al., 2024). Whereas recent studies on LEGO assemblies (Pun et al., 2025), Minecraft structures (Fan et al., 2022; Liu et al., 2025a), and procedural scene generation (Sun et al., 2025; Chen et al., 2025a; Jones et al., 2025; Yuan et al., 2024) introduce richer compositionality, they still fall short of the task of compositional machine design, which requires assemblies that both function under physical laws and exhibit the precise geometry of real objects.

**LLM agents.** LLM agents are language models organized to operate in iterative loops of perception and action (Yao et al., 2023b; Minaee et al., 2024; Hu et al., 2024c). They interact with external tools (Schick et al., 2023; Liu et al., 2024c; Kim et al., 2024; Qin et al., 2024), respond to signals from simulated or real environments (Savva et al., 2019; Shridhar et al., 2021), incorporate self-reflection to refine their outputs (Hu et al., 2024b; Alrashedy et al., 2025; Shinn et al., 2023; Yu et al., 2025), and are commonly organized into multi-agent systems that coordinate roles and exchange information (Li et al., 2023; Chen et al., 2024; Zhang et al., 2025) . These designs move beyond one-shot text generation and establish LLMs as adaptive decision makers capable of long-horizon reasoning. Approaches that introduce search over possible solutions (Yao et al., 2023a; Putta et al., 2024; Koh et al., 2024) or reflection on prior attempts (Besta et al., 2024; Deng et al., 2024; Renze & Guven, 2024; Xiao et al., 2025; Yu et al., 2025) have enabled progress on increasingly complex tasks. LLM agents have already been used in design tasks such as code synthesis (Gao et al., 2023; Novikov et al., 2025; Madaan et al., 2023), CAD design (Alrashedy et al., 2025) and game environments (Wang et al., 2024; Fan et al., 2022). Partially inspired by these developments, Makatura et al. (2023) proposed a prototypical agent-based design framework that generates mechanical structures from text prompts. Their system treats structure generation as a one-shot process and delegates the search for optimal geometric and physical parameters to external optimization tools. In contrast, our work with **BesiegeField** explores how LLM agents can directly and iteratively bridge compositional

structures to functional goals, framing design as a process of reasoning and adaptation with both accurate simulation and intuitive physics.

**Reinforcement learning with verifiable rewards (RLVR).** Recent studies indicate that, by running RL finetuning with verifiable rewards from simulators or verifiers, reasoning abilities emerge (Shao et al., 2024; Guo et al., 2025; Bai et al., 2022), even when single prompt is used during finetuning (Wang et al., 2025). Yet, many methods exhibit loss of diversity as output entropy collapses during reinforcement learning and thus do not fully enable LLMs to explore novel solutions. Examples of mitigation methods include explicit entropy or KL regularization (Cui et al., 2025; Ouyang et al., 2022), Pass@k training (Tang et al., 2025b; Chen et al., 2025c), and distribution-matching objectives like generative flow networks (Zhu et al., 2025b; Hu et al., 2024a). **BesiegeField** provides verifiable rewards and thus enables direct application of RLVR to compositional machine design.

**Robot Design.** The task of robot design, which determines the topological structure and geometric shape of a robot so that it can better accomplish a target task, shares many similarities with classical machine design. Robot design is typically formulated as a co-design problem, because the optimal morphology is tightly coupled with the optimal control policy. For rigid-body robots, many works focus on optimizing structural parameters such as link or leg lengths (Ha et al., 2016; Carlone & Pinciroli, 2019; Belmonte-Baeza et al., 2022; Ghansah et al., 2023). Although some studies explore changes in the morphology of rigid robots (Yuan et al., 2022), a large body of recent research focuses on soft robots (Bhatia et al., 2021; Dong et al., 2023), which pose greater challenges for both design and control. Recent work also investigates the use of LLMs to assist the morphology optimization process (Ringel et al., 2025; Qiu et al., 2024). While both compositional machine design and robot morphology design involve topological structure optimization, the focus of compositional machine design is the broader task of machine design in mechanical and structural engineering. Indeed, it concerns the assembly of heterogeneous mechanical components with intrinsic physical semantics, such as wheels, gears, springs, and propellers. This heterogeneous modularity produces a more hierarchical design space, where complex machines can be viewed as compositions of functional subsystems such as drivetrains or suspension systems. Moreover, although many machines require a controller to operate, there exist important classes of machines that do not, including engines and gear trains, which distinguishes compositional machine design from robot design in which morphology and control are almost always jointly considered.

**Benchmark Suites for LLM Agents.** Recent evaluations of LLM agents have progressed from isolated tool-use tasks to interactive multi-modal environments. Early benchmarks like ToolBench (Qin et al., 2024) and Gorilla (Patil et al., 2024) focused on API orchestration, while SWE-bench (Jimenez et al., 2024) raised the bar for code generation with real-world software engineering tasks. Embodied AI suites such as ProcTHOR (Deitke et al., 2022) and web navigation benchmarks like WebArena (Zhou et al., 2024) assess grounding and long-horizon reasoning in physical and digital domains. Comprehensive frameworks like AgentBench (Liu et al., 2024b) unify these tasks to test general-purpose agents. However, benchmarks requiring generative mechanical reasoning under physics remain scarce: while MineDojo (Fan et al., 2022) and LEGO-Puzzles (Tang et al., 2025a) evaluate spatial-related tasks, they lack rigorous simulation of functional performance. This gap necessitates new testbeds for compositional machine design where agents must translate abstract blueprints into physically valid, task-achieving assemblies.

**Concluding remarks**. We introduced *compositional machine design*, a simplified yet challenging task that reflects core aspects of real-world machine design. To evaluate LLM performance on this task, we developed **BesiegeField**, an interactive environment based on the game Besiege. Our results with agentic systems and reinforcement learning demonstrate that LLMs hold promise for solving this problem. While we did not exhaustively explore all designs or integrate multi-modal information, our findings underscore the need to advance fundamental LLM algorithms and capabilities, and point toward exciting future directions in machine design.

## ACKNOWLEDGMENT

We sincerely thank the developers of Besiege for creating such an inspiring game and for fostering an open, vibrant player community, without which our exploration of this idea would not have been possible. We also extend our gratitude to the developers of BepInEx, whose work made it possible for us to unlock the full potential of Besiege.

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

# Appendix

## Table of Contents

# A SUPPLEMENTARY TABLES

| Models | Meta-Designer & Designer | | | Blind Refinement | | | Modification w/ Env Feedback | | |
|---|---|---|---|---|---|---|---|---|---|
| | Valid | Mean | Max | Valid | Mean | Max | Valid | Mean | Max |
| *"Catapult" Task* | | | | | | | | | |
| Gemini 2.5 Pro | 5 | 8.49 | 9.14 | 5 | 8.18 | 11.07 | 5 | 15.73 | 18.19 |
| Claude Opus 4 | 4 | 4.17 | 4.36 | 2 | 5.38 | 5.8 | 2 | 9.10 | 9.32 |
| o3 | 3 | 0 | 0 | 3 | 0 | 0 | 3 | 5.34 | 11.11 |
| Qwen3-Coder-480B-A35B | 6 | 0.75 | 4.5 | 6 | 1.61 | 5.0 | 6 | 5.21 | 6.52 |
| Doubao Seed 1.6 | 3 | 4.34 | 4.37 | 3 | 0.31 | 0.49 | 3 | 4.62 | 4.76 |
| DeepSeek-V3 | 7 | 0 | 0 | 7 | 0.98 | 3.18 | 4 | 4.82 | 4.93 |
| Kimi K2 | 6 | 7.18 | 12.02 | 2 | 5.29 | 7.36 | 0 | - | - |
| GPT-5 | 8 | 3.87 | 4.92 | 1 | 5.35 | 5.35 | 0 | - | - |
| Llama 4 Scout 17B 16E | 8 | 3.59 | 11.83 | 0 | - | - | 0 | - | - |
| *"Car" Task* | | | | | | | | | |
| Gemini 2.5 Pro | 7 | 27.85 | 38.06 | 7 | 17.81 | 39.64 | 7 | 29.96 | 41.52 |
| Claude Opus 4 | 3 | 2.96 | 3.41 | 3 | 36.18 | 37.05 | 3 | 26.59 | 38.67 |
| o3 | 8 | 29.27 | 41.43 | 2 | 20.03 | 40.04 | 2 | 28.39 | 36.18 |
| Qwen3-Coder-480B-A35B | 6 | 5.43 | 8.72 | 6 | 3.90 | 11.25 | 6 | 11.75 | 34.05 |
| Doubao Seed 1.6 | 5 | 21.80 | 29.91 | 5 | 13.25 | 26.05 | 5 | 18.75 | 26.02 |
| DeepSeek-V3 | 3 | 0.27 | 0.47 | 3 | 16.94 | 29.87 | 3 | 17.92 | 31.94 |
| Kimi K2 | 1 | 6.74 | 6.74 | 1 | 0.39 | 0.39 | 1 | 14.99 | 14.99 |
| GPT-5 | 8 | 5.67 | 20.32 | 8 | 3.75 | 9.65 | 8 | 8.43 | 13.72 |
| Llama 4 Scout 17B 16E | 4 | 1.55 | 2.00 | 1 | 0.47 | 0.47 | 1 | 0.47 | 0.47 |

Table 3: Comparison between the performance of machines generated by different stages. The mean score is computed by taking the average of the scores of all valid machines. We sample 8 machines at the designer stage and keep only the valid machines. The maximum number of retries in the following stages is thus equal to the number of valid machines produced at the designer stage.

| Models | Designer | | | Blind Refinement | | | Modification w/ Env Feedback | | |
|---|---|---|---|---|---|---|---|---|---|
| | Valid | Mean | Max | Valid | Mean | Max | Valid | Mean | Max |
| *"Catapult" Task* | | | | | | | | | |
| Gemini 2.5 Pro | 3 | 6.13 | 9.0 | 3 | 8.10 | 12.09 | 3 | 11.08 | 21.95 |
| Claude Opus 4 | 2 | 4.76 | 4.91 | 0 | - | - | 0 | - | - |
| o3 | 8 | 2.87 | 5.22 | 8 | 2.98 | 9.17 | 8 | 9.14 | 14.01 |
| Qwen3-Coder-480B-A35B | 4 | 3.5 | 9.24 | 4 | 6.39 | 10.78 | 4 | 10.2 | 12.02 |
| Doubao Seed 1.6 | 6 | 4.24 | 8.2 | 6 | 4.61 | 8.75 | 6 | 6.43 | 9.10 |
| DeepSeek-V3 | 6 | 4.67 | 4.86 | 5 | 4.33 | 4.78 | 5 | 4.91 | 5.24 |
| Kimi K2 | 3 | 6.85 | 9.05 | 2 | 8.31 | 8.97 | 2 | 11.28 | 11.39 |
| GPT-5 | 5 | 1.50 | 1.88 | 5 | 5.86 | 12.77 | 5 | 7.53 | 9.48 |
| Llama 4 Scout 17B 16E | 7 | 3.63 | 5.64 | 2 | 5.88 | 6.95 | 2 | 5.12 | 5.94 |

Table 4: Performance of machines generated after different stages of the iterative editing workflow (without meta-designer). Mean scores are computed on valid machines.

| Models | Baseline | | | w/o Parsed 3D Information | | |
|---|---|---|---|---|---|---|
| | Valid | Mean | Max | Valid | Mean | Max |
| Gemini 2.5 Pro | 5 | 8.18 | 11.07 | 5 | 7.13 | 11.36 |
| o3 | 3 | 0 | 0 | 3 | 0 | 0 |
| Qwen3-Coder-480B-A35B | 5 | 1.61 | 5.0 | 0 | - | - |
| Claude Opus 4 | 2 | 5.38 | 5.8 | 2 | 0.18 | 0.26 |

Table 5: Ablation on the effect of parsed 3D information. We compute the blind refinement score under two machine representations. The average score is computed with respect to valid machines only; 8 tries for each experiment.

| Models | Refiner Avg Retry ↓ | | Refiner validity rate ↑ | |
|---|---|---|---|---|
| | Baseline | w/o Modify History | Baseline | w/o Modify History |
| Gemini 2.5 Pro | 1.42 | **1.33** | 100% | 100% |
| o3 | 1.94 | **2.37** | 97.87% | **88.89%** |
| Qwen3-Coder-480B-A35B | 2.50 | **2.75** | 82.65% | **91.94%** |
| Doubao Seed 1.6 | 2.74 | **2.93** | 85.18% | 85.18% |
| Claude Opus 4 | 3.24 | **3.69** | 94.12% | **53.85%** |
| DeepSeek-V3 | 1.54 | **1.68** | 100% | **98.31%** |

Table 6: Ablation of edit history as refiner inputs.

| Models | Machine Validity (pass/total) | | | Designer Score | |
|---|---|---|---|---|---|
| | File Valid | 3D Valid | Final Valid | Mean | Max |
| *Baseline (construction tree)* | | | | | |
| Gemini 2.5 Pro pro | 8/8 | 5/8 | 5/8 | 8.49 | 9.14 |
| o3 | 3/8 | 3/3 | 3/8 | 0 | 0 |
| Qwen3-Coder-480B-A35B | 8/8 | 6/8 | 6/8 | 0.75 | 4.5 |
| Doubao Seed 1.6 | 7/8 | 3/7 | 3/8 | 4.34 | 4.37 |
| Claude Opus 4 | 8/8 | 4/8 | 4/8 | 4.17 | 4.36 |
| DeepSeek-V3 | 7/8 | 7/7 | 7/8 | 0 | 0 |
| Kimi K2 | 6/8 | 6/6 | 6/8 | 7.18 | 12.02 |
| Llama 4 Scout 17B 16E | 8/8 | 8/8 | 8/8 | 3.59 | 11.83 |
| *Global position-based 3D representation* | | | | | |
| Gemini 2.5 Pro pro | 5/8 | 5/8 | 5/8 | 4.96 | 12.85 |
| o3 | 0/8 | - | - | - | - |
| Claude Opus 4 | 0/8 | - | - | - | - |
| Kimi K2 | 5/8 | 4/5 | 4/8 | 0 | 0 |
| Llama 4 Scout 17B 16E | 0/8 | - | - | - | - |

Table 7: Ablation study on machine representations.

## B   DETAILS ON THE **BESIEGEFIELD** ENVIRONMENT

We built **BesiegeField** by creating plug-in modules for the game Besiege that create interfaces to allow flexible composition of parts (once certain rules are obeyed), control policies on multiple powered parts (*e.g.*. powered cogs), recording of state information of any block (*e.g.*, position, orientation, part integrity, etc.) and settings of termination conditions (*e.g.*, some part passing through a line). **BesiegeField** supports multi-process launching and thus allows for efficient parallel RL training. As the game natively supports multi-player gameplay, **BesiegeField** can naturally be applied to multi-agent RL settings. As the game Besiege (shown in Fig. 9) is built with the (mostly) open-sourced Unity3D game engine[2], **BesiegeField** is highly-customizable: the environment 1) natively supports modification of physical parameters, external forces, terrains and obstacles (*e.g.*, stone buildings) and 2) allows for extension patches (known as mods[3]) to introduce other mechanisms, such as new block types, fluid simulation and many other components.

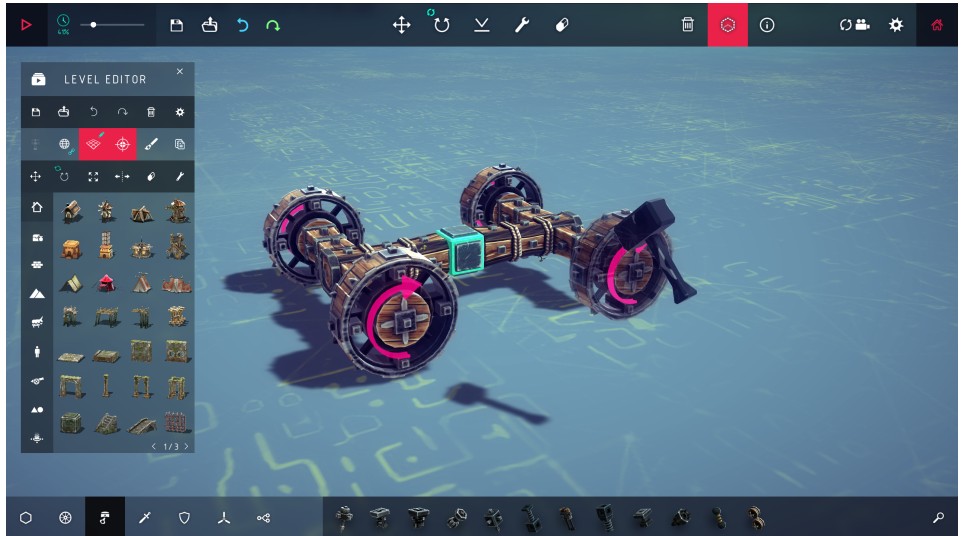

Figure 9: Besiege editor view.

### B.1   CONSTRUCTION RULE

Each machine is built by attaching new blocks to the existing structure, starting from a special root block. For convenience, we describe each construction step as an "attacher" block (child) connected to an "attachee" block (parent). As an attacher, each block has exactly one face available for connection; as an attachee, each block has none to several attachable faces. Once a face is used, it is considered occupied and cannot be reused. If, after construction, the free end of a block happens to coincide with an attachable face of an existing block, the two blocks are automatically connected.

A few special blocks violate the rule described above, such as spring. These blocks have two ends and thus must have two parent blocks, do not have physical volume can be attached to either vacant or occupied faces of other blocks.

Finally, each block can be rescaled and rotated after construction. Since post-construction scaling and rotation introduce unnecessary complexity into our pipeline, we exclude them from our experiments and leave their handling to future work.

### B.2   SIMULATION

Once constructed, the machine will be placed at the designated pose indicated by the position and orientation of the starting block (not necessary near the ground, but there is a maximum height

---

[2]https://en.wikipedia.org/wiki/Unity_(game_engine)
[3]https://en.wikipedia.org/wiki/Video_game_modding

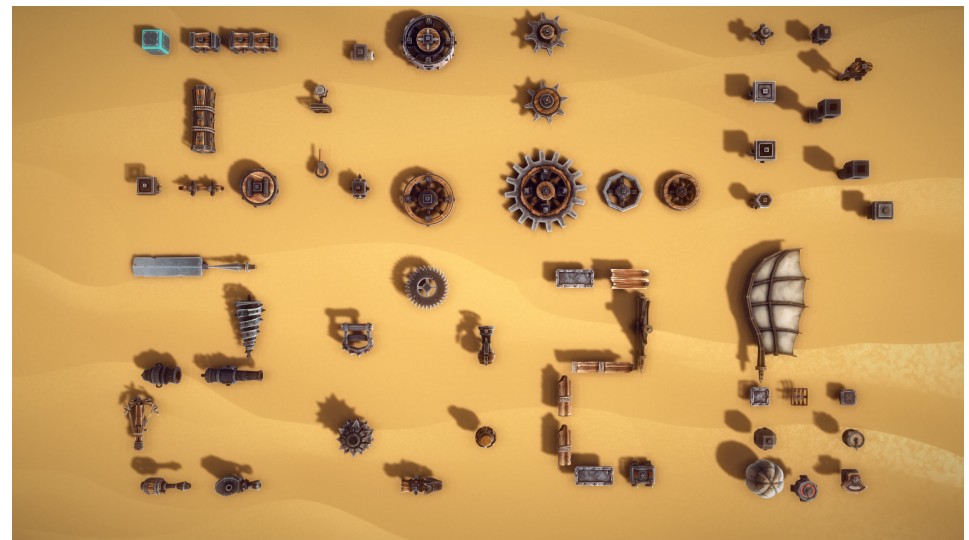

Figure 10: Common useful blocks in constructing standard machines in **BesiegeField**.

constraint). The machine will be subject to gravity and Newtonian physical laws (rigid and elastic ones) after placement.

## B.3 BLOCKS

Out of the 75 construction blocks provided by Besiege, we filter out a list of 27 blocks (shown in Fig. 10) that are most relevant to build machines with classical mechanical mechanisms such as levers and trusses.

- **Starting Block**: the root of any mechanism; initial orientation is along z+ axis.
- **Small Wooden Block**: a cubic basic construction block.
- **Wooden Block**: shaped like two small wooden blocks attached together.
- **Wooden Rod**: a slender, fragile construction block.
- **Log**: shaped like three small wooden blocks arranged in parallel.
- **Steering Hinge**: powered; controls rotation of sub-blocks, swinging left or right along the axis perpendicular to its placement axis.
- **Steering Block**: powered; rotates blocks along its placement axis.
- **Powered Wheel**: radius 1m; provides ground movement.
- **Unpowered Wheel**: identical to the powered wheel but requires external force to rotate.
- **Large Powered Wheel**: larger version of the powered wheel (radius 3m).
- **Large Unpowered Wheel**: unpowered version of the large powered wheel.
- **Small Wheel**: functions like a caster wheel (e.g., shopping cart), unpowered, 1.2m long.
- **Roller Wheel**: similar to the small wheel, but shorter (0.8m).
- **Universal Joint**: freely rotates around its placement axis, unpowered.
- **Hinge**: swings up and down along the axis perpendicular to its placement axis, unpowered.
- **Ball Joint**: swings freely in all directions, unpowered.
- **Axle Connector**: similar to a ball joint but allows unrestricted $360°$ rotation.
- **Suspension**: shaped like a wooden block, it can buffer forces from all directions.
- **Rotating Block**: powered; motor-like block that generates torque and rotates about its local y-axis.

- **Grabber**: grabs objects on contact and can release them.
- **Boulder**: a large rock, loosely attached; useful for throwing.
- **Grip Pad**: block with the highest friction.
- **Elastic Pad**: block with the highest elasticity.
- **Container**: typically used to hold a boulder.
- **Spring**: can contract; one of the special blocks that can have two parent attachments (without occupying attachable faces).
- **Brace**: reinforces structural strength.
- **Ballast**: a heavy cubic block used as a counterweight.

## B.4 TASKS

We define a set of tasks in which the goal is to construct machines within a designated building area to accomplish specific objectives.

- **Movement.** Referred to as the *car* task in the main text, the objective is to build a machine capable of driving along tracks and traversing various terrains.
- **Throw.** Referred to as the *catapult* task in the main text, the goal is to construct a machine that can launch boulders over long distances. To prevent unintended strategies (e.g., carrying the boulder instead of throwing it, or letting it roll along the ground), the building area is enclosed by a medium-height wall.

To test the robustness and richness of **BesiegeField**, we introduce additional tasks with evaluation results presented in Tab. 8 and Tab. 9. We omit the *pick* task evaluation as it requires co-design of control policies and mechanisms (creating a machine, driving it near the target, extracting mechanisms, and picking up the target), which is beyond our current scope; however, we include it to show **BesiegeField**'s potential for such evaluations. All tasks use the Hierarchical Design from Sec. 4.2 except the *gearing* task, which employs a single agent to place cogs and test for success. We forgo the Hierarchical Design for *gearing* as it is unsuitable for refinement: limited task definitions and placement locations reduce the process to random trials. Instead, we run a single agent 50 times, recording its success rate against a random baseline. The *gearing* task exemplifies contract-compositional reasoning, presenting a novel challenge beyond classic robotics. While these tasks enrich the evaluation scope and explore compositional machine spaces, they are neither more challenging nor more focused on block compositional reasoning; thus, we maintain that *car* and *catapult* are the most representative.

- **Delivery.** This task requires building a machine that can transport a large stone forward across different terrains (Fig. 16).
- **Pick.** The objective here is to design a machine that can retrieve a stone located at the bottom of a deep well (Fig. 15).
- **Jumping.** The task needs to design a machine that can jump over one or multiple obstacles in front of it (Fig. 12).
- **Lifting.** The lifting task requires designing a machine to raise a geometric solid of given weight several times (Fig. 12).
- **Flying.** The objective is to design a machine capable of achieving greater altitude or traversing multiple navigation marks while overcoming environmental barriers like strong winds (Fig. 12).
- **Gearing.** The task provides a plate or box containing a powered cog and a distant unpowered target cog. The objective is to drive the target cog's rotation using only unpowered cogs, which can be freely placed on the plate or box surface (Fig. 12).

For many of these tasks, we introduce multiple difficulty levels (not used in the experiments reported in this paper) to encourage progressively more sophisticated designs:

- **Movement and Delivery.** We consider: (1) randomized terrains with stones and wooden rods (*e.g.*, Fig. 13), (2) curved tracks (Fig. 18), and (3) obstacles such as height-limiting bars.
- **Throw.** We design: (1) varied objectives, such as requiring the boulder to pass through an aerial ring (Fig. 17) or land precisely within a small target zone, (2) environmental factors such as wind, and (3) obstacles, including height restrictions either within the building area or along the boulder's trajectory.

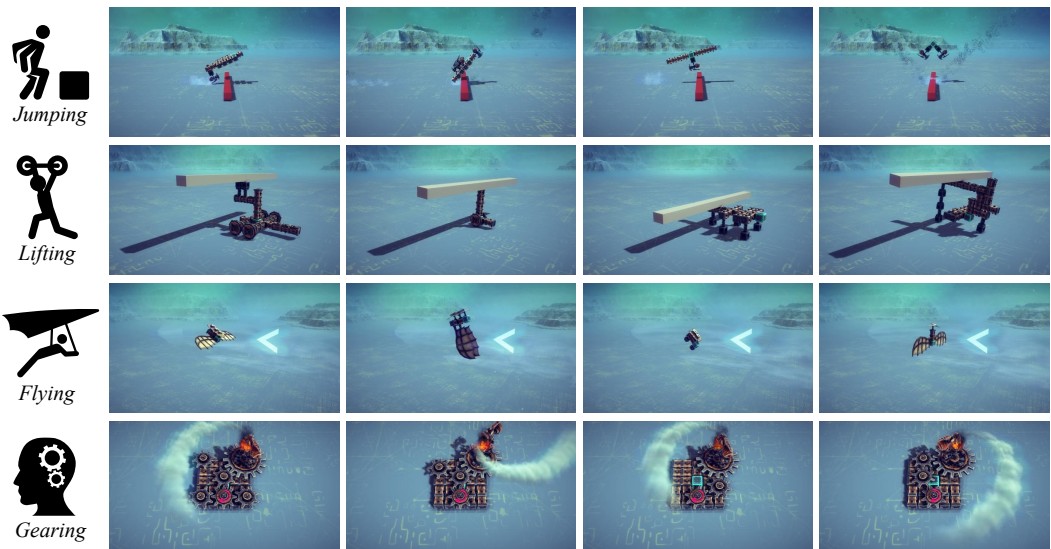

Figure 11: Illustration of the hierarchical agentic pipeline constructed machines under zero-shot tasks. All machines are generated by Gemini 2.5 Pro without finetuning.

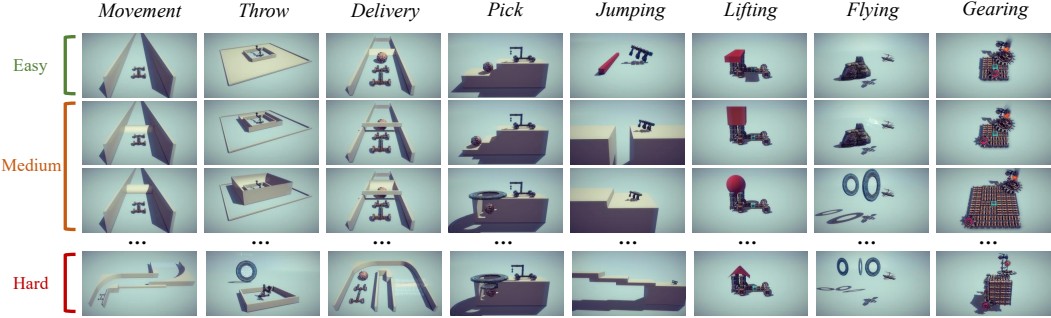

Figure 12: Illustration of the BesiegeField Tasks. We constructed 8 task types: *car*, *catapult*, *delivery*, *fetching*, *jumping*, *lifting*, *flying*, and *gearing*. For each task type, several instances were designed, spanning from easy, medium to hard. In total, BesiegeField comprises 40 tasks.

| Models | Flying (Height) | | | Jumping (Num./10) | | | Delivery (Distance) | | | Lifting (Num./10) | | |
|---|---|---|---|---|---|---|---|---|---|---|---|---|
| | Mean | Max | Std | Mean | Max | Std | Mean | Max | Std | Mean | Max | Std |
| Gemini 2.5 Pro | 30.51 | 38.50 | 10.68 | 5.50 | 10 | 3.54 | 32.57 | 45.80 | 12.07 | 9.8 | 10 | 0.4 |
| OpenAI o3 | 31.09 | 39.20 | 8.95 | 5.88 | 10 | 4.13 | 29.33 | 42.70 | 14.24 | 5.4 | 10 | 4.45 |
| Qwen3-Coder-480B-A35B | 3.73 | 6.96 | 3.22 | 1.5 | 2 | 0.5 | 0 | 0 | 0 | 5.5 | 10 | 4.5 |
| Doubao Seed 1.6 | 15.61 | 31.19 | 12.56 | 1.5 | 3 | 0.76 | 11.42 | 27.30 | 8.83 | 5.67 | 10 | 4.35 |
| Claude Opus 4 | 28.76 | 35.74 | 6.98 | 2.86 | 9 | 3.91 | 3.28 | 11.8 | 4.62 | 1.5 | 2 | 0.5 |
| DeepSeek-V3 | 0.31 | 0.51 | 0.2 | 0 | 0 | 0 | 0 | 0 | 0 | 0.75 | 1 | 0.43 |
| Kimi K2 | 1.02 | 1.52 | 0.51 | 3 | 3 | 0 | 0 | 0 | 0 | 0.33 | 1 | 0.47 |
| Llama 4 Scout 17B 16E | 0.26 | 0.51 | 0.25 | 1 | 1 | 0 | 0 | 0 | 0 | 0 | 0 | 0 |
| Human | 256.19 | 568.09 | 181.42 | 10 | 10 | 0 | 88.80 | 214.84 | 49.30 | 10 | 10 | 0 |

Table 8: Hierarchical Design performance comparison across different tasks. *Flying* task evaluates flying height, *Jumping* and *Lifting* task evaluates machine jump or lift object number under total 10 tries. *Delivery* task evaluates the machine deliver target object distance.

| Models | Gearing (Pass/50) |
|---|---|
| Gemini 2.5 Pro | 45 |
| OpenAI o3 | 38 |
| Qwen3-Coder-480B-A35B | 0 |
| Doubao Seed 1.6 | 36 |
| Claude Opus 4 | 4 |
| DeepSeek-V3 | 17 |
| Kimi K2 | 5 |
| Llama 4 Scout 17B 16E | 15 |
| Random Guess | 5 |

Table 9: *Gearing* task performance (Pass number out of 50 attempts).

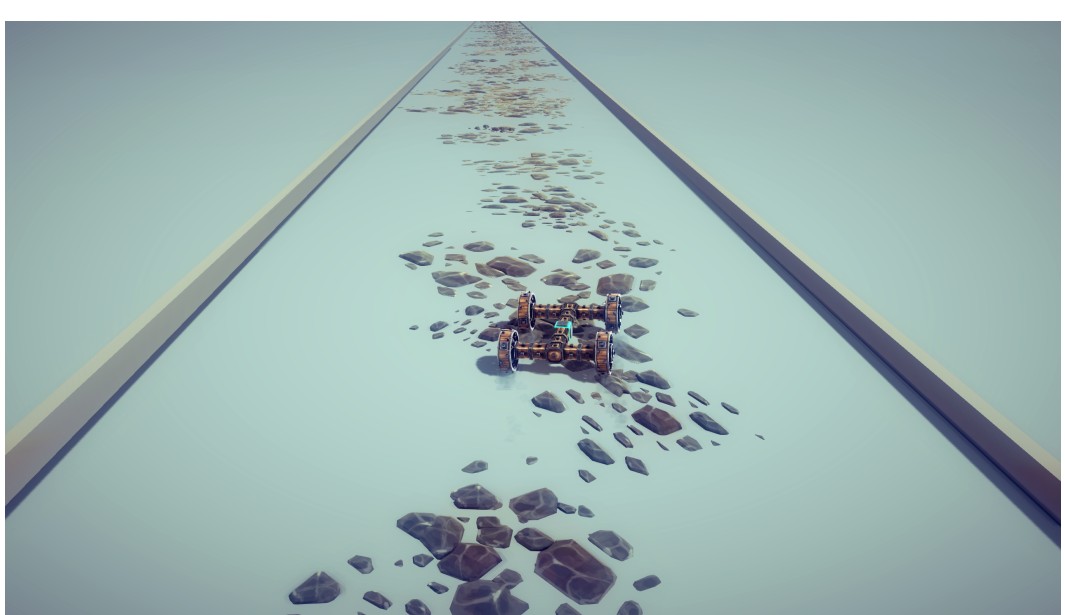

Figure 13: Illustration of the task *car / movement* on a rocky terrain, a more difficult setting compared to the environment used for the *car* task in our experiments.

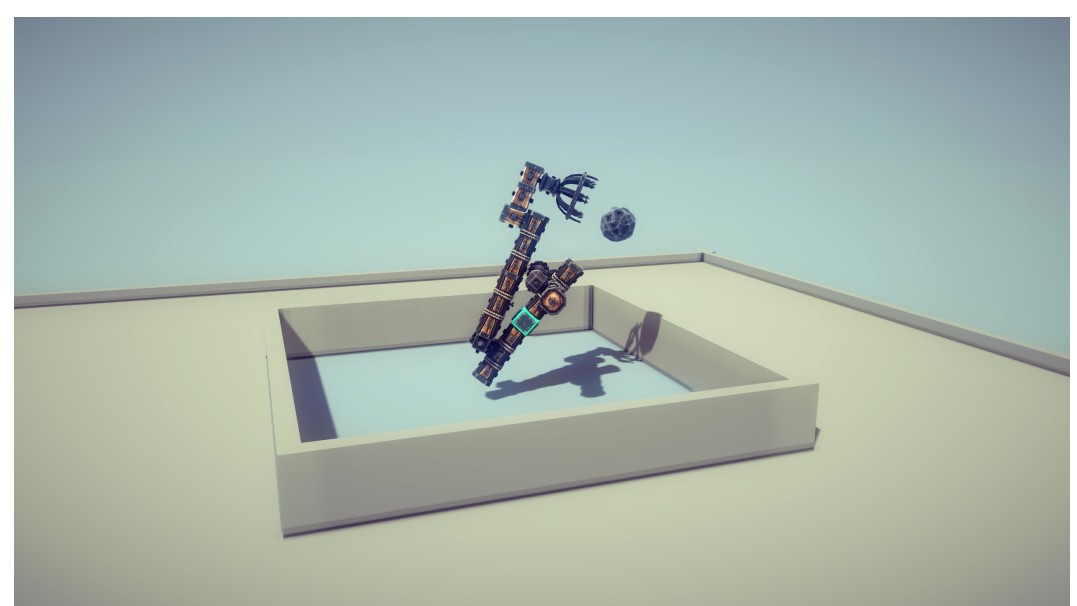

Figure 14: Illustration of the task *catapult / throw*.

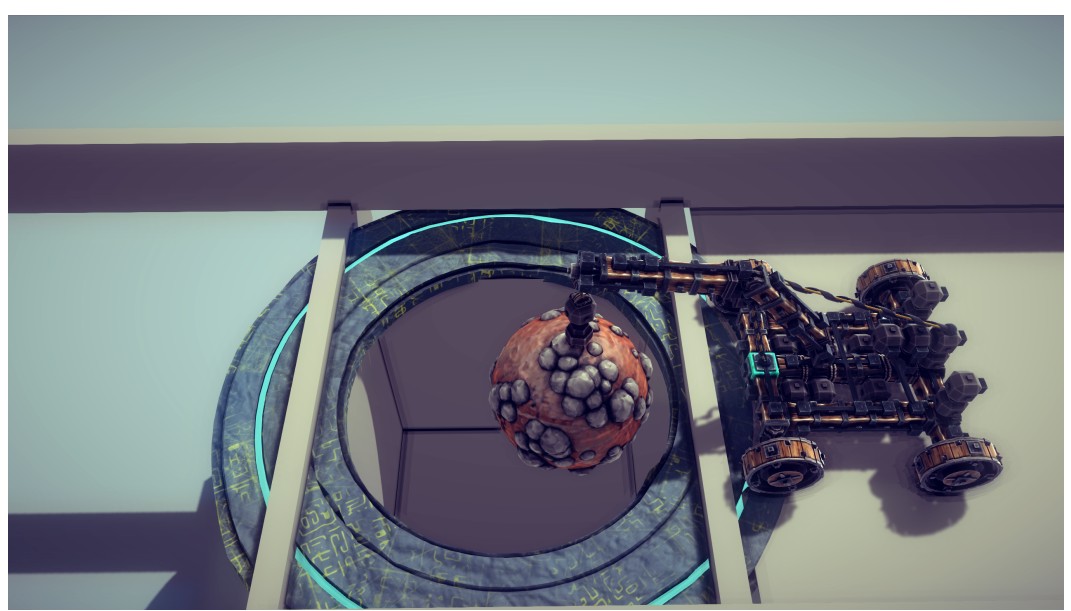

Figure 15: Illustration of the task *pick*.

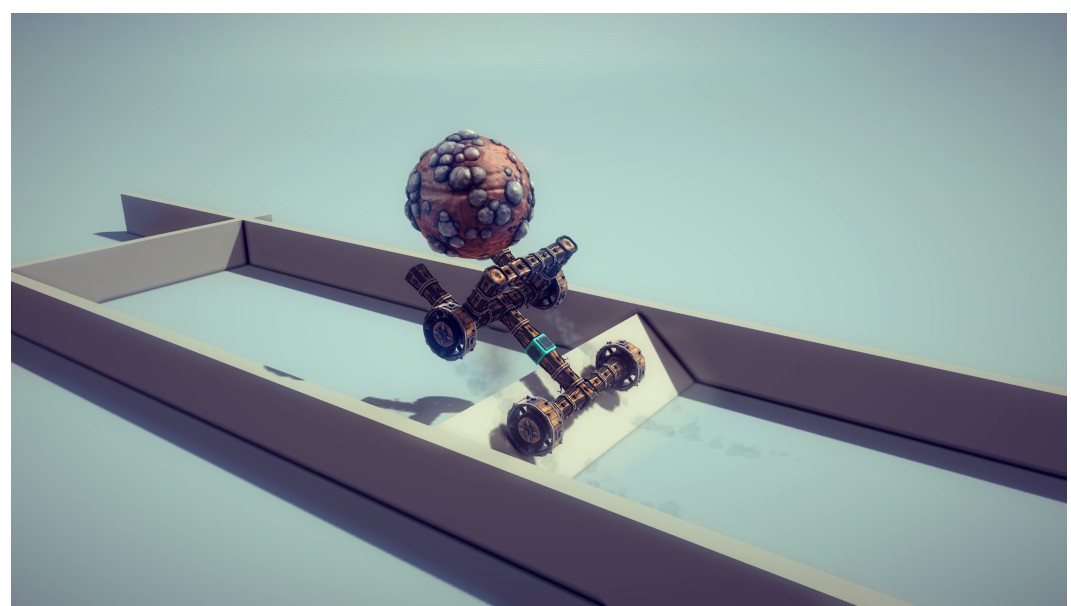

Figure 16: Illustration of the task *delivery* with a bump on the track.

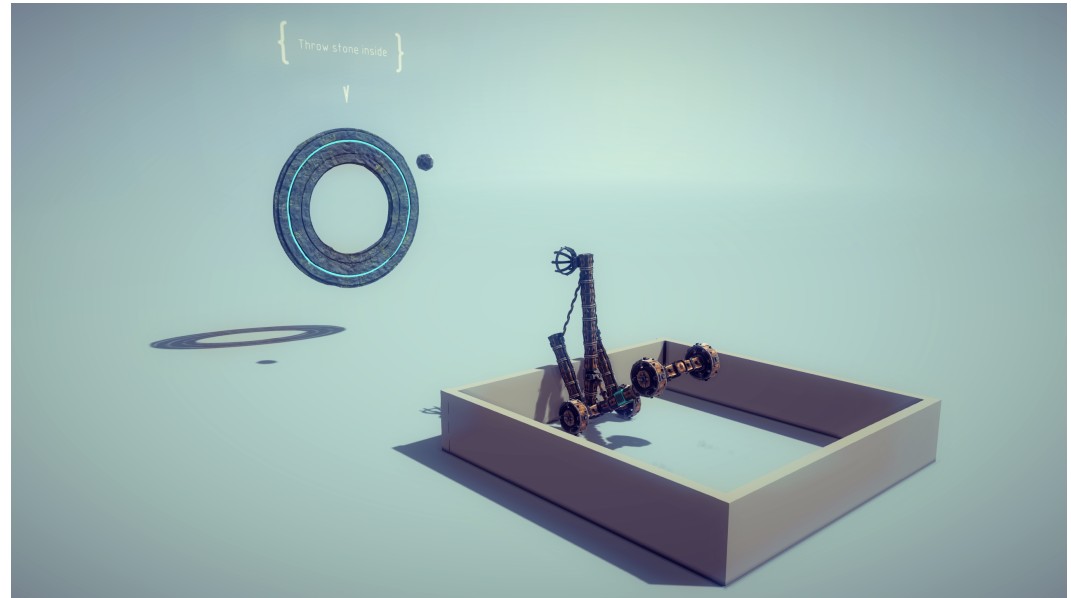

Figure 17: Illustration of the task *catapult / Throw* with the objective of throwing the boulder through the target ring.

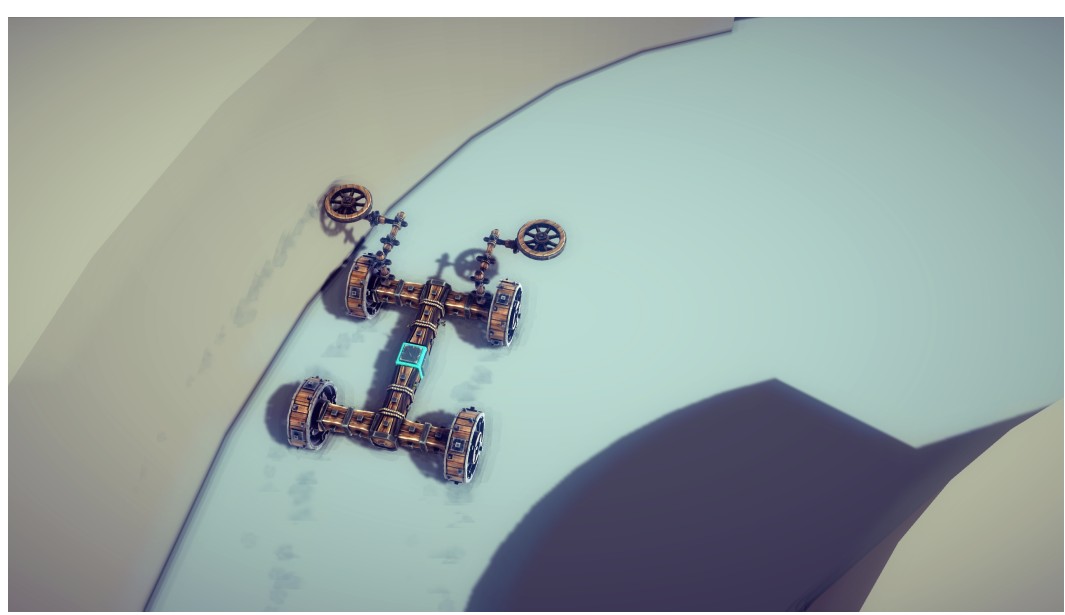

Figure 18: Illustration of the task *car / movement* with a curved track.

## C  SEARCH STRATEGIES IN MACHINE MODIFICATION LOOPS

Apart from the MCTS strategy used in the main experiments, we also evaluate two alternatives: (i) best-of-N, where we select the best-performing machine out of N candidates, and (ii) random search, which mimics best-of-N but instead selects a random candidate. For clarity, we refer to one consecutive "querier-refiner" call as a search node (consistent with our MCTS setup). Unlike classical MCTS or best-of-N, here each search node is allowed up to five retries to prevent child statistics from being too sparse. We perform $R$ search rounds, each aiming to obtain 5 valid candidate machines (though this may fail; if fewer than 5 are found, the parent node's machine is used as a candidate). Full algorithmic details are provided in Algorithm 1, Algorithm 2, and Algorithm 3. In Fig. 11 we show the improvement of machine performance with respect to the number of search rounds used. In Fig. 19 we compare the efficiency of different search methods in our agentic compositional machine design setting.

---

**Algorithm 1** Random Search Algorithm

---

**Require:** Agentic Search Node $N$
**Require:** Scoring function $S$, machine valid check function $F$
**Require:** Search Round $R$
**Ensure:** The Best result with the highest score
1: Input machine $ori\_machine$
2: $max\_retry \leftarrow 5$
3: $machine\_last\_round \leftarrow ori\_machine$
4: **for** $r = 1$ to $R$ **do**
5:     $best\_score \leftarrow -\infty$
6:     $retry \leftarrow 0$
7:     **while** $retry < max\_retry$ **do**
8:         $retry \leftarrow retry + 1$
9:         $machine\_next\_round \leftarrow N.generate(machine\_last\_round)$
10:         **if** $F(machine\_next\_round)$ **then**
11:             **break**
12:         **end if**
13:     **end while**
14:     $score \leftarrow S(machine\_next\_round)$
15:     $machine\_last\_round \leftarrow machine\_next\_round$
16: **end for**
17: **return** $(machine\_last\_round, score)$

---

| Models | Random Search | | | Best-of-N | | | MCTS | | |
|---|---|---|---|---|---|---|---|---|---|
| | Avg.I | Mean | Max | Avg.I | Mean | Max | Avg.I | Mean | Max |
| Gemini 2.5 Pro | 5 | 15.02 | 20.5 | 20 | 14.67 | 16.66 | 8 | 15.73 | 18.19 |
| Claude Opus 4 | 5 | 7.67 | 7.88 | 18 | 8.18 | 8.50 | 6 | 9.10 | 9.32 |
| o3 | 5 | 7.71 | 11.94 | 8 | 10.60 | 15.07 | 7 | 5.34 | 11.11 |
| Qwen3-Coder-480B-A35B | 5 | 4.50 | 7.64 | 11 | 5.61 | 9.87 | 8.5 | 5.21 | 6.52 |

Table 10: Ablation study on different search strategies. We compare the agentic workflow final scores. MCTS is executed for 5 rounds, with Random Search and Best-of-N also run for the same number of rounds. Avg.I denotes the average number of node expansions per search round.

**Algorithm 2** Best-of-N Algorithm

**Require:** Agentic Search Node $N$
**Require:** Scoring function $S$, machine valid check function $F$
**Require:** Search Round $R$, number of samples $n$
**Ensure:** The Best result with the highest score
1: Input machine $ori\_machine$
2: $best\_score \leftarrow -\infty$
3: $best\_machine \leftarrow ori\_machine$
4: $max\_retry \leftarrow 5$
5: **for** $r = 1$ to $R$ **do**
6:     $best\_score \leftarrow -\infty$
7:     $best\_machine\_this\_round \leftarrow best\_machine$
8:     **for** $i = 1$ to $n$ **do**
9:         $retry \leftarrow 0$
10:        **while** $retry < max\_retry$ **do**
11:           $retry \leftarrow retry + 1$
12:           $machine_i \leftarrow N.generate(best\_machine\_this\_round)$
13:           **if** $F(machine_i)$ **then**
14:             **break**
15:           **end if**
16:        **end while**
17:        $score_i \leftarrow S(machine_i)$
18:        **if** $score_i > best\_score$ **then**
19:           $best\_score \leftarrow score_i$
20:           $best\_machine \leftarrow machine_i$
21:        **end if**
22:     **end for**
23: **end for**
24: **return** $(best\_machine, best\_score)$

**Algorithm 3** Monte Carlo Tree Search (MCTS)

---

**Require:** Agentic Search Node $N$
**Require:** Root node $root$, maximum iterations $MAX\_ITER$
**Require:** Select(node): Traverse tree using UCB until leaf node
**Require:** Expand(node): Generate 4 child candidates via LLM. Validate them in parallel: keep valid ones, and regenerate invalid ones until they pass or hit the max retry limit. If all fail, use the parent node as the child.
**Require:** Simulate(node): Besiege simulation
**Require:** Backpropagate(node, reward): Update visit counts and rewards along path
**Require:** BestChild(node): Return child with highest simulation score
**Ensure:** Best action from the search tree

1: $root \leftarrow s_0$
2: $max\_retry \leftarrow 5$
3: **for** $i = 1$ to $MAX\_ITER$ **do**
4:     $retry \leftarrow 0$
5:     $node \leftarrow$ Select($root$)                                      ▷ Step 1: Selection
6:     **if** $node == root$ or $node.visited$ **then**
7:         $should\_expand \leftarrow True$
8:     **end if**
9:     **if** not $should\_expand$ **then**
10:         $child \leftarrow node$                         ▷ Unvisited leaf $node$; no children yet
11:     **end if**
12:     **while** $should\_expand$ and not all child nodes are valid **do**
13:         $retry \leftarrow retry + 1$
14:         $child \leftarrow$ Expand($node$)                     ▷ Step 2: Expansion
15:         **if** $retry \geq max\_retry$ **then**
16:             **break**
17:         **end if**
18:     **end while**
19:     $reward \leftarrow$ Simulate($child$)                  ▷ Step 3: Simulation
20:     Backpropagate($child, reward$)           ▷ Step 4: Backpropagation
21: **end for**
22: **return** BestChild($root$)                               ▷ Return best child

| Models | R2 | | R5 | | R10 | |
|---|---|---|---|---|---|---|
| | Mean | Max | Mean | Max | Mean | Max |
| Gemini 2.5 Pro | 15.04 | 17.31 | 15.73 | 18.19 | 16.44 | 18.19 |
| Claude Opus 4 | 8.61 | 9.32 | 9.10 | 9.32 | 9.43 | 9.98 |
| o3 | 5.33 | 11.11 | 5.34 | 11.11 | 8.46 | 14.52 |
| Qwen3-Coder-480B-A35B | 5.18 | 6.52 | 5.21 | 6.52 | 5.74 | 6.52 |

Table 11: Ablation study on the effect of search depth in MCTS. R2, R5, and R10 represent the running rounds of MCTS on the same search tree.

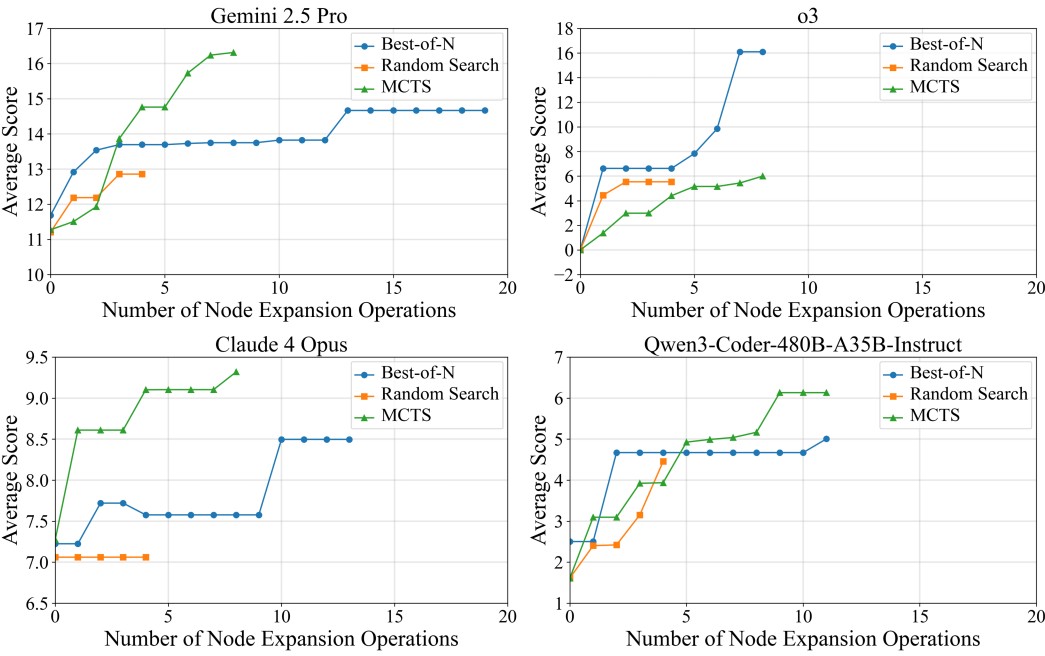

Figure 19: The variation in machine average scores with the increasing number of LLM node expansion operations under different search strategies.

## D ENVIRONMENT SETTINGS FOR AGENTIC DESIGN AND RL FINETUNING

### D.1 MACHINE REPRESENTATION

To reduce complexity in compositional machine design, our machine representation assumes all blocks remain at their default scale and are not further rotated after attachment (note: the attachment operation itself may rotate blocks).

#### D.1.1 GLOBAL POSITION-BASED REPRESENTATION

By simplifying the default XML representation that **BesiegeField** receives, we obtain the global position-based representation. Below is a concrete example:

```
[
    {"type": 0, "Position": [0,0,  0], "Rotation": [0,0,0,1]},
    {"type": 1, "Position": [0,0,0.5], "Rotation": [0,0,0,1]},
    {"type": 2, "Position": [0,0,2.5], "Rotation": [0,0,0,1]},
    {"type": 9, "Position": [0,0.5,2], "Rotation": [-0.707,0,0,0.707],
                "end-position": [0,2,0]}
]
```

Basically, each block in the machine is independently recorded without mentioning its adjacent blocks. For most of the block types, only the block type and its pose (position + orientation) are recorded. For special blocks that have two parents, the other end has to be specified, for which the corresponding dictionary has an additional entry of "end-position".

#### D.1.2 CONSTRUCTION TREE REPRESENTATION

With our parsimonious construction tree representation, the example machine above is represented by the following the following JSON list:

```
[
    {"type": 0, "id": 0, "parent"   : -1, "face_id"   : -1},
    {"type": 1, "id": 1, "parent"   :  0, "face_id"   :  0},
    {"type": 2, "id": 2, "parent"   :  1, "face_id"   :  0},
    {"type": 9, "id": 3, "parent_a":  0, "face_id_a":  4,
                         "parent_b":  1, "face_id_b":  6}
]
```

Specifically, the ordered list of dictionaries of the machine construction JSON file represents the construction order of blocks. Each dictionary contains the following information of corresponding block: 1) "type": block type; 2) "id": the order ID of this block (the same as the order in the list), included so that LLMs do not have to parse it by itself; 3) "parent", the ID of its parent block; 4) "face_id", the face of the block's parent to which the block is attached. In cases that the block has two parents (*e.g.*, a string that connects both parts), we use "parent_a" and "parent_b" to record both parents; similar for "face_id".

Note: the first block with "id" 0 is always the unique starting block, of which the local position and rotation are always zero.

### D.2 REWARD SETTING

Here we elaborate on the reward design for RL experiments in Sec. 5.1. Our reward is in the form of $R = \texttt{is\_valid} \times \texttt{performance}$ where $\texttt{is\_valid}$ is the boolean representing machine validity and $\texttt{performance}$ is the task-specific performance metric.

***Car.*** We set $\texttt{is\_valid}$ to 1 as long as the policy produces a machine that can be parsed from the generated construction tree and can be successfully placed into the environment without any self-collision; otherwise it is set to 0. $\texttt{performance}$ is set to the distance between the starting position and the end position of the root block.

***Catapult.*** For `is_valid` to be 1 in this task, the machine has to satisfy an additional constraint compared to the *car* task: the maximum height of the boulder position during simulation must be greater than a threshold of 3m. As explained in the main text, `performance` for *catapult* is the product of the maximum height and maximum distance (towards some pre-defined direction) during simulation.

### D.3 ENVIRONMENT FEEDBACK

In principle, we are able to obtain all state variables of each single part of a simulated machine. Due to the space complexity of the simulation results, not all information can be fed to LLM agents. Here we consider a minimal set of environment feedback information that the environment querier always gathers and returns to the refiner. Below are the minimal set of feedback information for each task:

*Car.* 1) machine orientation; 2) machine maximum moving distance (towards a designated direction); 3) machine max speed; 4) machine average speed per second; 5) machine position per 0.2 second (atomic time).

*Catapult.* 1) boulder maximum distance (horizontal, towards a designated distance); 2) boulder maximum height; 3) boulder position per 0.2 second (atomic time).

Beyond these basic pieces of feedback, the querier, after seeing the candidate machine and its simulation results, in our default setting selectively extract a subset of environment feedback given its speculation on the issues of the simulated machine. For instance, parts during simulation may collide with each other and break. Such behavior carries important hints on why machines fail, and an LLM agent with sufficient capability in spatial and physics understanding can possibly identify the vulnerable blocks in the design.

Below we elaborate on the additional information that the querier may gather:

- Block index to query;
- Time interval of interest (states outside this interval will not be considered);
- Feedback type of interest (one or more from the list)
  - Position;
  - Orientation;
  - Velocity;
  - Length (for spring only)

## E CHALLENGES IN COMPOSITIONAL MACHINE DESIGN

### E.1 FAILURE PATTERNS

Generated machines often fail in systematic ways. As shown in Fig. 20, we observe several recurring categories of errors, including flawed reasoning, structural attachment errors, incorrect part orientations and failures in instruction following. These diverse failure types highlight both the reasoning and execution challenges inherent in compositional machine design.

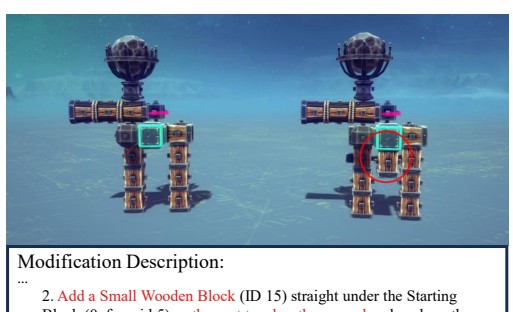

Modification Description:
...
2. Add a Small Wooden Block (ID 15) straight under the Starting Block (0, face_id 5) so the root touches the ground and anchors the frame.

(a) Flawed high level reasoning

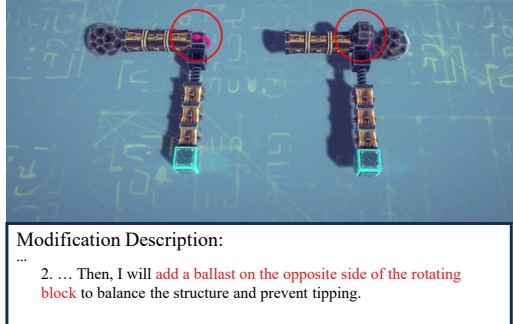

Modification Description:
...
2. ... Then, I will add a ballast on the opposite side of the rotating block to balance the structure and prevent tipping.

(b) Incorrect parents

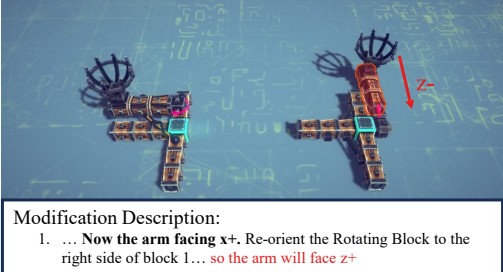

Modification Description:
1. ... **Now the arm facing x+.** Re-orient the Rotating Block to the right side of block 1... so the arm will face z+
...

(c) Incorrect part orientations

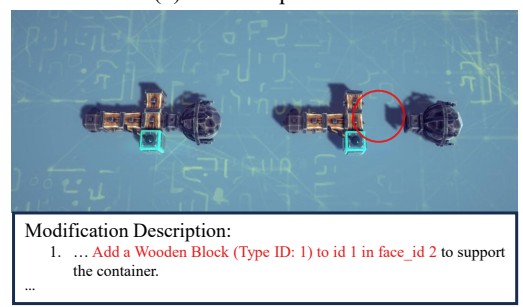

Modification Description:
1. ... Add a Wooden Block (Type ID: 1) to id 1 in face_id 2 to support the container.
...

(d) Instruction following failures

Figure 20: Examples to illustrate failure patterns. In each example, the original machine is shown on the left and the modified machine on the right. Failure patterns are sampled from Qwen3-Coder-480B-A35B-Instruct.

### E.2 NEED FOR PRECISION

In Fig. 21 we present a simple example to illustrate how the task of compositional machine design requires high precision in the spatial design of configurations of different parts. Even though the high-level design is feasible, the machine in the top row fails to throw the boulder out due to the incorrect position of the container.

### E.3 APPEARANCE VS. PERFORMANCE

As illustrated in Fig. 24, a machine's appearance does not necessarily reflect its actual performance. A design that seems well-aligned with human intuition can fail dramatically, while one that looks awkward or unintuitive may achieve superior results. For LLMs to design machines that are both effective and visually intuitive to humans, reward functions must account not only for task performance but also for stability and other factors that shape human perception of functionality.

### E.4 MACHINE DIVERSITY

Generating diverse machines is essential for compositional machine design. To evaluate this, we selected three high-performing LLMs: Gemini 2.5 Pro, o3, and Kimi-k2, and we sampled 20 valid machines from LLM results in the *catapult* task. We then assessed their diversity and compared it

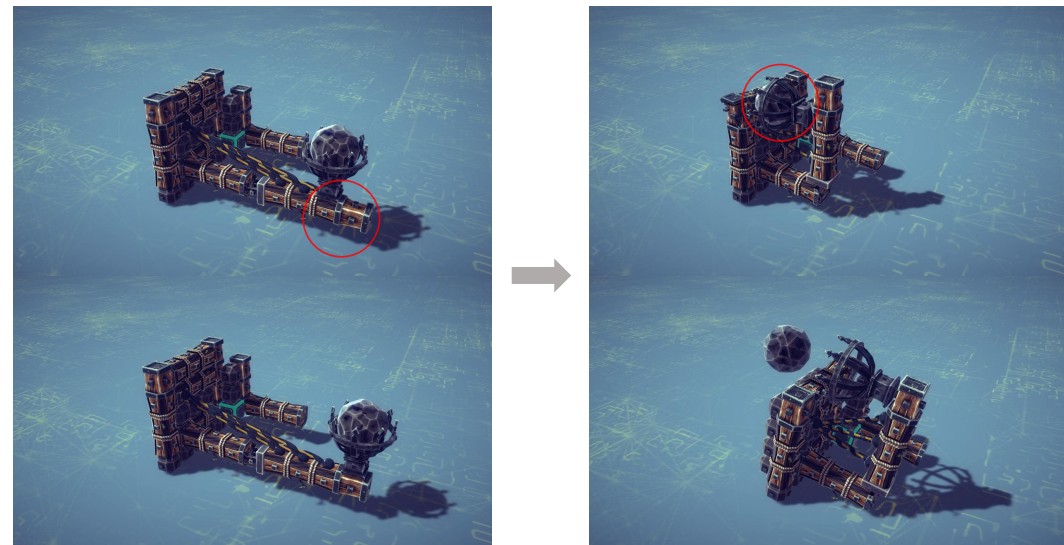

Figure 21: Illustration of how machines built with feasible high-level designs may fail due to inaccurate part placement. The upper row shows machines sampled from Gemini 2.5 Pro; the lower row shows manually modified versions. Left: designed machines; Right: simulation results.

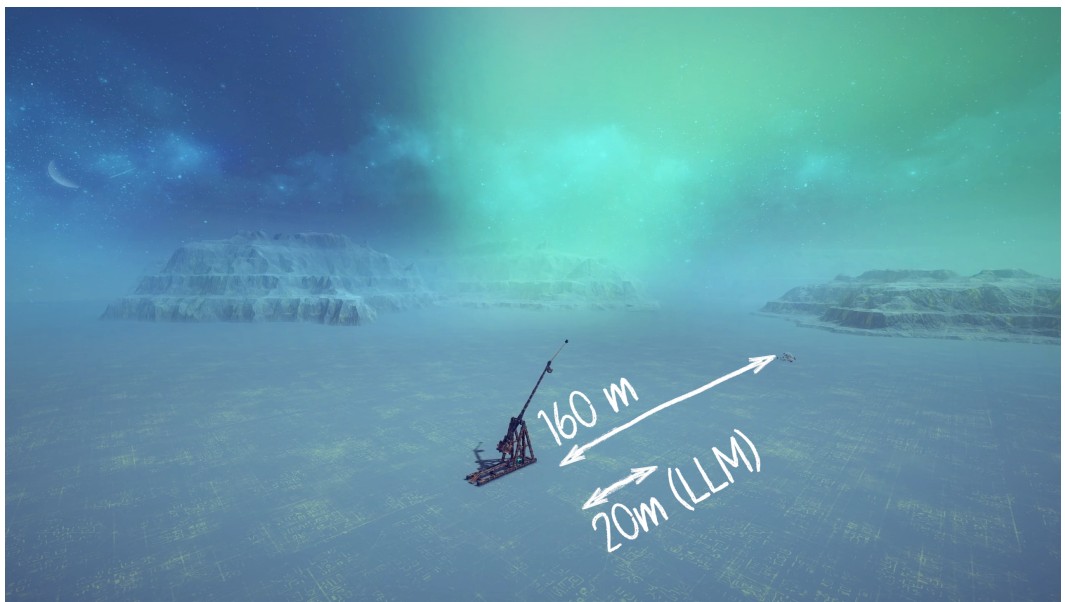

Figure 22: Human-constructed catapult vs LLM-designed catapult.

with that of 20 randomly sampled human-made catapults from Steam Workshop [4]. Additionally, we sampled 20 machines every 40 steps during the base RL training process and evaluated their diversity against the same human-made catapults. Specifically, we employed the following metrics:

**Shape diversity** was measured using Chamfer Distance (CD) (Yuan et al., 2018), Hausdorff Distance (HD) (Huttenlocher & Rucklidge, 1992), and Intersection-over-Union (IoU) (Qi et al., 2017). CD computes the bidirectional average point-to-nearest-point distance, providing a smooth global similarity score; HD captures the maximum such distance, indicating worst-case deviation; IoU voxelizes the 3D space to quantify the volume overlap ratio. Each machine was converted to a point

---
[4] https://steamcommunity.com/workshop/

| Model Name | Chamfer Dist (↑) | Hausdorff Dist (↑) | IoU (↓) | Graph Emb Similarity(↓) | Tree Edit dist(↑) |
|---|---|---|---|---|---|
| Gemini2.5Pro | 0.287 | 0.604 | 0.247 | 0.520 | 9.534 |
| o3 | **0.311** | **0.612** | 0.235 | 0.542 | 9.796 |
| kimik2 | 0.230 | 0.480 | 0.350 | **0.392** | 13.513 |
| Human | 0.275 | 0.538 | **0.206** | 0.505 | **15.712** |
| **RL Steps** | Chamfer Dist (↑) | Hausdorff Dist (↑) | IoU (↓) | Graph Emb Similarity(↓) | Tree Edit dist(↑) |
| 40 | 0.370 | 0.646 | 0.264 | 0.660 | 10.876 |
| 80 | 0.320 | **0.661** | 0.292 | 0.725 | 6.142 |
| 120 | 0.360 | 0.577 | 0.307 | 0.646 | 4.858 |
| 160 | 0.196 | 0.484 | 0.387 | 0.722 | 4.130 |
| 200 | 0.326 | 0.586 | 0.322 | 0.726 | 4.923 |
| 240 | **0.377** | 0.641 | 0.214 | 0.672 | 5.701 |
| 280 | 0.110 | 0.401 | 0.621 | 0.760 | 2.884 |
| 320 | 0.095 | 0.338 | 0.742 | 0.767 | 1.630 |
| 360 | 0.276 | 0.525 | 0.390 | 0.729 | 2.381 |
| 400 | 0.226 | 0.460 | 0.473 | 0.737 | 2.356 |
| Human | 0.275 | 0.538 | **0.206** | **0.505** | **15.712** |

Table 12: Diversity quantitative comparison of agentic systems, RL Qwen-2.5-14B-Instruct models and human creation under *"catapult"* task. Bold numbers mean the best performance in its column, and underlined number is the second best.

cloud and evaluated against all other machines generated by the same LLM, with the per-metric averages taken as the final diversity measures.

**Topology diversity** was assessed via Graph Embedding Similarity and Tree Edit Distance (TED) (Zhang & Shasha, 1989). All machines were represented as trees (human-authored designs were first parsed into tree structures). For Graph Embedding Similarity, each tree was treated as an unweighted graph and embedded using Node2Vec (Grover & Leskovec, 2016); we then averaged the cosine similarities between all pairs of graphs produced by the same LLM. TED complements this by computing the minimum-cost sequence of node insertions, deletions, and substitutions required to transform one tree into another, yielding a topology-aware dissimilarity score sensitive to node-label variations.

As shown in Table 12, LLMs can generate machines with diversity comparable to human creators. However, we observed that both entropy (Fig. 23) and diversity collapse during RL training. While generating task-completing machines with LLMs remains challenging, preserving diversity during RL training presents a promising direction for future research.

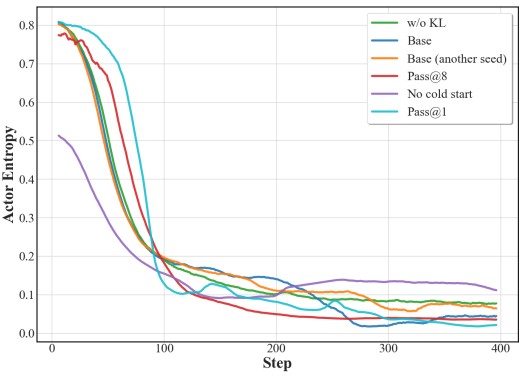

Figure 23: *Catapult task*. Entropy collapses with RL step increases.

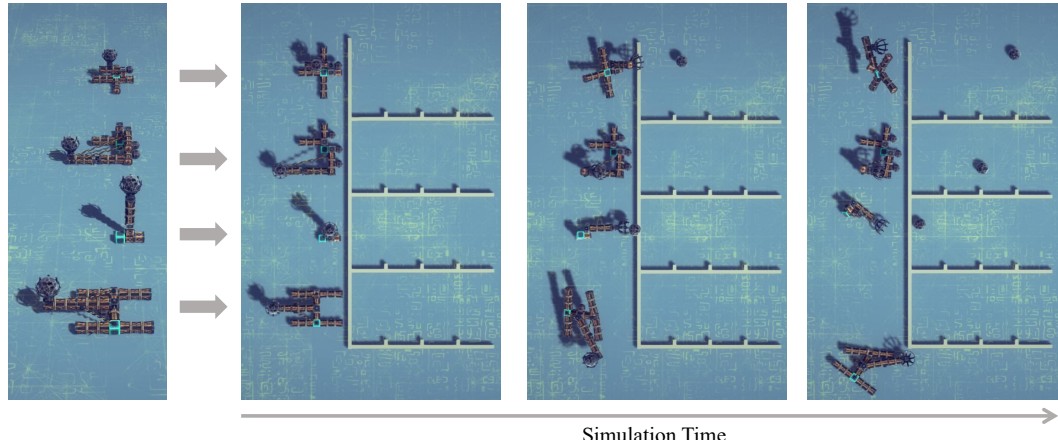

Simulation Time

Figure 24: Boulder-throwing trajectories for various machine designs generated by Gemini 2.5 Pro. From left to right, each row first shows the machine design, followed by a time-lapsed bird's-eye view of its throw.

# F  Settings for RL Finetuning

## F.1  Cold-Start Dataset Curation

Noticing that Gemini 2.5 Pro produces the most satisfactory machines with reasonable CoT, we adopt the single-agent generation setting and collect Gemini-generated machines along with their CoT. We curated 100 design objectives: 75 captions of machines created by Besiege players from the Internet and 25 authored by us. These 25 prompts are constructed by 1) first writing down simple design objectives that are realizable by **BesiegeField** and can emerge from some simple rewards, and 2) then introducing environment constraints and machine-specific requirements. Using this prompt dataset, we generate 250 machines per prompt, and after filtering out inappropriate ones (those that fail to parse, cannot be built in the environment, or do not have a specific physics-driven functionality, *e.g.*, a statue), we obtain 9,984 machines with their corresponding CoT. A sample gallery is shown in Fig. 25.

We present examples in the curated prompt set:

```
1. Build a machine that can provide an exciting spinning amusement ride
experience.
2. Build a machine that can mimic the movements of a humanoid figure for
entertainment or functional demonstrations.
3. Build a machine that can glide smoothly over snow or ice.
```

Below we present the text prompts with our simple authoring strategy, which can possibly be scaled with LLMs:

```
-Additional Environment Constraints-
1. On an uneven, bumpy straight road, build a small car that must travel
in a straight line to the finish.
2. On a straight road stands a stone wall; build a battering ram that
must accelerate straight ahead, smash the wall, and finish with minimal
damage to the machine.
-Modified Demands for Target Machines-
1. Build a tall tower that must keep a heavy block (id 36) at 15 m height
 for 5 s without collapse.
2. On a straight road stands a 10 m high wall; build a siege ladder that
must advance, extend its top above the wall, and remain upright
throughout.
```

## F.2  Cold-Start Details

In our experiment, we use Qwen2.5-14B-Instruct as the base model and train it on the Gemini synthesized dataset. To save GPU memory, we employ the parameter-efficient quantized OFT (QOFT) technique (Qiu et al., 2025c; 2023; Liu et al., 2024a; Qiu et al., 2025a) for updating the model parameters, with OFT block size 64. We use 8-bit training with the 8-bit AdamW optimizer implmented with bitsandbytes (Dettmers et al., 2022), a learning rate of 1e-6 and a linear warmup schedule (3% of the total training steps).

## F.3  RL Experiment Details

We use verl framework to implement our RL experiments. The LLM is finetuned from Qwen2.5-14B-Instruct (with LoRA of rank 64 on all linear layers) using the Gemini-synthesized dataset described above. We set learning rate to 5e-6 with gradient clipping threshold set to 0.5. The GRPO advantage estimator uses an advantage clipping ratio of 0.2. We add a KL penalty (weight 0.001) with respect to the pretrained LLM and do not introduce any entropy regularization. For rollouts, we use a temperature of $1.0$ and top-$p$ value of 0.95. Maximum input and output lengths are 3440 and 1168 tokens, respectively. We train each model for 400 update steps which take approximately 48 hours.

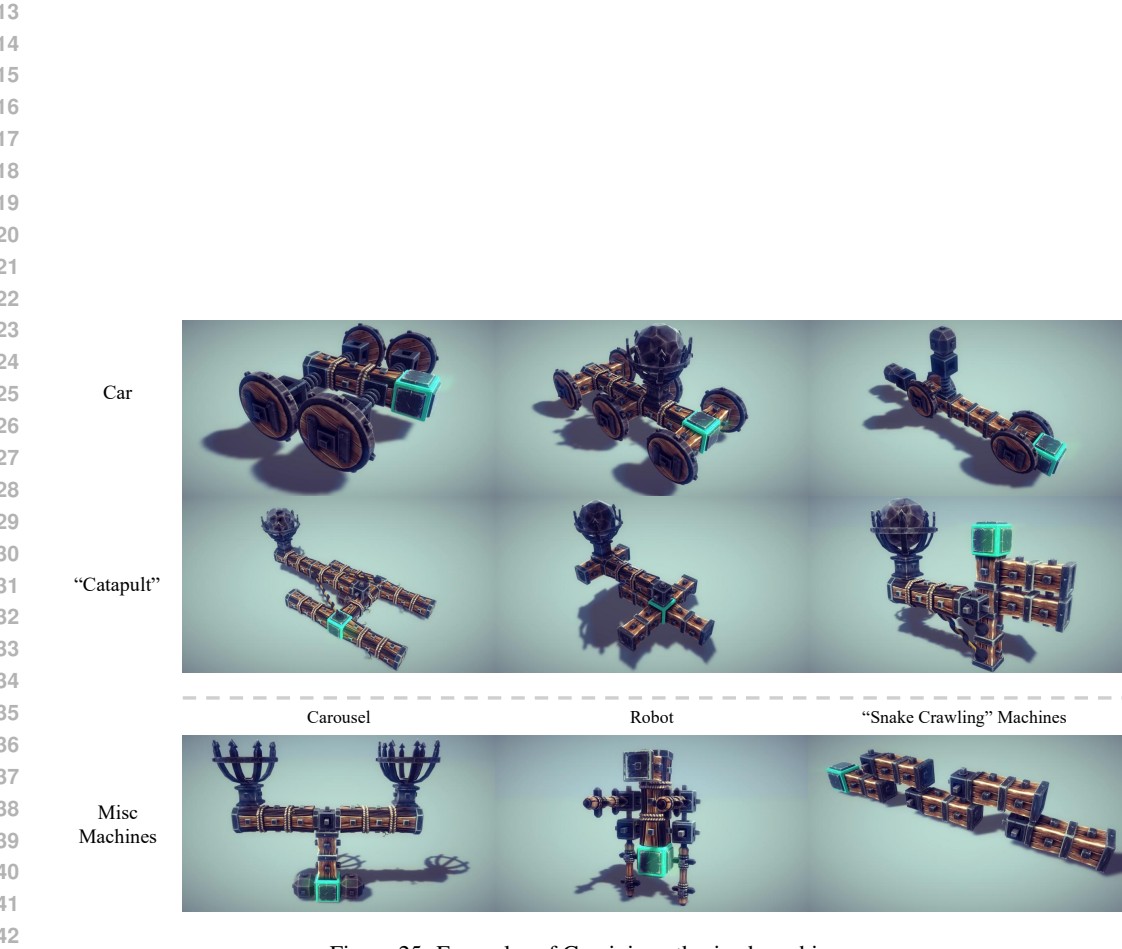

Figure 25: Examples of Gemini-synthesized machines.

# G  ADDITIONAL ABLATION STUDIES

**Meta-Designer in hierarchical design.** In Table 13, we show that how a meta-designer for hierarchical design may benefit compositional machine design. Leveraging the knowledge on existing machines, meta-designers can identify the key macro-level mechanical components that are easier to design compared to the whole task, as shown in the results for Gemini 2.5 Pro, Kimi K2 and Llama 4 Scout. However, introducing an additional stage can introduce compounding error and, if the LLM agent is not capable of integrating different macro-level mechanical components, they may lead to lower scores, which we hypothesize is the reason for the failure of hierarchical design in models like Qwen3. Moreover, we examine if the meta-designer should provide step-by-step building instruction for the designer (Fig. 26), or simply provide high-level mechanical component descriptions. We find that a meta-designer that provides more detailed information is beneficial mostly when the base model is powerful enough (*e.g.*, Gemini 2.5 Pro).

**Effect of feedback-free self-critic.** In Table 14, we show that the inspector agent which does self-critic before running any environment simulation tend to improve performance for models like Gemini 2.5 Pro (the most powerful model for the task of compositional machine design in **BesiegeField**) but can fail drastically for models like o3.

**Effect of active feedback queries.** In Table 15, we show that the active queries on the environment feedbacks help most of the models achieve better performance, compared to the setting with no environment feedback and that with only environment simulation final scores.

**Additional RL results.** In Fig. 27 and 29 and , we show the maximum scores achieved in the environments with different RL methods plus the validity rate of machines. We visualize the maximum score since, in the case when one is allowed to use inference-time scaling techniques, the best performing machines are the ones people care most about. We show that our settings with Pass@64 training achieves the best maximum score with two different random seeds. In additiona, in Fig. 31, we visualize the corresponding Best@N metrics.

For completeness, we also visualize the results with our default setting on the task *car* in Fig. 28, 30 and 32.

---

**Meta Designer:**

"Machine Structure": {
  ...
  "Base Frame": {
    ...
    "Guidance": "Wooden Blocks (ID 1) and Small Wooden Blocks (ID 15) are used to build a wide, heavy base that prevents tipping. They also form a tower to elevate the pivot point and a physical stop to arrest the arm's rotation, which is crucial for the release mechanism."
  },
  "Throwing Mechanism ": {
    ...
    "Guidance": "A Rotating Block (ID 22) provides the high-speed rotational power. This is attached to a Log (ID 63), which acts as a long, robust lever arm. A Container (ID 30) is placed at the end of the arm to hold the Boulder (ID 36) projectile."
  }
}

**Detailed Meta Designer:**

"Machine Structure": {
  ...
  "Base Frame": {
    ...
    "Guidance ": "First, build the foundation. Attach a Log (63) to the left side of the Start Block (0) and another Log to the right side to create a wide horizontal base. Then, place a Small Wooden Block (15) on top of the Start Block. This will elevate the pivot point of the throwing mechanism for a better launch angle and provide a central connection point."
  },
  "Throwing Mechanism": {
    ...
    "Guidance": "On top of the Small Wooden Block from the Base Frame, place a Rotating Block (22). This will serve as the powered pivot. Attach a Log (63) to the rotating part of the Rotating Block, positioning it as the throwing arm. At the far end of this Log, attach a Container (30) with its opening facing upwards. Finally, place the Boulder (36) inside the Container."
  }
}

Figure 26: Construction guidance comparison of *Meta Designer* and *Detailed Meta Designer*, sampled with Gemini 2.5 Pro.

| Models | Machine Validity (pass/total) | | | Designer Score | |
|---|---|---|---|---|---|
| | File Valid | 3D Valid | Final Valid | Mean | Max |
| *Baseline (Meta-Designer & Designer)* | | | | | |
| Gemini 2.5 Pro | 8/8 | 5/8 | 5/8 | 8.49 | 9.14 |
| o3 | 3/8 | 3/3 | 3/8 | 0 | 0 |
| Qwen3-Coder-480B-A35B | 8/8 | 6/8 | 6/8 | 0.75 | 4.5 |
| Doubao Seed 1.6 | 7/8 | 3/7 | 3/8 | 4.34 | 4.37 |
| Claude Opus 4 | 8/8 | 4/8 | 4/8 | 4.17 | 4.36 |
| DeepSeek-V3 | 7/8 | 7/7 | 7/8 | 0 | 0 |
| Kimi K2 | 6/8 | 6/6 | 6/8 | 7.18 | 12.02 |
| Llama 4 Scout 17B 16E | 8/8 | 8/8 | 8/8 | 3.59 | 11.83 |
| *Single Agent* | | | | | |
| Gemini 2.5 Pro | 6/8 | 3/6 | 3/8 | 6.13 | 9.00 |
| o3 | 8/8 | 8/8 | 8/8 | 2.87 | 5.22 |
| Qwen3-Coder-480B-A35B | 8/8 | 4/8 | 4/8 | 3.5 | 9.24 |
| Doubao Seed 1.6 | 7/8 | 6/7 | 6/8 | 4.24 | 8.2 |
| Claude Opus 4 | 8/8 | 2/6 | 2/8 | 4.76 | 4.91 |
| DeepSeek-V3 | 7/8 | 6/7 | 6/8 | 4.67 | 4.86 |
| Kimi K2 | 8/8 | 3/8 | 3/8 | 6.85 | 9.05 |
| Llama 4 Scout 17B 16E | 8/8 | 7/8 | 7/8 | 3.63 | 5.64 |
| *w/ detailed Meta-Designer & Designer* | | | | | |
| Gemini 2.5 Pro | 8/8 | 7/8 | 7/8 | 9.19 | 11.94 |
| o3 | 7/8 | 6/7 | 6/8 | 0.92 | 1.18 |
| Qwen3-Coder-480B-A35B | 8/8 | 2/8 | 2/8 | 4.87 | 4.87 |
| Doubao Seed 1.6 | 0/8 | - | - | - | - |
| Claude Opus 4 | 7/8 | 5/7 | 5/8 | 4.13 | 4.79 |
| DeepSeek-V3 | 8/8 | 8/8 | 8/8 | 6.12 | 9.0 |
| Kimi K2 | 8/8 | 0/8 | - | - | - |
| Llama 4 Scout 17B 16E | 8/8 | 7/8 | 7/8 | 4.01 | 6.93 |

Table 13: Ablation study on the meta-designer. Machine validity is evaluated in two aspects: file validity, 3D validity. Note that 3D validity requires the machine to first pass file validity. Final validity refers to a fully valid machine (satisfying both file and 3D validity). The mean simulation score is calculated based solely on the final valid outputs. *Detailed Meta-Designer* provides more concisely, step-by-step construction guidance to the *Designer*. Compared to *Baseline*, *Single Agent* is slightly harder to construct valid machines, but the simulation scores are better. *Detailed Meta-Designer* improves both metrics, but requires LLMs to have a strong 3D understanding and a large context window. The comparison between *Meta-Designer* and *Detailed Meta-Designer* is illustrated in Fig. 26.

| Models | Blind Refinement Simulation Scores | | | | | |
| --- | --- | --- | --- | --- | --- | --- |
| | Baseline | | | w/o Inspector | | |
| | Valid | Mean | Max | Valid | Mean | Max |
| Gemini 2.5 Pro | 5 | **8.18** | **11.07** | 5 | 5.67 | 9.37 |
| o3 | 3 | 0 | 0 | 3 | **3.08** | **9.24** |
| Qwen3-Coder-480B-A35B | 6 | **1.61** | **5.0** | 6 | 0.75 | 4.51 |
| Doubao Seed 1.6 | 3 | 0.31 | **0.49** | 3 | **0.47** | 1.41 |
| Claude Opus 4 | 2 | **5.38** | 5.8 | 4 | 5.20 | **8.25** |
| DeepSeek-V3 | 7 | **0.98** | **3.18** | 7 | 0.38 | 2.16 |
| Kimi K2 | 2 | **5.29** | 7.36 | 6 | 2.31 | **8.91** |
| Llama 4 Scout 17B 16E | 0 | - | - | 0 | - | - |

Table 14: Ablation study on inspector agentic design. The mean simulation score is calculated based solely on the valid machines after blind refinement. Removing the inspector from the agentic flow lowers the blind refiner's mean performance on LLMs with weaker 3D understanding, while barely affecting other models.

| Models | Refiner Simulation Scores | | | | | | | | |
| --- | --- | --- | --- | --- | --- | --- | --- | --- | --- |
| | Baseline | | | w/o Env Querier | | | Score Only | | |
| | std | Mean | Max | std | Mean | Max | std | Mean | Max |
| Gemini 2.5 Pro | 2.47 | **15.73** | 18.19 | 4.18 | 14.89 | **19.77** | 2.05 | 9.68 | 13.18 |
| o3 | 5.36 | 5.34 | **11.11** | 4.24 | 4.06 | 8.55 | 2.76 | **7.05** | 10.24 |
| Qwen3-Coder-480B-A35B | 0.95 | **5.21** | **6.52** | 2.32 | 4.05 | 6.89 | 2.53 | 2.81 | 5.56 |
| Claude Opus 4 | 0.31 | **9.10** | **9.32** | 0.82 | 8.50 | 9.08 | 0.42 | 5.75 | 6.05 |
| Doubao Seed 1.6 | 0.23 | 4.62 | 4.76 | 0.08 | **4.89** | 4.94 | 0.29 | 4.79 | **5.05** |
| DeepSeek-V3 | 0.09 | **4.82** | 4.93 | 1.96 | 4.37 | **6.00** | 1.91 | 2.76 | 5.15 |

Table 15: Ablation study on the environment querier agent. For the refiner, the baseline includes simulation scores, basic environment feedback, and querier-required feedback. The "w/o env querier" setting provides only simulation scores and basic environment feedback. In the "pure score only" setting, only simulation scores are provided. Removing the environment querier causes a slight drop in average machine performance. With reward signals only, the performance markedly degrades across most LLMs.

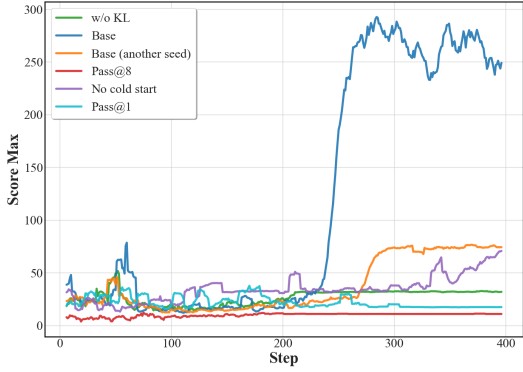

Figure 27: *Catapult* task machine scores across RL steps. KL regularization helps the model discover better structure designs. Pass@64 is greatly more efficient at uncovering powerful machine designs. Pass@8 (roll-out 8) outperforms Pass@1 (roll-out 64) in efficiency and matches its performance with fewer roll-outs. No cold start models lack the advanced knowledge needed to find better machines.

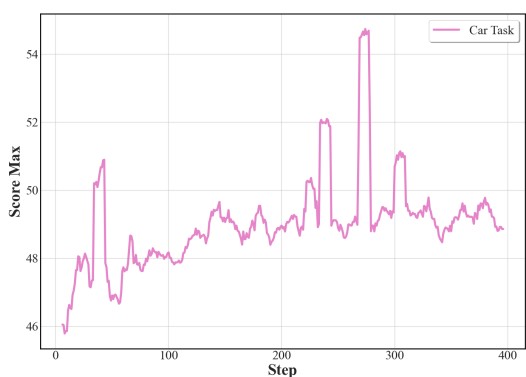

Figure 28: *Car* machine scores across RL steps. The RL finetuning hyperparameter setting is the same as the base hyperparameter setting of *Catapult*. Machine performance slightly rises as training steps increase.

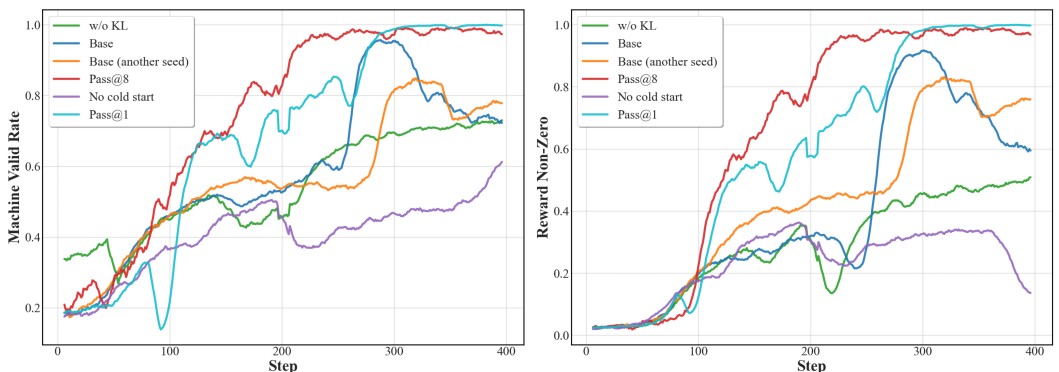

Figure 29: *Catapult task* machine validity rate and reward non-zero rate across RL steps. The machine validity rate refers to the proportion of machines that can successfully run simulations. The reward non-zero rate represents the ratio of machines that can simulate with a non-zero reward. LLM constructs more legal machines as training steps increase, and rewards non-zero machines. Pass@8 and Pass@1 converge early. "No KL" fills roll-outs with failure cases, slowing performance gains. "No cold start" lacks design knowledge, encounters more failures than no KL, and improves validity rate most slowly. The base setting balances convergence and performance improvement.

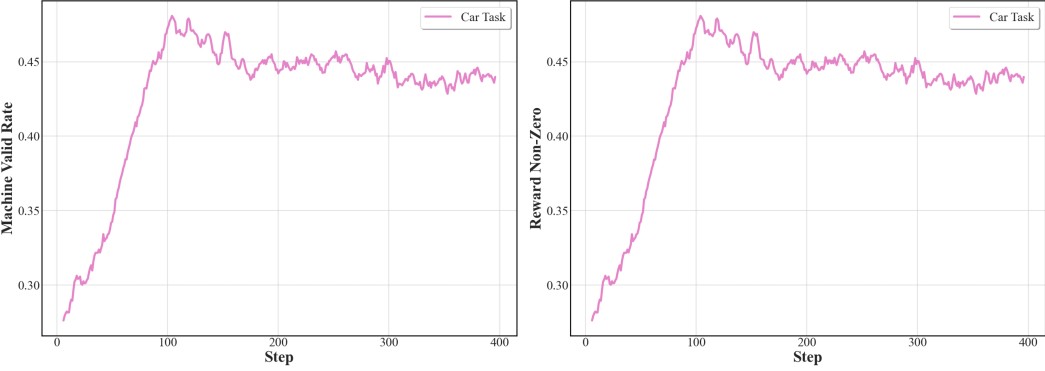

Figure 30: *Car task* machine validity rate and reward non-zero rate across RL steps. The machine validity converges early and remains stable during further training.

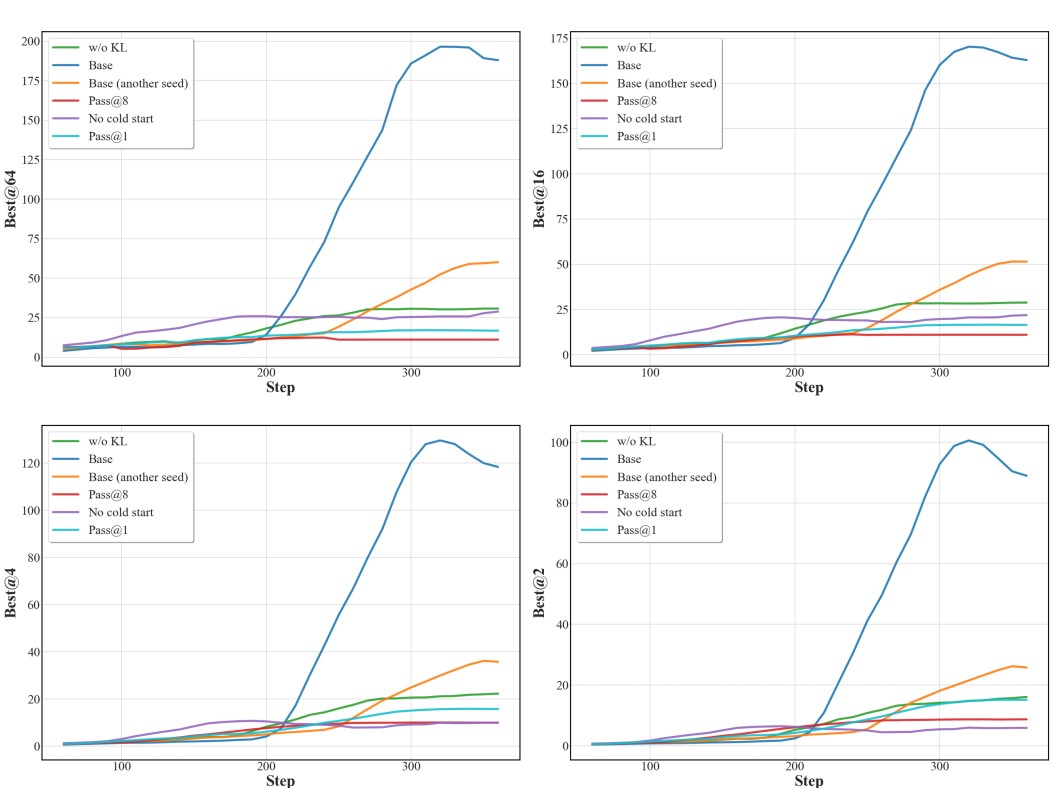

Figure 31: *Catapult task*. Average Best@N metric. At each test step, the LLM generates 64 samples, selects the top $N$ samples, and records the maximum score. This process is repeated 1,000 times, and the mean value is calculated. Base settings (both seeds) dominates Best@N performance; excluding base settings, "no KL" dominates the rest. Pass@1 and Pass@8 spawn only a handful of high-performance machines. No cold start produces machines of more average quality.

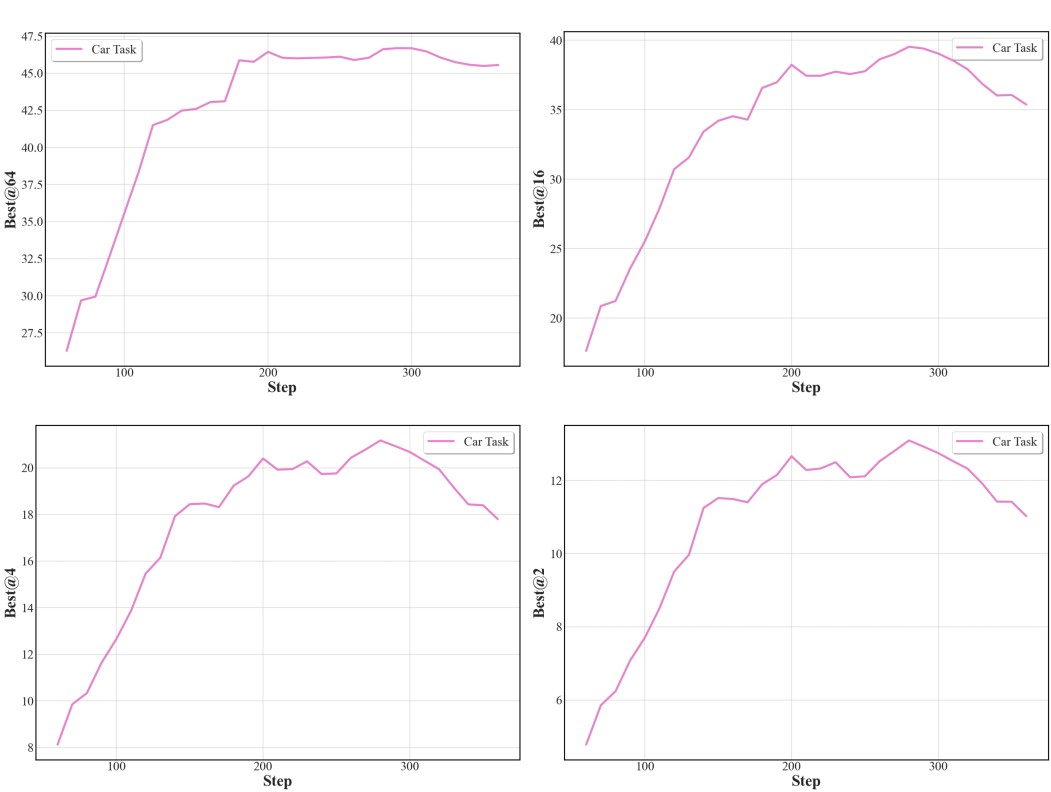

Figure 32: *Car task*. Mean Best@N metrics. Similar to the machine validity rate, the Best@N performance increases quickly and remains stable in rest training periods.

# H GENERATED SAMPLES

## H.1 FROM RL-FINETUNED MODELS

Here, we present some of the best RL samples from rollouts, as well as examples from the agentic workflow. Fig. 33 displays the RL rollout samples, while Fig. 34 illustrates the agentic workflow samples.

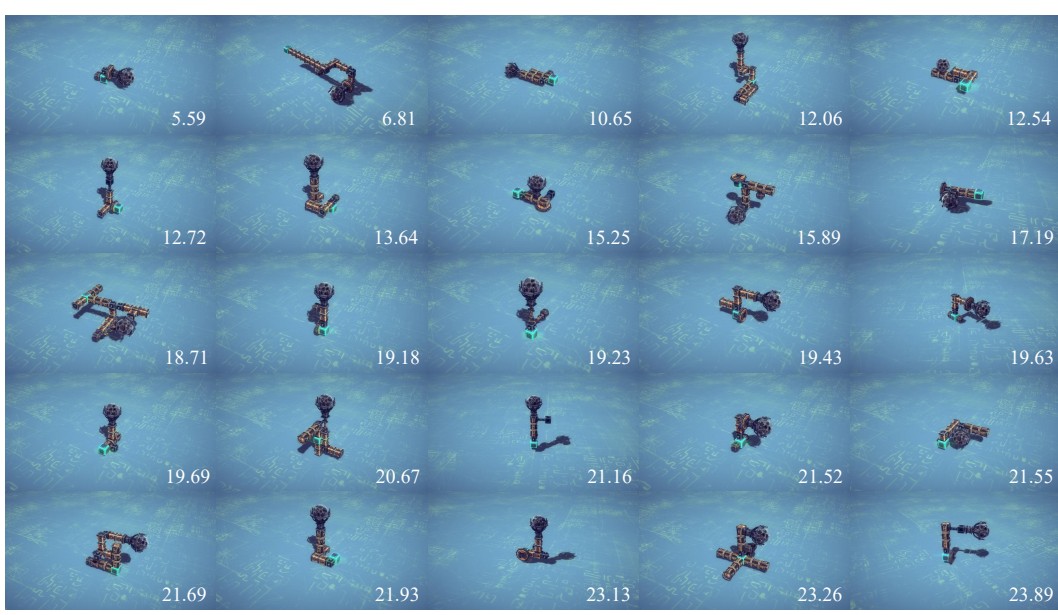

Figure 33: Qwen2.5-14B-Instruct cold started RL model *catapult* task sample from roll-out. Throwing distances are labeled on the bottom-right corner of the image.

## H.2 FROM AGENTIC WORKFLOW

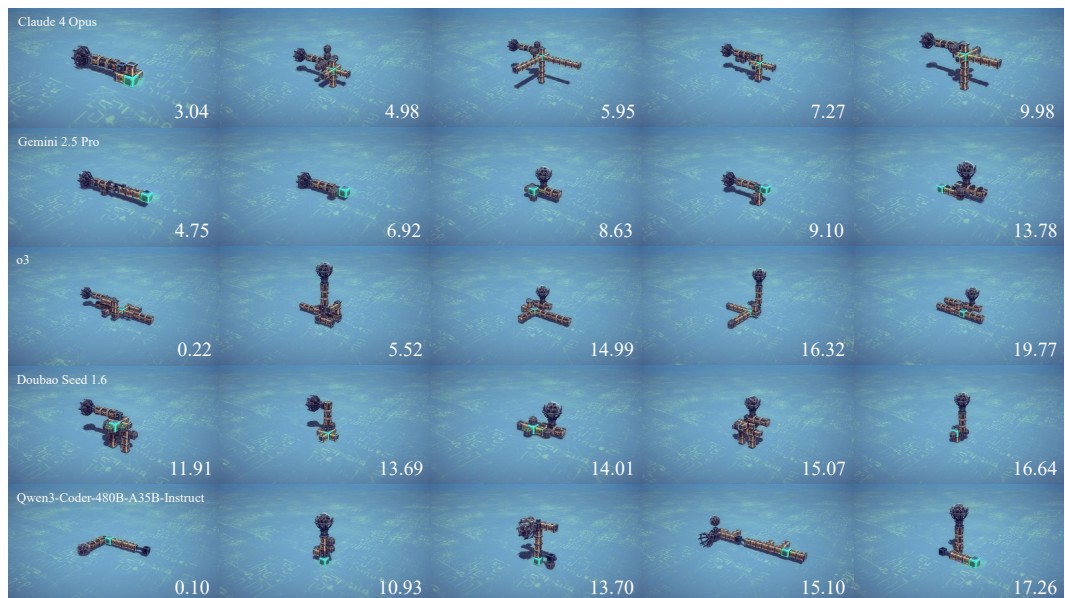

Figure 34: The LLM inference gallery of machine-generated samples. The rows, from top to bottom, were inferred by the following models, respectively: Claude 4 Opus, Gemini 2.5 Pro, o3, Doubao Seed 1.6, and Qwen3-Coder-480B-A35B-Instruct. Throwing distances are labeled on the bottom-right corner of the image.

## I SPATIAL REASONING IN LARGE LANGUAGE MODELS

In Table 1, we observe a clear gap in machine construction capabilities among LLMs. This raises the question: do current LLMs fundamentally lack strong spatial priors, or are they simply unfamiliar with Besiege? To investigate the spatial reasoning capabilites of LLMs in our task, we designed a spatial QA dataset where LLMs must identify spatially overlapping block pairs from construction trees of invalid machine designs. Each correctly identified overlap earns 1 point, with final scores calculated as the ratio of correctly identified overlaps to total overlaps. Example QA instances are shown below:

```
[System Prompt with block sizes]
Q: Give overlap id pairs of this machine in format [(...),(...)], return
an empty list if there are no overlaps.
```json
[
    {"type": 0, "id": 0, "parent": -1, "face_id": -1},
    {"type": 63, "id": 1, "parent": 0, "face_id": 0},
    {"type": 63, "id": 2, "parent": 0, "face_id": 1},
    {"type": 22, "id": 3, "parent": 1, "face_id": 4},
    {"type": 22, "id": 4, "parent": 1, "face_id": 2},
    {"type": 22, "id": 5, "parent": 2, "face_id": 5},
    {"type": 22, "id": 6, "parent": 2, "face_id": 5},
    {"type": 22, "id": 7, "parent": 1, "face_id": 2}
]
```
A:[(4,7),(5,6)]
```

The experimental results in Table. 16 shows that Gemini 2.5 Pro, the best performing model, possesses the strongest spatial reasoning capability.

| Models | Accuracy |
|---|---|
| Gemini 2.5 Pro | **0.416** |
| OpenAI o3 | 0.143 |
| Qwen3-Coder-480B-A35B | 0.208 |
| Doubao Seed 1.6 | 0.208 |
| Claude Opus 4 | 0.039 |
| DeepSeek-V3 | 0.065 |
| Kimi K2 | 0.026 |
| Llama 4 Scout 17B 16E | 0.026 |

Table 16: Quantitative results of LLMs in the spatial QA task. Gemini 2.5 Pro achieves the highest performance.

## J RELATIONS BETWEEN CoT AND MACHINES

To further investigate if high-level machine blueprint or low-level part placement is more important, we experiment with machine generation of LLMs by generating machine details conditioned on Gemini-generated CoT (instead of on CoT produced by themselves). We find that with Gemini CoT, almost all LLMs design machines that are more visually similar to "catapults", as shown in Fig. 35. We therefore hypothesize that the major gap between other LLMs, especially open-source ones, and Gemini 2.5 Pro is the abstract-level spatial and physics reasoning on machine designs.

## K MODEL SNAPSHOTS

- Claude Opus 4: claude-sonnet-4-20250514
- Gemini 2.5 pro: gemini-2.5-pro-preview-03-25
- OpenAI o3: o3-2025-04-16
- DeepSeek V3: deepseek-chat-v3.1

o3

Kimi K2

Claude Opus 4

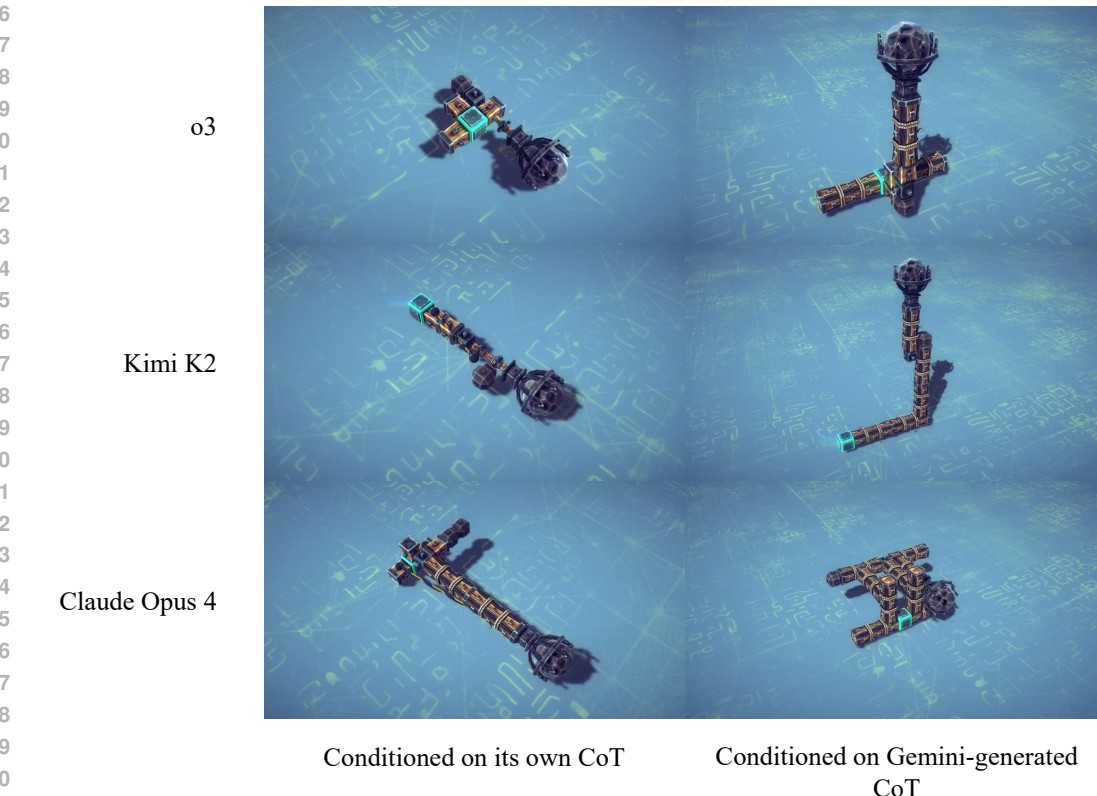

Conditioned on its own CoT      Conditioned on Gemini-generated CoT

Figure 35: Comparison between generated machines conditioned on their own CoT or Gemini-generated CoT.

- Doubao Seed 1.6: doubao-seed-1-6-250615
- Kimi-K2: kimi-k2-0711-preview
- Qwen-3: Qwen3-Coder-480B-A35B
- GPT-5: gpt-5-2025-08-07
- Llama 4: Llama 4 Scout 17B 16E

## L  CoT Samples from Gemini 2.5 Pro

### L.1  Single Agent

```
system:
<System Prompt>
user:
Design a machine to throw a boulder (type id 36) in a parabolic
trajectory. You must use exactly one boulder. Optimize for throw distance
, height, and speed. Apply the principle of leverage correctly, balancing
 the weight of the projectile.

assistant:
<think>
**Analyzing Catapult Components**
I'm now building a catapult with a rigid frame using 'Wooden Block's to
ensure stability. I plan to use a 'Log' as the throwing arm, attached to
a constructed pivot point. The 'Container' will hold the boulder at the
arm's end.

**Devising Launch Mechanisms**
I'm reconsidering my approach. With the provided constraints, a trebuchet
 is out. The single boulder and lack of hinges necessitate a catapult. I
must leverage the 'Rubber Band'. My plan is to design a tension-based
catapult for the boulder, prioritizing stability and throwing arm
efficiency.

**Optimizing for Distance**
I'm now focused on the tension-powered catapult. My plan is for a solid
base, a pivot, a lever arm, a boulder-holding container, and rubber bands
. I'm working on the arm's length and the positioning of the rubber bands
 for maximum launch power and the best trajectory for the boulder.

**Assessing the Lever Arm**
I'm now working on the arm, crucial for force. I must decide on its
length and how it attaches to the pivot. I'm considering different
attachment methods to create a stable but rotatable lever, exploring
options for maximum launch power and height.

**Evaluating Construction Strategies**
I'm stuck! I need to use a catapult given the constraints. But, the lack
of a hinge restricts my options. I'm exploring different ways to connect
the throwing arm to the frame and leverage the 'Rubber Band' for
propulsion.

**Designing a Hybrid Catapult**
I've sketched out the structure, pivoting on the 'Rotating Block', with a
 base and uprights. The throwing arm will comprise a 'Log', with a '
Container' at the tip. I am struggling with the rubber band attachment. I
 will have to ensure the right angles and tension.

**Evaluating Rubber Band Placement**
I'm now focusing on the final adjustments. With the core design finalized
, the key is the 'Rubber Band' placement. The goal is maximizing launch
power by carefully considering the band's attachment points and the angle
 of force application. I aim to achieve a long throw with this
arrangement.

**Evaluating Lever Arm Placement**
I'm now focused on the throwing arm's construction and its link to the
rotating pivot. I've built the base and the pivot structure. The lever
arm's length and the location for the 'Container' are my next priorities.
 The rubber band attachments also need to be evaluated.
```

**Engineering the Launch Mechanism**
I'm now focused on a tension-based catapult. It's time to build, starting with a solid base and a stable pivot point using the `Rotating Block`. My priorities are a strong throwing arm, the right `Container` for the boulder, and, critically, the strategic placement of the `Rubber Band`s.

**Refining the Design**
I'm now focused on the core catapult design. The need for a simple build, given the constraints, favors a straightforward tension catapult. I am planning a solid base, a lever arm (most likely a `Log`), a container for the boulder, and the crucial application of `Rubber Band`s to generate the launch force.

**Finalizing Catapult Design**
I've spent time analyzing the limitations and refining the catapult build. I've now decided on the layout. The catapult will be built around a sturdy frame, a pivot using a `Rotating Block`, an arm (lever), and the utilization of the `Rubber Band` for propulsion. I am ready for the final assembly.

**Constructing the Base**
I'm now focused on the basic frame. I've decided to construct it from logs for stability. The goal is a sturdy structure to anchor the catapult's components, primarily the `Rotating Block` and lever arm. I'm focusing on the geometry.

**Analyzing Leverage Application**
I'm now focused on catapult design. Given the one-boulder rule and lack of hinges, I'll focus on a lever-based catapult. I am now determining the lever arm length.

**Building a Lever Catapult**
I'm planning a basic tension-based catapult using the `Rotating Block` as a pivot. I'm focusing on the throwing arm's dimensions, the `Rubber Band` placement to maximize launch power, and the `Container` block to hold the boulder.

**Constructing Launch Support**
I'm now focused on the basic design. With the user's need to launch a single boulder, a tension-based catapult is optimal. I'll build it around the `Rotating Block`. I'm using a `Log` for the arm and planning a base, rubber band connections, and the boulder's container.

**Planning the Build**
I'm streamlining the design. The key will be a stable base, a `Rotating Block` for the pivot, and a lever arm. I'll use a `Container` to hold the boulder, and the `Rubber Band`s will be the power source. Now I am evaluating the dimensions and the best placement of each part.

**Constructing the Launching Mechanism**
I'm designing a simplified catapult, keeping the focus on its functionality and the user's requirement for a single boulder and the available blocks. I plan to use a `Rotating Block` as the pivot. The rest of the design must center around this `Rotating Block` component.

**Constructing the Catapult**
I've established the foundation of the catapult, focusing on a solid base and pivot mechanism. The challenge now is to construct the lever arm and connect it effectively to the `Rotating Block`, while also determining how to hold the boulder. I will prioritize a design maximizing force transfer for launch.

**Constructing a Lever Arm**
I'm solidifying the basic build: a catapult, powered by rubber bands, given the constraints. The next step is constructing the lever arm,

deciding its dimensions and how it will attach to the pivot point for optimal force transfer.

**Constructing the Final Catapult**
I've finalized the construction plan. The tension-based catapult, using the `Rotating Block` for the pivot and a lever arm, is my focus. I will use a `Container` for the boulder, and the `Rubber Band` for launching, paying close attention to force application and frame design.

**Outlining Catapult Construction**
I've sketched out the build, focusing on a tension-powered catapult. A base, a pivoting lever arm, and a boulder holder are planned. My design will depend on a stable foundation and the proper application of rubber bands for generating enough force for distance and height.

**Constructing the Catapult**
I've arrived at a straightforward plan for a tension-powered catapult, using the available blocks. I'm focusing on a design to maximize launch potential. The base, pivot point, arm, and rubber bands will be the key components for an efficient, functional build.

**Devising the Final Catapult**
I've finalized the core design. I am now sure of a tension-based catapult; the concept is simple and effective. My focus is now on the construction idea and creating the blocks. I am convinced it will be a simple build and the machine can leverage the `Rotating Block` and `Rubber Band` to meet the user's launch goals.

**Constructing a Launching System**
I'm finalizing the blueprint now. The tension-based catapult will use a base, a pivot with the `Rotating Block`, and a lever arm for the boulder. I'll ensure maximum launch power through strategic rubber band placement and precise measurements. I need a strong yet streamlined design.

**Constructing a Functional Catapult**
I've finalized the plan. The design uses a tension-based catapult, with a "Rotating Block" for a pivot. A sturdy base with two Log arms ensures stability, while two Rubber Bands generate the launch force. The boulder holder is a "Container", and the system prioritizes lever arm efficiency for maximum height and distance.

**Formulating Final Design**
I'm now certain: a tension-based catapult, incorporating the `Rotating Block` for the pivot, will meet the user's needs. The goal is maximum launch power within the given constraints. I'll sketch out the construction, outlining the key blocks and how they fit together. I've a solid plan.

Construction Idea:
I will design a tension-powered catapult to launch the boulder. The design prioritizes a long throwing distance and height by utilizing a long lever arm.

The machine has four main parts:
1.  **Base Frame:** A T-shaped base, constructed from a central 'Wooden Block' and two 'Log's extending sideways, provides a wide, stable footprint to handle the throwing forces and prevent the machine from tipping over.
2.  **Support Tower:** Two 'Log's rise vertically from the ends of the base, creating a tall, sturdy support structure for the pivot axle.
3.  **Lever Arm Assembly:** This is the core of the catapult. A 'Rotating Block' is used as a functional hinge/axle, mounted on one of the support uprights. A 'Small Wooden Block' acts as a hub on this axle, with a long 'Log' attached pointing up (the throwing arm) and a 'Small Wooden Block' attached pointing down (the short lever). This structure creates the

```
necessary leverage. At the end of the throwing arm, a 'Container' is
mounted via an adapter block to hold the 'Boulder'.
4.   **Power System:** Two 'Rubber Band' blocks provide the propulsive
force. They connect from a forward-projecting 'Log' on the base to the
short lever arm. The stretched rubber bands contract rapidly, pulling the
 short arm down and forward, which in turn whips the long throwing arm
upwards and forwards, launching the boulder in a high parabolic
trajectory.

This design maximizes the lever ratio to achieve high velocity for the
projectile, while the robust frame maintains stability during the launch.

necessary_blocks ```[1, 63, 22, 15, 30, 36, 9]```
</think>

JSON:
```json
[
{"type": 0, "id": 0, "parent": -1, "face_id": -1}
{"type": 1, "id": 1, "parent": 0, "face_id": 1}
{"type": 63, "id": 2, "parent": 1, "face_id": 1}
{"type": 63, "id": 3, "parent": 1, "face_id": 3}
{"type": 63, "id": 4, "parent": 2, "face_id": 6}
{"type": 63, "id": 5, "parent": 3, "face_id": 6}
{"type": 22, "id": 6, "parent": 4, "face_id": 4}
{"type": 15, "id": 7, "parent": 6, "face_id": 3}
{"type": 63, "id": 8, "parent": 7, "face_id": 3}
{"type": 15, "id": 9, "parent": 7, "face_id": 4}
{"type": 15, "id": 10, "parent": 8, "face_id": 0}
{"type": 30, "id": 11, "parent": 10, "face_id": 3}
{"type": 36, "id": 12, "parent": 11, "face_id": 0}
{"type": 63, "id": 13, "parent": 0, "face_id": 0}
{"type": 9, "id": 14, "parent_a": 13, "face_id_a": 1, "parent_b": 9, "
face_id_b": 1}
{"type": 9, "id": 15, "parent_a": 13, "face_id_a": 3, "parent_b": 9, "
face_id_b": 2}
]
```
```

## L.2 META DESIGNER

```
system:
<System Prompt>

user:
Design a machine to throw a boulder (type id 36) in a parabolic
trajectory. You must use exactly one boulder. Optimize for throw distance
, height, and speed. Apply the principle of leverage correctly, balancing
the weight of the projectile.

meta-designer output:
```json
{
    "definition": "Construct a machine that can throw boulders in a
parabolic trajectory, designed to maximize throw distance and height by
utilizing the principle of leverage.",
    "function_points": [
        "Establish a stable foundation to withstand recoil and prevent
tipping.",
        "Implement a powered lever arm for projectile propulsion.",
        "Carry and release a boulder projectile at the apex of the arm's
swing."
    ],
```

```
    "design_structure": [
        {
            "function_name": "Base and Support Frame",
            "description": "A wide, static frame built from the Start
Block. It provides stability for the machine and includes a vertical
support to mount the rotating mechanism, as well as a stop block to halt
the throwing arm's motion.",
            "related_function_points": [
                "Establish a stable foundation to withstand recoil and
prevent tipping."
            ]
        },
        {
            "function_name": "Powered Throwing Arm",
            "description": "A composite structure consisting of a powered
 pivot, a long lever arm, and a projectile holder. The Rotating Block
provides rapid rotational force, the Log acts as the lever to amplify
speed, and the Container holds the Boulder, releasing it via inertia.",
            "related_function_points": [
                "Implement a powered lever arm for projectile propulsion
.",
                "Carry and release a boulder projectile at the apex of
the arm's swing."
            ]
        }
    ],
    "build_order": [
        "Base and Support Frame",
        "Powered Throwing Arm"
    ],
    "machine_structure": {
        "Base and Support Frame": {
            "type": [
                1,
                15
            ],
            "guidance": "Wooden Blocks (ID 1) and Small Wooden Blocks (ID
 15) are used to build a wide, heavy base that prevents tipping. They
also form a tower to elevate the pivot point and a physical stop to
arrest the arm's rotation, which is crucial for the release mechanism."
        },
        "Powered Throwing Arm": {
            "type": [
                22,
                63,
                30,
                36
            ],
            "guidance": "A Rotating Block (ID 22) provides the high-speed
 rotational power. This is attached to a Log (ID 63), which acts as a
long, robust lever arm. A Container (ID 30) is placed at the end of the
arm to hold the Boulder (ID 36) projectile."
        }
    }
}
```

## L.3 DESIGNER

```
system:
<System Prompt>

designer-layer1 input:
```

```
On the basis of the existing structure, generate [Base and Support Frame
], [A wide, static frame built from the Start Block. It provides
stability for the machine and includes a vertical support to mount the
rotating mechanism, as well as a stop block to halt the throwing arm's
motion.], as ['Establish a stable foundation to withstand recoil and
prevent tipping.'], using block type[1, 15], [Wooden Blocks (ID 1) and
Small Wooden Blocks (ID 15) are used to build a wide, heavy base that
prevents tipping. They also form a tower to elevate the pivot point and a
 physical stop to arrest the arm's rotation, which is crucial for the
release mechanism.]

designer-output-layer1:
Based on the existing structure, a Base and Support Frame was generated
to establish a stable foundation to withstand recoil and prevent tipping.
 Ultimately, the block types [1, 15] were decided upon. No blocks were
removed as they were all suitable for the design.
JSON:
```json
[
{"type": "0", "id": 0, "parent": -1, "face_id": -1}
{"type": "1", "id": 1, "parent": 0, "face_id": 1}
{"type": "1", "id": 2, "parent": 0, "face_id": 2}
{"type": "1", "id": 3, "parent": 0, "face_id": 0}
{"type": "1", "id": 4, "parent": 1, "face_id": 2}
{"type": "1", "id": 5, "parent": 2, "face_id": 4}
{"type": "1", "id": 6, "parent": 0, "face_id": 3}
{"type": "1", "id": 7, "parent": 6, "face_id": 6}
{"type": "1", "id": 8, "parent": 7, "face_id": 6}
{"type": "15", "id": 9, "parent": 8, "face_id": 0}
]
```

designer-layer2:
On the basis of the existing structure, generate [Powered Throwing Arm],
[A composite structure consisting of a powered pivot, a long lever arm,
and a projectile holder. The Rotating Block provides rapid rotational
force, the Log acts as the lever to amplify speed, and the Container
holds the Boulder, releasing it via inertia.], as ['Implement a powered
lever arm for projectile propulsion.', "Carry and release a boulder
projectile at the apex of the arm's swing."], using block type [22, 63,
30, 36], [A Rotating Block (ID 22) provides the high-speed rotational
power. This is attached to a Log (ID 63), which acts as a long, robust
lever arm. A Container (ID 30) is placed at the end of the arm to hold
the Boulder (ID 36) projectile.]
<3D Info>

designer-output-layer2:
Based on the existing structure, a Powered Throwing Arm was generated as
Implement a powered lever arm for projectile propulsion.,Carry and
release a boulder projectile at the apex of the arm's swing..
Ultimately, the block types [22, 63, 30, 36] were decided upon, while []
were removed due to [].
JSON:
```json
[
  {"type": "0", "id": 0, "parent": -1, "face_id": -1},
  {"type": "1", "id": 1, "parent": 0, "face_id": 1},
  {"type": "1", "id": 2, "parent": 0, "face_id": 2},
  {"type": "1", "id": 3, "parent": 0, "face_id": 0},
  {"type": "1", "id": 4, "parent": 1, "face_id": 2},
  {"type": "1", "id": 5, "parent": 2, "face_id": 4},
  {"type": "1", "id": 6, "parent": 0, "face_id": 3},
  {"type": "1", "id": 7, "parent": 6, "face_id": 6},
  {"type": "1", "id": 8, "parent": 7, "face_id": 6},
  {"type": "15", "id": 9, "parent": 8, "face_id": 0},
```

```
{"type": "22", "id": 10, "parent": 8, "face_id": 4},
{"type": "63", "id": 11, "parent": 10, "face_id": 3},
{"type": "30", "id": 12, "parent": 11, "face_id": 0},
{"type": "36", "id": 13, "parent": 12, "face_id": 0}
]
```

## L.4 INSPECTOR

```
system:
<System Prompt>

inspector input:
Task Introduction

Construct a machine that can throw boulders in a parabolic trajectory,
designed to maximize throw distance and height by utilizing the principle
 of leverage.

JSON Information
<JSON>

3D Information
<3D Information>

Questions
<Inspector Questions>

inspector-output:
<Three-Dimensional Perception of the World>
{
    "Coordinate System": "Left-Handed System",
    "Up": "y",
    "Right": "x",
    "Forward": "z",
    "Frontmost Block": {"3": [0.0, 0.0, 1.5]},
    "Rearmost Block": {"9": [2.0, 2.0, -3.0]},
    "Leftmost Block": {"5": [-2.0, 0.0, 1.5]},
    "Rightmost Block": {"7": [2.0, 1.5, 0.0]},
    "Topmost Block": {"13": [1.0, 7.4, -2.0]},
    "Lowest Block": {"0": [0, 0, 0]}
}
</Three-Dimensional Perception of the World>

<Question Answer>
1.  **Dynamic Blocks Analysis:**
    The only dynamic block in this machine is the `Rotating Block` [10].
    *   **Position and Orientation**: The `Rotating Block` [10] is
located at center coordinates [1.0, 2.0, -2.0] with a global orientation
of 'x-'.
    *   **Analysis**:
        a. **Impact on the machine**: The `Rotating Block` [10] serves as
 the pivot for the throwing arm. It is the core component intended to
generate the throwing motion.
        b. **Direction of force**: The block is oriented along the 'x-'
axis, meaning its rotation axis is the global x-axis. When activated, it
will generate a torque, causing its sub-blocks to rotate in the y-z plane
.
        c. **Impact on sub-blocks and machine**: The `Rotating Block`
[10] will cause its child blocks ([11], [12], [13]) to swing upwards and
forwards in a powerful arc. Simultaneously, it will exert an equal and
opposite reaction torque on its parent block [8] and the entire support
structure, which will likely cause the machine to violently tip over or
break.
```

```
2.  **Special Static Blocks Analysis:**
    The special static blocks are the `Container` [12] and the `Boulder`
[13].
    *   **Container [12]**: It is located at the end of the throwing arm
(`Log` [11]) with its center at [1.0, 6.9, -2.0]. Its orientation is 'y+'
 (upwards). This placement and orientation are rational, as it correctly
positions it to hold the `Boulder` [13] like a cup at the end of the
throwing arm.
    *   **Boulder [13]**: It is placed inside the `Container` [12] with
its center at [1.0, 7.4, -2.0]. As the projectile, its placement is
logical and necessary for the machine's function.

3.  **Balance Check (self-gravity):**
    a. **Center of Gravity**: The calculated center of gravity for the
entire machine is approximately at (0.85, 4.22, -1.46). The block closest
 to this point is the `Log` [11].
    b. **Sinking/Collapse**: Yes, parts of the machine will collapse. The
 entire throwing mechanism, including the heavy `Boulder` [13] (mass 5),
rests on a tall, slender support tower made of single `Wooden Block`s
([6], [7], [8]). This structure is extremely fragile and not braced. The
machine is severely top-heavy, and the immense weight concentrated on
this single tower will cause it to bend and collapse under gravity alone,
 even before the `Rotating Block` [10] is activated.

4.  **Comprehensive Analysis:**
    a. **Force Summary**: The primary force is gravity, which makes the
top-heavy machine highly unstable due to the high position of the `
Boulder` [13]. Upon activation, the `Rotating Block` [10] will generate a
 massive torque on the throwing arm and an equal reaction torque on the
flimsy support tower. The combination of static top-heaviness and dynamic
 reaction forces will lead to immediate structural failure.
    b. **Logically Unreasonable Blocks**:
        *   **Hierarchical Structure**: `Starting Block` [0] -> `Wooden
Block` [6] -> `Wooden Block` [7] -> `Wooden Block` [8].
        *   **Reason for Unreasonableness**: The support tower consisting
 of blocks [6], [7], and [8] is fundamentally flawed. It is a single,
unbraced column tasked with supporting the entire throwing arm and the
heavy `Boulder` [13]. This design is far too weak to handle the static
weight, let alone the powerful dynamic forces and reaction torque from
the `Rotating Block` [10]. The machine is guaranteed to collapse.
</Question Answer>

<Summary of Design Defects>
1. Problem description, involving blocks: [6, 7, 8], belonging to
structure: "Base and Support Frame"
   The support tower for the throwing mechanism, composed of blocks [6],
[7], and [8], is critically weak. It is a single, unbraced column made of
 wooden blocks, which is completely inadequate to support the massive
weight of the throwing arm and the `Boulder` [13]. The structure will
collapse under its own weight or instantly shatter upon activation of the
 `Rotating Block` [10] due to the immense reaction torque.
</Summary of Design Defects>
```

## L.5 ENVIRONMENT QUERIER

```
system:
<System Prompt>

environment-querier input:
Task Introduction
Construct a machine that can throw boulders in a parabolic trajectory,
designed to maximize throw distance and height by utilizing the principle
 of leverage.
```

```
Mechanical JSON Structure
<JSON>

3D Information
<3D Information>

Environment Feedback
machine damaged:
    machine parts
        DoubleWoodenBlock order_id:8 occurred at 0.4 sec block order_id
13
boulder throwing distance 5.63
boulder max height 7.3
boulder actual position in first 5 seconds
[1.0, 7.01, -1.5], [0.84, 7.34, -2.44], [0.64, 5.69, -6.2], [0.61, 3.28,
-9.97], [0.57, 1.68, -10.14], [0.38, 1.78, -9.45], [0.17, 1.72, -8.76],
[-0.03, 1.52, -8.13], [-0.22, 1.22, -7.61], [-0.36, 0.85, -7.23], [-0.47,
 0.48, -6.98], [-0.53, 0.16, -6.86], [-0.55, -0.07, -6.83], [-0.55,
-0.18, -6.87], [-0.53, -0.17, -6.92], [-0.52, -0.03, -6.94], [-0.52,
0.21, -6.9], [-0.55, 0.51, -6.76], [-0.6, 0.85, -6.52], [-0.69, 1.18,
-6.16], [-0.81, 1.45, -5.72], [-0.95, 1.66, -5.2], [-1.11, 1.76, -4.64],
[-1.28, 1.76, -4.07], [-1.44, 1.66, -3.53]

environment-querier-output:
<Thought Process>
The user wants me to analyze a boulder-throwing machine that is failing
its task.

**1. Analyze the Task and Machine Design:**
*   **Objective:** Build a leverage-based machine to throw a boulder for
maximum distance and height. This implies a catapult or trebuchet design.
*   **Machine Structure:**
    *   **Base:** A wide base is constructed using `Wooden Block`s (ID 1,
 2, 3, 4, 5, 6) connected to the `Starting Block` (ID 0). This seems
designed for stability.
    *   **Tower:** A vertical tower is built on the right side of the
base with `Wooden Block`s (ID 6, 7, 8). Block 8 is the apex of this
support structure.
    *   **Lever Arm:** A `Rotating Block` (ID 10) is mounted on top of
the tower (on block 8), serving as the fulcrum. A `Log` (ID 11) acts as
the throwing arm, with a `Container` (ID 12) and `Boulder` (ID 13) at the
 top. A `Ballast` (ID 14) is attached to the other side of the `Rotating
Block` as a counterweight.
    *   **Power System:** The machine uses a combination of a powered `
Rotating Block` (ID 10) and two `Rubber Band`s (ID 15, 16). The rubber
bands connect the top of the tower (block 8) to points on the base (
blocks 3 and 5).

**2. Analyze the Environmental Feedback:**
*   **`machine damaged`**: This is the most critical piece of feedback.
The machine is breaking during operation.
*   **`machine parts: DoubleWoodenBlock order_id:8`**: The failure point
is identified as block 8. This is the `Wooden Block` that supports the
entire rotating lever arm assembly.
*   **`occurred at 0.4sec`**: The failure is almost instantaneous,
indicating a massive, immediate overload on the structure.
*   **Boulder Performance:** The boulder travels a mere 5.63 units and
ends up behind its starting Z coordinate. This confirms a catastrophic
failure of the launch mechanism, not just a weak throw. The arm is likely
 collapsing or swinging backward.

**3. Synthesize and Hypothesize the Root Cause:**
*   The point of failure (block 8) is subjected to several immense forces
:
```

1.  **Static Load:** The weight of the entire lever arm assembly (
rotating block, log, container, boulder, ballast) rests on it.
2.  **Dynamic Load:** The torque from the `Rotating Block` (ID 10)
creates a powerful reaction force on its mount point (block 8).
3.  **Tension Load:** The two `Rubber Band`s (ID 15, 16) are
stretched from block 8 down to the base. When the simulation starts, they
 will contract with immense force, pulling down and forward on block 8.
*   **Hypothesis:** The combination of these forces, particularly the
sudden, immense tension from the rubber bands, is causing block 8 to
shear off from the rest of the tower (block 7). The tower itself, being a
 single column of wooden blocks, is structurally weak and cannot
withstand this combination of bending and shear stress. The rubber bands
are incorrectly placed; instead of powering the lever arm, they are
destroying the support structure. The machine is literally tearing itself
 apart.

**4. Formulate a Plan for Further Investigation:**
To confirm this hypothesis, I need to observe the behavior of the key
components involved in the structural failure around the time of the
collapse (0.4s).

*   **Block 8 (Failure Point):** I need to see its position and rotation
to confirm it's being ripped from its mounting.
*   **Block 10 (Rotating Block/Fulcrum):** Its movement will show the
forces being applied by the lever arm onto block 8.
*   **Block 11 (Lever Arm):** Its motion will confirm the direction of
the attempted throw and the overall collapse dynamics.
*   **Rubber Bands (15 & 16):** I need to check their `length` property.
A rapid decrease in length will prove they are contracting and applying
the destructive force I suspect.

Based on this analysis, I will request detailed feedback on these
specific components to verify the cause of the failure.
</Thought Process>
<Required Feedback>
[
    {
        "id": 8,
        "duration": [0.0, 0.8],
        "properties": ["position", "rotation"]
    },
    {
        "id": 10,
        "duration": [0.0, 0.8],
        "properties": ["position", "rotation"]
    },
    {
        "id": 11,
        "duration": [0.0, 0.8],
        "properties": ["position", "rotation"]
    },
    {
        "id": 15,
        "duration": [0.0, 0.8],
        "properties": ["length"]
    },
    {
        "id": 16,
        "duration": [0.0, 0.8],
        "properties": ["length"]
    }
]
</Required Feedback>

## L.6 Refiner

```
system:
<System Prompt>

refiner input:
Task Introduction
Construct a machine that can throw boulders in a parabolic trajectory,
designed to maximize throw distance and height by utilizing the principle
 of leverage.

Mechanical JSON Structure
<JSON>

3D Information
<3D Information>

Environment Feedback
machine damaged:
    machine parts
        DoubleWoodenBlock order_id:8 occurred at 0.4 sec block order_id
13
boulder throwing distance 5.63
boulder max height 7.3
boulder actual position in first 5 seconds
[1.0, 7.01, -1.5], [0.84, 7.34, -2.44], [0.64, 5.69, -6.2], [0.61, 3.28,
-9.97], [0.57, 1.68, -10.14], [0.38, 1.78, -9.45], [0.17, 1.72, -8.76],
[-0.03, 1.52, -8.13], [-0.22, 1.22, -7.61], [-0.36, 0.85, -7.23], [-0.47,
 0.48, -6.98], [-0.53, 0.16, -6.86], [-0.55, -0.07, -6.83], [-0.55,
-0.18, -6.87], [-0.53, -0.17, -6.92], [-0.52, -0.03, -6.94], [-0.52,
0.21, -6.9], [-0.55, 0.51, -6.76], [-0.6, 0.85, -6.52], [-0.69, 1.18,
-6.16], [-0.81, 1.45, -5.72], [-0.95, 1.66, -5.2], [-1.11, 1.76, -4.64],
[-1.28, 1.76, -4.07], [-1.44, 1.66, -3.53]

Querier required feedback
block order_id 8
    type_id
        1
    info duration(s)
        [0.0, 1.0]
    block broken before below time(s)
        0.2
    block position
        [[2.01, 2.49, 0.01]] Due to block broken, the position is
incomplete or not recorded
    block velocity
        [[0.15, 1.7, 4.1]] Due to block broken, the velocity is
incomplete or not recorded
block order_id 10
    block type_id
        22
    block info duration(s)
        [0.0, 1.0]
    block position
        [[ 1.5 ,  2.52, -1.49], [ 1.5 ,  3.34, -0.56], [1.48, 5.27,
1.08], [0.73, 5.66, 2.42], [0.11, 4.45, 3.64]])]
    block rotation
        [[ 0.  , -0.71,  0.01,  0.71], [-0.16, -0.68,  0.18,  0.69],
[-0.47, -0.47,  0.51,  0.54], [-0.73, -0.04,  0.08,  0.67], [-0.69,
0.06, -0.04,  0.72]]
block order_id 0
    block type_id 0
    block info duration(s)
        [0.0, 2.0]
    block rotation
```

```
        [[0., 0., 0., 1.], [ 0.07,  0.  , -0.01,  1.  ], [ 0.45,  0.06,
-0.04,  0.89], [ 0.39,  0.14, -0.09,  0.91], [0.  , 0.15, 0.03, 0.99],
[-0.09,  0.05,  0.36,  0.93], [0.2 , 0.01, 0.09, 0.98], [ 0.13, -0.31,
-0.13,  0.93], [-0.06, -0.26, -0.1 ,  0.96], [-0.08, -0.18, -0.1 ,
0.97]]
```

Initial State of the Machine
The machine is initially placed on the ground, facing the z+ direction,
with the target direction being z+.
Please remember that the task failure is due to the mechanical design
defects.
Your ultimate goal is to modify the mechanical design defects so that it
can complete the task.
The historical modification information represents your previous attempts
 to make changes, and these steps did not comply with the game rules.
Based on the information about the task failure, please make
modifications to the mechanics.
When designing the modification steps, avoid making the same mistakes as
those in the historical modification information.

refiner-output:
<Thought Process>
The user wants me to fix a trebuchet-like machine that is failing.

**1. Analysis of the Machine and Failure:**
*    **Machine Structure:** The machine consists of a base, a vertical
tower on the right side, and a rotating arm mounted on top of the tower.
The arm has a container with a boulder on one end and was intended to
have a counterweight on the other. Power is provided by a `Rotating Block
` (ID 10) at the pivot and a `Ballast` (ID 14) acting as a counterweight.
 Two `Rubber Band`s (ID 15, 16) are connected from the top of the tower (
ID 8) to the base, seemingly for stabilization.
*    **Failure Mode:** The simulation feedback consistently reports that `
DoubleWoodenBlock id:8` is damaged at 0.4 seconds. Block 8 is the top-
most block of the support tower, directly holding the entire rotating
assembly (`Rotating Block`, arm, boulder, and ballast).
*    **Root Cause Analysis:**
    *    **Overwhelming Force:** The combination of the `Rotating Block`'s
 torque and the heavy `Ballast` (ID 14, mass 3) dropping creates an
immense rotational force. The `Boulder` (ID 13, mass 5) at the end of a
long lever arm (ID 11, `Log`) amplifies this stress. The tower,
constructed from simple `Wooden Block`s, cannot withstand this force,
leading to the failure of block 8.
    *    **Instability:** The feedback shows the entire machine lifting
and tilting (`machine orientation in long time x-`, `machine frequently
changes in the following orientations`). This indicates the base is too
small and light to counteract the violent motion of the throwing arm. The
 `Rubber Band`s (ID 15, 16) are connected asymmetrically (from the tower
to the front and left parts of the base), which likely introduces a
twisting force, contributing to the instability and structural failure
rather than preventing it.

**2. Modification Strategy:**
The primary goal is to ensure the machine's structural integrity so it
can actually launch the boulder without self-destructing.

*    **Step 1: Reduce Destructive Forces.** The most critical issue is the
 excessive force breaking block 8. The simplest way to address this is to
 remove the primary source of this force, the `Ballast` (ID 14). The `
Rotating Block` (ID 10) by itself will still provide sufficient power to
launch the boulder, but with much less violent acceleration, which the
wooden structure should be able to handle.
*    **Step 2: Remove Ineffective Components.** The `Rubber Band`s (ID 15
and 16) are not stabilizing the structure effectively and are likely
contributing to the twisting failure due to their asymmetric placement.
```

```
Removing them will simplify the forces acting on the tower and improve
stability.

This plan directly targets the identified points of failure. By removing
the ballast and the rubber bands, the stress on the critical component (
block 8) is significantly reduced, and the machine's overall stability
should improve. The throwing distance might be reduced, but a successful,
 non-destructive launch is the first priority.

**3. Final Plan:**
1.  Remove 'Ballast' (ID 14).
2.  Remove 'Rubber Band' (ID 15).
3.  Remove 'Rubber Band' (ID 16).

This is a minimal but highly effective set of changes to fix the core
problem.
</Thought Process>
<Modification Description>
The machine's support tower (block 8) is breaking due to excessive torque
 from the ballast and rotating block. The rubber bands are placed
asymmetrically, causing twisting forces and instability. I will remove
the ballast (ID 14) to reduce the strain on the structure and remove the
two rubber bands (ID 15, 16) to eliminate the destabilizing twisting
forces.
</Modification Description>
<Simulation Prediction After Modification>
*   **Power Blocks Analysis:** The launch will now be powered solely by
the 'Rotating Block' (10). The removal of the 'Ballast' (14)
significantly reduces the peak torque, resulting in a smoother, less
violent rotation of the throwing arm. This will prevent the structural
failure of the tower.
*   **Balance Check:** Without the heavy ballast dropping and the
asymmetric pull from the rubber bands, the machine will be much more
stable during operation. The base should remain mostly on the ground,
with minimal tipping as the arm swings. The overall center of gravity
will be more manageable.
*   **Comprehensive Analysis:** The primary defect, the destruction of
block 8, will be resolved. The machine will successfully complete the
launch sequence without breaking. While the throwing distance and height
may be lower due to the reduced power, the machine will now be functional
 and can complete its primary task of throwing the boulder.
*   **10-second game simulation prediction:**
    *   **0.0s - 0.5s:** The 'Rotating Block' activates, and the arm
begins to swing upwards smoothly. The tower (block 8) remains intact.
    *   **0.5s - 1.0s:** The arm reaches its maximum rotation, and the
boulder is released from the container. The machine's base remains stable
 on the ground.
    *   **1.0s - 5.0s:** The boulder travels in a clear parabolic arc.
The machine settles without any damage. The final throwing distance will
be shorter than the broken attempts but will represent a successful
launch.
</Simulation Prediction After Modification>

<Modification Steps>
Remove [14]
Remove [15]
Remove [16]
</Modification Steps>
```

# M  SINGLE-AGENT PROMPT

```
You are a machine builder. Your task is to generate a complete machine as
 a JSON file based on the user's request. Add new blocks to the initial
structure; do not modify or delete it.

I. Rules:
1.  Coordinate System: Left-handed coordinate system, y+ upwards, z+
forward and x+ right.
2.  Block Placement: New blocks must attach to 'attachable_faces' of
existing blocks. Blocks cannot overlap.
3.  Size Limit: The final machine must not exceed dimensions of 17 (
Length, Z), 17 (Width, X), 9.5 (Height, Y).
4.  Functionality: Ensure functional blocks are oriented correctly.
5.  Ground Interaction: The ground automatically conforms to the machine'
s lowest block. Account for potential collisions between the machine and
the ground throughout operation.
6.  Gravity: Every block is subject to gravity; the greater a block's
mass, the stronger its downward force. Consider this in your design when
the machine is in operation.
7.  Physical rules: Classical mechanical laws such as conservation of
momentum are applied.

II. Block Data:
Notes:
You can only use blocks from this list. A block's default orientation is
Z+.
1. Attachable face:
    a. 'id': The i-th attachable_face of this block.
    b. 'pos': Coordinates relative to the building center(which is the
attachable_face of the parent block) of this block.
    c. 'orientation': Orientation relative to the building center of this
block.
2. Tags:
    a. 'Non-static': Block can generate force or movement.
    b. 'Non-stable': Connection to parent is not rigid (e.g., hinges,
boulders).
    c. 'Linear': Do not collide with other blocks, but will occupy two
attachable_faces.
3. Special Blocks:
    a. Boulder (id 36): Does not physically connect to other blocks.
    b. Spring (id 9): A special block that pulls its two connection points
 together.

Detailed Infos:
<Block Infos without explanations>

III. JSON Output Format:
1. type: block's type_id
2. id: this is i-th block
3. parent: parent block's id
4. face_id: parent block's constructible_point id
5. Standard Block: '{"type": <int>, "id": <int>, "parent": <int>, "
face_id": <int>}'
6. special block (id: 9): '{"type": 9, "id": <int>, "parent_a": <int>, "
face_id_a": <int>, "parent_b": <int>, "face_id_b": <int>}'

IV. Final Response Format:
Your response must contain only these two parts:
1. 'Chain of thoughts:'
    a. You need to think step by step, analyse each block's usage, and
where to place them. Put your cot in <cot></cot>
```

```
    b. `Construction Idea:` A brief explanation of your design, remember
to consider necessary block types, note them in ```necessary_blocks [
type_1,type_2 ...]```, no more than 300 words.
2.  `JSON:` The complete JSON code inside a ```json ... ``` block. Here
is an example:
```json
    [
        {"type":"0","id":0,"parent":-1,"face_id":-1},
        {"type": <int>, "id": <int>, "parent": <int>, "face_id": <int>},
        ...
    ]
```
```

# N MULTI-AGENT PROMPTS

## N.1 SHARED PROMPTS

### N.1.1 GAME INTRODUCTION WITH 3D KNOWLEDGE

```
1. Coordinate System: The game uses a left-handed coordinate system, with
 the Y-axis pointing upwards. In global coordinates, z+ is forward and x+
 is to the right.
2. Construction: New blocks must be connected to the "attachable faces"
of existing blocks. The default orientation of blocks is z+.
3. Block Types:
    a. Regular Blocks: Have fixed dimensions and multiple attachable
faces.
    b. special blocks (ID 7, 9): Connect two attachable faces, do not
collide with other blocks, but will occupy the connection points.
4. Size Limitations: The mechanical dimensions must not exceed Length (z)
 17, Width (x) 17  Height (y) 9.5.
```

### N.1.2 MACHINE 3D JSON FORMAT

```json
[
    {"type":"0","id":0,"parent":-1,"face_id":-1},
    {"type":"Block Type ID","id":"Block Order ID","parent":"Parent Block
ID","face_id":"Attachable Face ID in Parent Block"},
    ...
]
```
If it is a special block (Type ID is 7 or 9, other blocks are not special
 blocks), it will be:
```json
{
    "type":"Block Type ID",
    "id":"Block Order ID",
    "parent_a":"Parent A Block Order ID",
    "face_id_a":"Attachable Face ID in Parent A Block",
    "parent_b":"Parent B Block Order ID",
    "face_id_b":"Attachable Face ID in Parent B Block"
}
```

### N.1.3 BUILD GUIDANCE

```
Your task is to: Add new blocks based on the initial machine JSON and
construction requirements provided by the user, without deleting the
initial structure, and output the final complete JSON.

User Input Format:
1.  Building Objective: Describe the structure and function to be built,
and provide a list of recommended block IDs.
2.  Initial JSON: The structural data of the existing machine.

Core Building Rules:
1.  Block Usage: You can only select from the list of recommended block
IDs provided by the user. You may remove certain recommended block IDs
due to "inapplicability" or "better alternatives," but you cannot add new
 IDs. If any are removed, the reason must be stated.
2.  Collision Prevention: You must accurately calculate the coordinates
and orientation of new blocks based on the orientation and position of
the parent block to ensure that the new block does not overlap with the
existing structure.
```

```
3.  Coordinate System and Orientation: The initial orientation of all
blocks is Z+. The final orientation of new blocks must be transformed
based on the parent block's orientation and the relative direction of the
 building point, according to the following rules:
      Oriented z+: Front z+, Back z-, Left x-, Right x+, Up y+, Down y-
      Oriented z-: Front z-, Back z+, Left x+, Right x-, Up y+, Down y-
      Oriented x-: Front x-, Back x+, Left z-, Right z+, Up y+, Down y-
      Oriented x+: Front x+, Back x-, Left z+, Right z-, Up y+, Down y-
      Oriented y+: Front y+, Back y-, Left x-, Right x+, Up z-, Down z+
      Oriented y-: Front y-, Back y+, Left x-, Right x+, Up z+, Down z-

Your Output Format:
1.  Building Plan:
    'Generated [structure summary] to achieve [function]. Finally used
blocks [ID1, ID2,...]. Removed [ID3,...] because [reason for removal].'
2.  Final JSON:
    ```json
    [
      // The complete JSON including both the initial structure and the
new blocks
    ]
    ```
```

### N.1.4 META DESIGNER SYSTEM PROMPT

```
You are a mechanical designer, and your task is to design a machine in
the game Besiege based on the user's requirements. Please gain a general
understanding of the game based on the following information:

I. Game Introduction:
1. Besiege is a physics-based construction game developed using Unity.
Players need to build various machines to complete different tasks.
2. Besiege only contains basic mechanics and physical laws, such as mass,
 friction, and collision.
3. Blocks are used to build machines. Each block has its unique functions
, advantages, and disadvantages.

II. Block Introduction:
1. Blocks are mainly divided into five major categories: Basic Blocks,
Mobility Blocks, Mechanical Blocks, Weapon Blocks, and Armor Blocks.
- Basic Blocks are the fundamental components of many machines -
structural blocks and some basic moving parts.
- Mobility Blocks are primarily designed for movement functions - powered
 and unpowered wheels, steering blocks, and gears.
- Mechanical Blocks provide various useful auxiliary functions - joints,
suspension devices, winches, grabbers, etc.
- Weapon Blocks offer various types of violent output at different ranges
 - swords and saws for close combat, and cannons and rockets for long-
range.
- Armor Blocks can protect the machine from damage or provide useful
shapes for carrying other blocks - armor plates and wooden panels, as
well as half-pipes and brackets.

2. Here is a detailed introduction to the properties and functions of
each block:
| Name | Category | Type ID | Function |
|------|----------|---------|----------|
| Starting Block | Basic | 0 | The Starting Block is the root block of
the machine; it is placed at the starting position by default, cannot be
moved, cannot be deleted, and only one can exist at a time. |
| Small Wooden Block | Basic | 15 | A basic structural block, cube-shaped
, to which other blocks can be attached from any side, making it
particularly suitable for constructing the basic framework of machines. |
```

| Wooden Block | Basic | 1 | A basic mechanical block, twice the length of a Small Wooden Block. |
| Wooden Rod | Basic | 41 | A basic mechanical block, twice the length of a Small Wooden Block, with the same weight as a Wooden Block, but very fragile. |
| Log | Basic | 63 | A basic mechanical block, more robust, three times the length of a Small Wooden Block. |
| Brace | Basic | 7 | A non-placeable block used for reinforcement, built by "attaching" to other blocks, with no collision volume. It is often used to increase the stability of static structures and is not suitable for any dynamic structures. |
| Steering Hinge | Mobility | 28 | The Steering Hinge can rotate blocks along an axis perpendicular to the placement axis. This block can rotate child blocks to a 180-degree direction to the left or right, commonly used for vehicle steering. |
| Steering Block | Mobility | 13 | The Steering Block can rotate blocks along its placement axis, similar to the rotating part of a helicopter's rotor. |
| Powered Wheel | Mobility | 2 | Similar to a car wheel, it can drive itself but cannot turn independently. It is a mechanical device used for moving objects on the ground. |
| Unpowered Wheel | Mobility | 40 | A wheel that does not rotate without external force, otherwise similar to a Powered Wheel. |
| Powered Large Wheel | Mobility | 46 | Similar to a Powered Wheel, but with a radius and thickness twice that of a Powered Wheel. |
| Unpowered Large Wheel | Mobility | | A wheel that does not rotate without external force, otherwise similar to a Powered Large Wheel. |
| Small Wheel | Mobility | 50 | It works almost the same as a caster wheel (like a shopping cart wheel), unpowered. |
| Universal Joint | Mechanical | 19 | A block that can freely rotate around its placement axis, similar to a Steering Block but without power. |
| Hinge | Mechanical | 5 | Similar to a Steering Hinge, but without power. |
| Ball Joint | Mechanical | 44 | Can swing 360 degrees along the axis perpendicular to the placement axis, but without power. |
| Axle Connector | Mechanical | 76 | Similar to a Ball Joint. |
| Rotating Block | Mechanical | 22 | Powered, it can rotate clockwise or counterclockwise along the axis perpendicular to the placement axis. |
| Suspension | Mechanical | 16 | Shaped like a wooden block, it can buffer forces from all directions. |
| Grabber | Mechanical | 27 | It will grab and hold onto any object it comes into contact with. |
| Spring | Mechanical | 9 | A special block that attaches to two other blocks and can quickly pull them together. Its pulling force is almost entirely dependent on its length. |
| Boulder | Weapon | 36 | A stone that does not directly connect to other blocks even when built on them. It can be used as a projectile weapon and is also commonly used as a target in transportation tests. |
| Elastic Pad | Armor | 87 | Increases the elasticity of the contact surface, providing an effect of rebounding and increasing kinetic energy. |
| Container | Armor | 30 | Can hold child blocks like a bowl, mainly used to carry blocks that cannot be directly connected to the machine. The container has some anti-slip capability, and only one block (the target to be carried) can be placed inside. No other blocks can be added. |
| Roller Wheel | Locomotion | 86 | Similar to the small wheel, but shorter (0.8m). |
| Grip Pad | Armour | 49 | Block with the highest friction. |
| Ballast | Flight | 35 | A heavy cubic block used as a counterweight. |

III. Mechanical Design Requirements:
1. When designing the machine, you should adopt a "layered design" approach. Break down the user's requirements into the functions that the machine needs to achieve, and list the functional points.

```
2. For each functional point, design a structure that can meet the
function. A structure can be understood as a "group of blocks," and
several structures combined form the machine.
3. For each structure, determine the types of blocks to be used.
4. Determine the construction order of the structures to make the machine
-building process layered. List which structure is the foundation and
which is the upper-layer structure, and establish the construction
sequence chain.

IV. Output Format Requirements:
```json
{
    "definition": "Construct a machine that can fulfill the user's
requirements",
    "function_points": ["Function Point 1", "Function Point 2", "Function
 Point 3"],
    "design_structure": [
        {
            "function_name": "Structure 1 Name",
            "description": "Description of Structure 1",
            "related_function_points": ["Function Point 1", "Function
Point 2"]
        },
        {
            "function_name": "Structure 2 Name",
            "description": "Description of Structure 2",
            "related_function_points": ["Function Point 3"]
        }
    ],
    "build_order": ["Structure 2 Name", "Structure 1 Name"],
    "machine_structure": {
        "Structure 1 Name": {
            "block_id": [ID1, ID2, ID3...],
            "guidance": "Guidance here"
        },
        "Structure 2 Name": {
            "block_id": [ID4, ID5, ID6...],
            "guidance": "Guidance here"
        }
    }
}
```
V. Note:
1. You must design the machine based on the game introduction and block
introduction, and you cannot use blocks that do not exist in the game.
2. Strictly follow the output format requirements. Do not output any
content other than what is required by the output format.
3. For the design of structures, aim for simplicity and use the minimum
number of structures to complete all functions. Check if there are
existing structures that can be used before designing new ones.
4. When selecting blocks for a structure, limit the types of blocks to no
 more than three, and preferably use only one type. Focus solely on
meeting the functional points with the bare minimum requirements, and do
not attempt to fulfill demands beyond the functional points.

I will provide the user input below. Please generate a mechanical
overview in JSON format based on the user's description.
```

## N.2 DESIGNER SYSTEM AND USER PROMPT

```
<system>
You are a mechanical builder in the game "Besiege."
Your task is to add new blocks to an existing machine structure according
 to user requests and finally output the complete machine JSON data.
```

```
I. Game Introduction:
<Game Introduction With 3D Knowledge>
II. Introduction to Blocks:
<Block Infos>
III. Introduction to JSON Format:
<Machine 3D JSON Format>
IV. Construction Guidance:
<Build Guidance>
V. Output Format Requirements:
Based on the existing structure, a [structural summary] was generated as
[functional implementation].
Ultimately, the block types [ID1, ID2, ...] were decided upon, while [ID3
, ...] were removed due to [reason for removal].
JSON:
```json
...
```
VI. Note:
Building Principles
1. Correct Orientation: Ensure that functional blocks such as wheels and
hinges are oriented correctly to achieve the intended function.
2.  Efficiency First:
    a. Complete the design goal with the fewest blocks possible.
    b. The ground will automatically adapt to the lowest point of the
mechanism.

Output Requirements
1.  Strict Structure: Your response must only contain two parts:
Construction Plan and Final JSON. Prohibit any additional greetings,
explanations, or comments.
2.  Pure JSON: The complete JSON code block must be placed within ```json
 ... ```. Prohibit modifying the initial structure. Prohibit modifying
the `scale` property of any block. Prohibit adding comments or non-
existent properties in the JSON.

Next, I will provide user input. Please generate a JSON based on the
description.
<user>
<designer_output["design_structure"][i]["description"]>+<Output Machine
Json>
```

## N.3 INSPECTOR SYSTEM AND USER PROMPT

```
<system>
I'll provide you with a mission in the game Besiege,
along with the machine designed for it in JSON format and its 3D
information.
Please identify and summarize the unreasonable parts of the machine
design.
Here's the introduction to the game and construction knowledge.
I. Game Introduction:
<Game Introduction With 3D Knowledge>
II. Introduction to Blocks:
<Block Infos>
III. Introduction to JSON and 3D Information:
<Machine 3D JSON Format>
IV. Introduction to Output Format:
<Three-Dimensional Perception of the World>
{
    "Coordinate System": "Left-Handed System or Right-Handed System",
    "Up": "x or y or z",
    "Right": "x or y or z",
    "Forward": "x or y or z",
```

```
     "Frontmost Block": {"id": [Center Point Coordinates]},
     "Rearmost Block": {"id": [Center Point Coordinates]},
     "Leftmost Block": {"id": [Center Point Coordinates]},
     "Rightmost Block": {"id": [Center Point Coordinates]},
     "Topmost Block": {"id": [Center Point Coordinates]},
     "Lowest Block": {"id": [Center Point Coordinates]},
}
</Three-Dimensional Perception of the World>

<Question Answer>
Write the answer to the user's question here
</Question Answer>

<Summary of Design Defects>
1. Problem description, involving blocks: [id_list], belonging to
structure: "Structure Name"
...
</Summary of Design Defects>
V. Notes:
1. Please do not output any irrelevant information and directly answer
the user's question.
2. The id of the block must be enclosed in "[]". Additionally, do not use
 "[]" for any other numbers; consider using "()" instead.

Below, I will provide you with JSON and 3D information. Please answer the
 user's question based on this information.

<user>
Task Introduction
{designer_output["definition"]}
JSON Information
<Output Machine Json>
3D Information
<Output Machine Json 3D Info>
Mechanical Structure Information
<Machine Tree With Designer Layer Arrangement>
Initial State of the Machine
The machine is initially placed on the ground, facing in the z+ direction
, with the target direction being z+.

Questions:
1. Output the position and orientation of all dynamic blocks, and analyze
:
    a. The impact of dynamic blocks on the machine
    b. The direction of force provided by dynamic blocks
    c. The impact on sub-blocks and the direction of force on the machine

2. Output static blocks other than basic structural blocks, and analyze
the rationality of their orientation and position.

3. Balance Check (self-gravity)
    a. The center of gravity of the machine (find the block closest to
the center of gravity)
    b. Whether parts of the machine will sink due to gravity
4. Comprehensive Analysis
    a. Summarize the direction of all forces to analyze the movement of
the machine
    b. Identify logically unreasonable blocks, output their hierarchical
structure and reasons for unreasonableness
```

## N.4  REFINER SYSTEM PROMPT

```
I will give you a task in the game Besiege, as well as the 3D information
 of the machine designed to complete this task. There are some
```

```
unreasonable aspects in the design of this machine, and I would like you
to modify these parts:
I. Game Introduction:
<Game Introduction With 3D Knowledge>
II. Block Introduction:
<Block Infos>
III. Input Introduction:
1. Task & Context
Task Objective:
   <designer_output["user_input"]>
Preceding Information (if any):
   Defect Report:
      <quizzer_output>
   Modification History:
      <modify_history>
   Environmental Feedback:
      <environment_feedback>

2. Machine Data
<Machine 3D JSON Format>

IV. Modification Method Introduction:
Please follow the steps below to analyze and modify the machine structure
.
Step 1: Analyze & Plan
1.  Diagnose the Current Machine: Analyze its power, balance, structure,
and overall movement to identify design flaws or areas for optimization.
2.  Devise a Modification Plan: Based on your diagnosis, decide which
blocks to move, remove, or add.
3.  Evaluate Modification Impact: When planning, you must consider the
impact of your changes.
4.  Briefly Describe Modifications: Before generating commands, describe
your modification plan in a sentence or two using natural language.

Step 2: Execute Modification Operations
Use the following three command formats for modifications. Output only
the operation commands, with each command on a new line.

1.  Add Block (Add)
   Format: Non-Linear: Add [block type ID] to [id] in [attachable_face_id
]
             Linear: Add [block type ID] to [id_a] in [
attachable_face_id_a] to [id_b] in [attachable_face_id_b]
   Rules:
       Can only be added to original blocks. You cannot add new blocks
onto other newly added blocks.
       special blocks require specifying two connection points.

2.  Remove Block (Remove)
   Format: Remove [id]
   Rules:
       Can only remove original blocks.
       Cannot remove a block that has child blocks.

3.  Move Block (Move)
   Move [id] to [new_parent_id] in [new_attachable_face_id]
   Rules:
       Moves the target block and all its child blocks as a single unit.
       The new parent's `id` must be smaller than the `id` of the block
being moved.
       special blocks cannot be moved.
       The move must change the block's original position.

<Build Guidance: Coordinate System and Orientation>
```

```
V. Output Format Introduction:
<Thought Process>
</Thought Process>
<Modification Description>
</Modification Description>
<Simulation Prediction After Modification>
</Simulation Prediction After Modification>
<Required Feedback>
</Required Feedback>
<Modification Steps>
</Modification Steps>
VI. Note:
1. Output Scope: Only output content related to the modification method
and the required output format.
2. Ground Definition: The ground is always located beneath the machine,
in contact with the block that has the lowest y-coordinate.
3. Task Adherence: Pay close attention to the task requirements. Do not
delete important blocks by mistake.
4. Parenting Constraint: A block that has been deleted cannot be used as
a parent for new or moved blocks within the same set of operations.
5. Format Integrity: Ensure the output format is complete. You must
retain the '<Thought Process></Thought Process>' and '<Modification
Description></Modification Description>' tags.
6. Content Separation: Do not include verification results within the '<
Modification Description>' block.
7. Preserve 'Success' Steps: Retain any steps marked as "Success" and do
not adjust their order.
8. Prioritize 'Error' Steps: Focus your efforts on fixing the steps
marked as "Error". Do not modify steps marked as "Unverified" prematurely
.
9. Error History: I will provide the modification history for the "Error"
 steps. Use this information to avoid repeating invalid operations.

Below, I will provide you with the JSON and 3D information. Please modify
 the machine accordingly.
```

## N.5 ENVIRONMENT QUERIER SYSTEM PROMPT

```
I will give you a task in the game Besiege, as well as the information on
 the machine designed to complete this task.
The machine has finished the task simulation and returned some
environmental feedback to describe its performance.
Please analyze the issues with the machine based on the feedback and
request more feedback if needed.
I. Game Introduction:
<Game Introduction With 3D Knowledge>
II. Block Introduction:
<Block Infos>
III. Input Introduction:
<Machine 3D JSON Format>
IV. Query Introduction:
A. Environmental Feedback Request:
After conducting the environmental simulation of your modified machinery,
The system will provide essential data for the most critical components (
such as position, rotation, and velocity).
Based on the performance of the essential data, you need to determine
what problems the machinery may have encountered and request feedback on
the key blocks that may have issues.
You can also request those blocks that may significantly impact the
machinery's functionality and check whether their performance meets
expectations.
The format for the environmental feedback request is as follows.
Please adhere strictly to this format and avoid including any extraneous
information:
```

```
<Required Feedback>
[
    {
        "id": int,
        "duration": [float, float],
        "properties": ["position", "rotation", "velocity", "length"],
    },
    ...
]
</Required Feedback>
Both "id" and "duration" are mandatory fields.
Note that the game runs for a total of 5 seconds, game state samples per
0.2s; do not exceed this duration.
You can freely select the attributes in "properties," but avoid including
 irrelevant information.
The "length" attribute is only applicable to linear components.
V. Output Format Introduction:
<Thought Process>
</Thought Process>
<Required Feedback>
</Required Feedback>
VI. Note:
1. Please do not output any irrelevant information.
2. The ground will always correctly appear beneath the machine, making
normal contact with the block that has the lowest y-axis coordinate.
3. All blocks are affected by gravity. If a block with a large self-
weight is built on certain non-powered non-static blocks, the non-static
blocks may rotate due to the gravitational force of the sub-blocks.
4. Similarly, the power generated by powered blocks also needs to
counteract the gravitational force of the sub-blocks.
5. Please adhere to the output format and avoid adding any irrelevant
information in the JSON.

Below, I will provide you with the JSON and 3D information, as well as
the environmental feedback. Please request more feedback as needed.
```

## N.6   BLOCK INFORMATIONS

```
Explanations:

This is a concise list of the blocks you can use for this construction.
Please read and follow the rules carefully.
I. Block Information Format
Each block's information follows the dict format. Attributes that a block
 does not have will not appear in the keys.
1. Characteristic Tags:
   a. Non-static: The block can actively generate force or movement.
   b. Non-stable: The connection between the block and its parent block
is non-rigid, allowing for movement or rotation.
   c. Linear: The block is used to connect two existing points rather
than being attached to a single point.
   If there are no tags, it is a regular static and stable block.
2. Attachable Faces:
   The key is in the format of attachable face ID, coordinates (relative
coordinates), and orientation (relative orientation).
II. Key Special Rules
1. Powered Wheel Rule (applicable to all powered wheels): The direction
of power provided by the wheel is not the same as its orientation.
   - Forward (Z+ direction): The wheel should face sideways (X+ or X-).
   - Left turn (power pushes towards X-): The wheel should face forward (
Z+).
   - Right turn (power pushes towards X+): The wheel should face backward
 (Z-).
```

```
    - When the wheel faces up or down (Y+ or Y-), it does not provide
power.
2. Special Blocks (Brace, Spring):
    - They do not have their own connection points but connect to two
other blocks.
    - Brace: Used to reinforce static structures with no collision volume.
    - Spring: Generates a contracting pull force along the line connecting
 its two ends when stretched.
3. Non-connecting Blocks (bombs, boulders):
    - These blocks are placed at the specified location but do not form a
physical connection with any other block. Containers are usually needed
to hold them.

Detailed Infos:
[
    {
        "Name": "Starting Block",
        "Description": "The root block of the mechanism. It cannot be
placed or deleted, and only one can exist at a time. Its initial position
 is fixed, and its initial orientation is z+.",
        "Type ID": 0,
        "Size": [1, 1, 1],
        "Attachable Faces Properties": [
            {"ID": 0, "Coordinates": [0, 0, 0.5], "Orientation": "Front
"},
            {"ID": 1, "Coordinates": [0, 0, -0.5], "Orientation": "Back
"},
            {"ID": 2, "Coordinates": [-0.5, 0, 0], "Orientation": "Left
"},
            {"ID": 3, "Coordinates": [0.5, 0, 0], "Orientation": "Right
"},
            {"ID": 4, "Coordinates": [0, 0.5, 0], "Orientation": "Up"},
            {"ID": 5, "Coordinates": [0, -0.5, 0], "Orientation": "Down"}
        ],
        "Mass": 0.25
    },
    {
        "Name": "Small Wooden Block",
        "Description": "A basic construction block, cubic in shape.",
        "Type ID": 15,
        "Size": [1, 1, 1],
        "Attachable Faces Properties": [
            {"ID": 0, "Coordinates": [0, 0, 1], "Orientation": "Front"},
            {"ID": 1, "Coordinates": [-0.5, 0, 0.5], "Orientation": "Left
"},
            {"ID": 2, "Coordinates": [0.5, 0, 0.5], "Orientation": "Right
"},
            {"ID": 3, "Coordinates": [0, 0.5, 0.5], "Orientation": "Up"},
            {"ID": 4, "Coordinates": [0, -0.5, 0.5], "Orientation": "Down
"}
        ],
        "Mass": 0.3
    },
    {
        "Name": "Wooden Block",
        "Description": "A basic construction block.",
        "Type ID": 1,
        "Size": [1, 1, 2],
        "Attachable Faces Properties": [
            {"ID": 0, "Coordinates": [0, 0, 2], "Orientation": "Front"},
            {"ID": 1, "Coordinates": [-0.5, 0, 0.5], "Orientation": "Left
"},
            {"ID": 2, "Coordinates": [-0.5, 0, 1.5], "Orientation": "Left
"},
```

```
4050                {"ID": 3, "Coordinates": [0.5, 0, 0.5], "Orientation": "Right
4051    "},
4052                {"ID": 4, "Coordinates": [0.5, 0, 1.5], "Orientation": "Right
4053    "},
4054                {"ID": 5, "Coordinates": [0, 0.5, 0.5], "Orientation": "Up"},
4055                {"ID": 6, "Coordinates": [0, 0.5, 1.5], "Orientation": "Up"},
4056                {"ID": 7, "Coordinates": [0, -0.5, 0.5], "Orientation": "Down
        "},
4057                {"ID": 8, "Coordinates": [0, -0.5, 1.5], "Orientation": "Down
4058    "}
4059            ],
4060            "Mass": 0.5
        },
4061        {
4062            "Name": "Wooden Rod",
4063            "Description": "A basic construction block, slender and fragile
4064    .",
4065            "Type ID": 41,
4066            "Size": [1, 1, 2],
            "Attachable Faces Properties": [
4067                {"ID": 0, "Coordinates": [0, 0, 2], "Orientation": "Front"},
4068                {"ID": 1, "Coordinates": [-0.5, 0, 0.5], "Orientation": "Left
4069    "},
4070                {"ID": 2, "Coordinates": [-0.5, 0, 1.5], "Orientation": "Left
        "},
4071                {"ID": 3, "Coordinates": [0.5, 0, 0.5], "Orientation": "Right
4072    "},
4073                {"ID": 4, "Coordinates": [0.5, 0, 1.5], "Orientation": "Right
4074    "},
4075                {"ID": 5, "Coordinates": [0, 0.5, 0.5], "Orientation": "Up"},
4076                {"ID": 6, "Coordinates": [0, 0.5, 1.5], "Orientation": "Up"},
4077                {"ID": 7, "Coordinates": [0, -0.5, 0.5], "Orientation": "Down
        "},
4078                {"ID": 8, "Coordinates": [0, -0.5, 1.5], "Orientation": "Down
4079    "}
4080            ],
4081            "Mass": 0.5
        },
4082        {
4083            "Name": "Log",
4084            "Description": "A basic construction block.",
4085            "Type ID": 63,
4086            "Size": [1, 1, 3],
            "Attachable Faces Properties": [
4087                {"ID": 0, "Coordinates": [0, 0, 3], "Orientation": "Front"},
4088                {"ID": 1, "Coordinates": [-0.5, 0, 0.5], "Orientation": "Left
4089    "},
4090                {"ID": 2, "Coordinates": [-0.5, 0, 1.5], "Orientation": "Left
        "},
4091                {"ID": 3, "Coordinates": [-0.5, 0, 2.5], "Orientation": "Left
4092    "},
4093                {"ID": 4, "Coordinates": [0.5, 0, 0.5], "Orientation": "Right
4094    "},
4095                {"ID": 5, "Coordinates": [0.5, 0, 1.5], "Orientation": "Right
4096    "},
4097                {"ID": 6, "Coordinates": [0.5, 0, 2.5], "Orientation": "Right
        "},
4098                {"ID": 7, "Coordinates": [0, 0.5, 0.5], "Orientation": "Up"},
4099                {"ID": 8, "Coordinates": [0, 0.5, 1.5], "Orientation": "Up"},
4100                {"ID": 9, "Coordinates": [0, 0.5, 2.5], "Orientation": "Up"},
4101                {"ID": 10, "Coordinates": [0, -0.5, 0.5], "Orientation": "
        Down"},
4102                {"ID": 11, "Coordinates": [0, -0.5, 1.5], "Orientation": "
4103    Down"},
```

```
                {"ID": 12, "Coordinates": [0, -0.5, 2.5], "Orientation": "
    Down"}
        ],
        "Mass": 1
    },
    {
        "Name": "Steering Hinge",
        "Description": "Powered, used to control the rotation of sub-
    blocks. It can swing left and right along the axis perpendicular to the
    placement axis.",
        "Type ID": 28,
        "Size": [1, 1, 1],
        "Attachable Faces Properties": [
            {"ID": 0, "Coordinates": [0, 0, 1], "Orientation": "Front"}
        ],
        "Special Attributes": {
            "Swing Direction": ["Left", "Right"],
            "Angle": [-90, 90],
            "NonStatic":"True",
            "NonStable":"True"
        },
        "Mass": 1
    },
    {
        "Name": "Steering Block",
        "Description": "Powered, used to control the rotation of sub-
    blocks. It can rotate clockwise or counterclockwise along the placement
    axis.",
        "Type ID": 13,
        "Size": [1, 1, 1],
        "Attachable Faces Properties": [
            {"ID": 0, "Coordinates": [0, 0, 1], "Orientation": "Front"},
            {"ID": 1, "Coordinates": [-0.5, 0, 0.5], "Orientation": "Left
    "},
            {"ID": 2, "Coordinates": [0.5, 0, 0.5], "Orientation": "Right
    "},
            {"ID": 3, "Coordinates": [0, 0.5, 0.5], "Orientation": "Up"},
            {"ID": 4, "Coordinates": [0, -0.5, 0.5], "Orientation": "Down
    "}
        ],
        "Special Attributes": {
            "Rotation Axis": "Front",
            "NonStatic":"True",
            "NonStable":"True"
        },
        "Mass": 1
    },
    {
        "Name": "Powered Wheel",
        "Description": "Powered, a mechanical device used to move objects
     on the ground.",
        "Type ID": 2,
        "Size": [2, 2, 0.5],
        "Attachable Faces Properties": [
            {"ID": 0, "Coordinates": [0, 0, 0.5], "Orientation": "Front"}
        ],
        "Special Attributes": {
            "Rotation Axis": "Front",
            "PoweredWheel":"True",
            "NonStatic":"True",
            "NonStable":"True"
        },
        "Mass": 1
    },
    {
```

```
       "Name": "Unpowered Wheel",
       "Description": "A wheel that does not rotate without external
force, similar to the powered wheel.",
       "Type ID": 40,
       "Size": [2, 2, 0.5],
       "Attachable Faces Properties": [
           {"ID": 0, "Coordinates": [0, 0, 0.5], "Orientation": "Front"}
       ],
       "Special Attributes": {
           "Rotation Axis": "Front",
           "NonStable":"True"
       },
       "Mass": 1
   },
   {
       "Name": "Large Powered Wheel",
       "Description": "Similar to the powered wheel, but larger.",
       "Type ID": 46,
       "Size": [3, 3, 1],
       "Attachable Faces Properties": [
           {"ID": 0, "Coordinates": [0, 0, 1], "Orientation": "Front"},
           {"ID": 1, "Coordinates": [-1.5, 0, 1], "Orientation": "Front
"},
           {"ID": 2, "Coordinates": [1.5, 0, 1], "Orientation": "Front
"},
           {"ID": 3, "Coordinates": [0, 1.5, 1], "Orientation": "Front
"},
           {"ID": 4, "Coordinates": [0, -1.5, 1], "Orientation": "Front
"},
           {"ID": 5, "Coordinates": [-1.5, 0, 0.5], "Orientation": "Left
"},
           {"ID": 6, "Coordinates": [1.5, 0, 0.5], "Orientation": "Right
"},
           {"ID": 7, "Coordinates": [0, 1.5, 0.5], "Orientation": "Up"},
           {"ID": 8, "Coordinates": [0, -1.5, 0.5], "Orientation": "Down
"}
       ],
       "Special Attributes": {
           "Rotation Axis": "Front",
           "PoweredWheel":"True",
           "NonStatic":"True",
           "NonStable":"True"
       },
       "Mass": 1
   },
   {
       "Name": "Large Unpowered Wheel",
       "Description": "Similar to the unpowered wheel, but larger.",
       "Type ID": 60,
       "Size": [3, 3, 1],
       "Attachable Faces Properties": [
           {"ID": 0, "Coordinates": [0, 0, 1], "Orientation": "Front"},
           {"ID": 1, "Coordinates": [-1.5, 0, 1], "Orientation": "Front
"},
           {"ID": 2, "Coordinates": [1.5, 0, 1], "Orientation": "Front
"},
           {"ID": 3, "Coordinates": [0, 1.5, 1], "Orientation": "Front
"},
           {"ID": 4, "Coordinates": [0, -1.5, 1], "Orientation": "Front
"},
           {"ID": 5, "Coordinates": [-1.5, 0, 0.5], "Orientation": "Left
"},
           {"ID": 6, "Coordinates": [1.5, 0, 0.5], "Orientation": "Right
"},
           {"ID": 7, "Coordinates": [0, 1.5, 0.5], "Orientation": "Up"},
```

```
                {"ID": 8, "Coordinates": [0, -1.5, 0.5], "Orientation": "Down
"}
        ],
        "Special Attributes": {
            "Rotation Axis": "Front",
            "NonStable":"True"
        },
        "Mass": 1
    },
    {
        "Name": "Small Wheel",
        "Description": "It works almost the same as a caster wheel (e.g.,
 shopping cart wheel), but it is not powered.",
        "Type ID": 50,
        "Size": [0.5, 1, 1.5],
        "Special Attributes": {"NonStable":"True"},
        "Mass": 0.5
    },
    {
        "Name": "Roller Wheel",
        "Description": "Same as the small wheel.",
        "Type ID": 86,
        "Size": [1, 1, 1],
        "Special Attributes": {
            "NonStable":"True"
        },
        "Mass": 0.5
    },
    {
        "Name": "Universal Joint",
        "Description": "A block that can freely rotate around its
placement axis, but it is not powered.",
        "Type ID": 19,
        "Size": [1, 1, 1],
        "Attachable Faces Properties": [
            {"ID": 0, "Coordinates": [0, 0, 1], "Orientation": "Front"},
            {"ID": 1, "Coordinates": [-0.5, 0, 0.5], "Orientation": "Left
"},
            {"ID": 2, "Coordinates": [0.5, 0, 0.5], "Orientation": "Right
"},
            {"ID": 3, "Coordinates": [0, 0.5, 0.5], "Orientation": "Up"},
            {"ID": 4, "Coordinates": [0, -0.5, 0.5], "Orientation": "Down
"}
        ],
        "Special Attributes": {
            "Rotation Axis": "Front",
            "NonStable":"True"
        },
        "Mass": 0.5
    },
    {
        "Name": "Hinge",
        "Description": "It can swing up and down along the axis
perpendicular to the placement axis, but it is not powered.",
        "Type ID": 5,
        "Size": [1, 1, 1],
        "Attachable Faces Properties": [
            {"ID": 0, "Coordinates": [0, 0, 1], "Orientation": "Front"},
            {"ID": 1, "Coordinates": [-0.5, 0, 0.5], "Orientation": "Left
"},
            {"ID": 2, "Coordinates": [0.5, 0, 0.5], "Orientation": "Right
"},
            {"ID": 3, "Coordinates": [0, 0.5, 0.5], "Orientation": "Up"},
            {"ID": 4, "Coordinates": [0, -0.5, 0.5], "Orientation": "Down
"}
```

```
            ],
            "Special Attributes": {
                "Swing Direction": ["Up", "Down"],
                "Angle": [-90, 90],
                "NonStable":"True"
            },
            "Mass": 0.5
        },
        {
            "Name": "Ball Joint",
            "Description": "It can swing freely in all directions, but it is
    not powered.",
            "Type ID": 44,
            "Size": [1, 1, 1],
            "Attachable Faces Properties": [
                {"ID": 0, "Coordinates": [0, 0, 1], "Orientation": "Front"},
                {"ID": 1, "Coordinates": [-0.5, 0, 0.5], "Orientation": "Left
    "},
                {"ID": 2, "Coordinates": [0.5, 0, 0.5], "Orientation": "Right
    "},
                {"ID": 3, "Coordinates": [0, 0.5, 0.5], "Orientation": "Up"},
                {"ID": 4, "Coordinates": [0, -0.5, 0.5], "Orientation": "Down
    "}
            ],
            "Special Attributes": {
                "Swing Range": "All directions outward from the build surface
    ",
                "NonStable":"True"
            },
            "Mass": 0.5
        },
        {
            "Name": "Axle Connector",
            "Description": "Similar to a ball joint.",
            "Type ID": 76,
            "Size": [1, 1, 1],
            "Attachable Faces Properties": [
                {"ID": 0, "Coordinates": [0, 0, 1], "Orientation": "Front"}
            ],
            "Special Attributes": {
                "Swing Range": "All directions outward from the build surface
    ",
                "NonStable":"True"
            },
            "Mass": 0.3
        },
        {
            "Name": "Rotating Block",
            "Description": "When powered, this motor-like block generates
    torque and rotates about its local y-axis. Blocks connected at
    attachable_face 1 or 4 rotate with it as part of a rigid assembly. The
    rotation block has its own mass and obeys classical mechanics: it applies
     torque to connected parts when powered, and it can also be moved,
    rotated, or stopped by external forces or torques, depending on
    constraints.",
            "Type ID": 22,
            "Size": [1, 1, 1],
            "Attachable Faces Properties": [
                {"ID": 0, "Coordinates": [0, 0, 1], "Orientation": "Front"},
                {"ID": 1, "Coordinates": [-0.5, 0, 0.5], "Orientation": "Left
    "},
                {"ID": 2, "Coordinates": [0.5, 0, 0.5], "Orientation": "Right
    "},
                {"ID": 3, "Coordinates": [0, 0.5, 0.5], "Orientation": "Up"},
```

```
4320             {"ID": 4, "Coordinates": [0, -0.5, 0.5], "Orientation": "Down
4321    "}
4322         ],
4323         "Special Attributes": {
4324             "Rotation Axis": "Front",
4325             "NonStatic":"True",
4326             "NonStable":"True"
4327         },
4328         "Mass": 1
4329     },
4330     {
4331         "Name": "Grabber",
4332         "Description": "If the build point is unoccupied, it will grab
4333    any object that comes into contact with the build point and hold it
4334    firmly.",
4335         "Type ID": 27,
4336         "Size": [1, 1, 1],
4337         "Attachable Faces Properties": [
4338             {"ID": 0, "Coordinates": [0, 0, 1], "Orientation": "Front"}
4339         ],
4340         "Special Attributes": {
4341             "Grip Direction": "Front",
4342             "NonStable":"True"
4343         },
4344         "Mass": 0.5
4345     },
4346     {
4347         "Name": "Boulder",
4348         "Description": "A rock that will not directly connect to other
4349    blocks even if built on them, high mass.",
4350         "Type ID": 36,
4351         "Size": [1.9, 1.9, 1.9],
4352         "Special Attributes": {
4353             "NonStable":"True"
4354         },
4355         "Mass": 5
4356     },
4357     {
4358         "Name": "Grip Pad",
4359         "Description": "The block with the highest friction.",
4360         "Type ID": 49,
4361         "Size": [0.8, 0.8, 0.5],
4362         "Mass": 0.3
4363     },
4364     {
4365         "Name": "Elastic Pad",
4366         "Description": "The block with the highest elasticity.",
4367         "Type ID": 87,
4368         "Size": [0.8, 0.8, 0.2],
4369         "Mass": 0.3
4370     },
4371     {
4372         "Name": "Container",
4373         "Description": "It has a railing around the building point. If
      oriented towards +y, it can hold sub-blocks like a bowl. It is mainly
      used to hold blocks that cannot directly connect to the mechanism, such
      as boulders and bombs. Do not place other blocks nearby to avoid overlap
      .",
         "Type ID": 30,
         "Size": [2.4, 3, 2.8],
         "Attachable Faces Properties": [
             {"ID": 0, "Coordinates": [0, 0, 1], "Orientation": "Front"}
         ],
         "Mass": 0.5
     },
```

```
     {
         "Name": "Suspension",
         "Description": "It primarily serves as a buffer and shock
absorber. It is similar in shape to a wooden block, with all Attachable
Faces Properties located at the far end of the block.",
         "Type ID": 16,
         "Size": [1, 1, 2],
         "Attachable Faces Properties": [
             {"ID": 0, "Coordinates": [0, 0, 2], "Orientation": "Front"},
             {"ID": 1, "Coordinates": [-0.5, 0, 1.5], "Orientation": "Left
"},
             {"ID": 2, "Coordinates": [0.5, 0, 1.5], "Orientation": "Right
"},
             {"ID": 3, "Coordinates": [0, 0.5, 1.5], "Orientation": "Up"},
             {"ID": 4, "Coordinates": [0, -0.5, 1.5], "Orientation": "Down
"}
         ],
         "Mass": 0.5
     },
     {
         "Name": "Brace",
         "Description": "The brace can be used for reinforcement. Its
construction principle is to 'attach' to other blocks. It has no
collision volume. Since it is often used to stabilize static structures,
it is not suitable for any dynamic structures.",
         "Type ID": 7,
         "Special Attributes": {
             "Linear": "True",
             "Anti Tension Direction": "Towards the center of the line
segment between the two Attachable Faces Properties",
             "Anti-Compression Direction": "Outward from the center of the
 line segment between the two Attachable Faces Properties"
         },
         "Mass": 0.5
     },
     {
         "Name": "Spring",
         "Description": "A special block that attaches to two other blocks
 and can quickly pull the two ends together. Its tension force is almost
entirely dependent on its length.",
         "Type ID": 9,
         "Special Attributes": {
             "Linear": "True",
             "NonStatic":"True",
             "Tension Direction": "Towards the center of the line segment
between the two Attachable Faces Properties"
         },
         "Mass": 0.4
     },
     {
         "Name": "Ballast",
         "Description": "It serves as a counterweight, has a large mass,
and is shaped like a cube.",
         "Type ID": 35,
         "Size": [1, 1, 1],
         "Attachable Faces Properties": [
             {"ID": 0, "Coordinates": [0, 0, 1], "Orientation": "Front"},
             {"ID": 1, "Coordinates": [-0.5, 0, 0.5], "Orientation": "Left
"},
             {"ID": 2, "Coordinates": [0.5, 0, 0.5], "Orientation": "Right
"},
             {"ID": 3, "Coordinates": [0, 0.5, 0.5], "Orientation": "Up"},
             {"ID": 4, "Coordinates": [0, -0.5, 0.5], "Orientation": "Down
"}
         ],
```

```
        "Mass": 3
    }
]
```

