# OpenReview forum: "Agentic Design of Compositional Machines"
_ICLR.cc/2026/Conference — Submitted to ICLR 2026_

### Official Review · Reviewer_4v71 · 2025-10-27

**Soundness:** 4
**Presentation:** 3
**Contribution:** 2
**Rating:** 6
**Confidence:** 3

**Summary:**

The authors propose a novel benchmark task for LLM agents: compositional machine design, where the task is to assemble a machine by selecting and orienting different mechanical parts and then evaluating through physical simulation.  They will release BesiegeField, an environment based on a physics based game.  They describe an agentic LLM framework for machine design and benchmark SOTA LLMs with zero-shot CoT prompting and with RL finetuning, on two tasks: building a car and building a catapult.

**Strengths:**

1. The authors propose a very interesting and novel benchmark task that requires multiple capabilities not commonly seen in existing benchmarks.
2. They provide comprehensive benchmarking experiments over SOTA models.
3. They detail a reasonable but somewhat complex baseline LLM agent and data collection pipeline for the environment.

**Weaknesses:**

1. The authors propose this work as an "environment"/"testbed" instead of a full benchmark suite of tasks, which makes it a weaker contribution.  While the task domain is very interesting, they only propose two tasks (car and catapult), which does not provide a diverse variety of tasks for future works to evaluate on.  Furthermore it's not clear how easy this environment would be to use as a benchmark (ie. installation, environment interface, standardized observation/action spaces etc.)

2. The benchmark result scores are hard to interpret without success rates or reference score.  Adding some human or optimal scores here could be helpful to see what the gap is.  Also, it would be nice to have some more discussion and analysis of failures modes.

3. Related work is missing a discussion of existing benchmark suites for LLM agents.

**Questions:**

1. Section 4.2 was difficult to understand due to the many components and steps.

---

> ### Author Response · Authors · 2025-11-20
> **Response to Reviewer 4v71 (Part 1)**
>
> We appreciate the reviewer's acknowledge on our contribution and their time and efforts in reviewing our paper.
>
> > The authors propose this work as an "environment"/"testbed" instead of a full benchmark suite of tasks, which makes it a weaker contribution. While the task domain is very interesting, they only propose two tasks (car and catapult), which does not provide a diverse variety of tasks for future works to evaluate on.
>
> We agree with the reviewer that a greater set of tasks are important to evaluate the performance of LLMs. In Appendix B.4, we present some of the tasks that we have implemented to test LLMs. Furthermore, we have included four new tasks: **flying, jumping, lifting and gearbox design** in Appendix B.4 Fig.11 and show the qualitative results. We also show the quantitative results for the flying task in the table below. Due to time constraints, we only present the results on the top-performing LLMs. We will continue working on the rest of the models and present complete benchmark results in the final draft.
>
> | **Models**      | **Reward (Mean)** | **Reward (Max)** | **Reward (Std)** |
> |-----------------|-------------------|------------------|------------------|
> | Gemini 2.5 Pro  | 30.51             | 38.50            | 10.68            |
> | OpenAI o3       | 31.09             | 39.20            | 8.95             |
> | Kimi K2         | 1.02              | 1.52             | 0.51             |
>
>
> > Furthermore it's not clear how easy this environment would be to use as a benchmark (ie. installation, environment interface, standardized observation/action spaces etc.)
>
>
> The environment is very easy to use and we have cleaned the codebase and verified the installation procedure on different machines. Please check the anonymous link here for the code and installation guide: https://github.com/Anonymous4Researchhh/ICLR2026SubmissionNumber3510. We will also make a clean list of APIs the environment provides.
>
> Regarding observation and action spaces, the environment provides full access to all feedback (positions, orientations, velocities and force maps of all existing blocks in the environment), control commands of all powered blocks (like powered wheels). It supports arbitrary insertion and deletion of blocks and arbitrary control on physical properties. It also supports multi-agent scenarios (though we have not done any empirical experiments that use this function yet). It also supports environmental controls like external forces, winds, etc. All are implemented with simple APIs.
>
> > Related work is missing a discussion of existing benchmark suites for LLM agents.
>
> We thank the reviewer for their suggestion. We have now included a paragraph on existing LLM agent benchmark suites in the related work section in the updated draft.
>
> > Section 4.2 was difficult to understand due to the many components and steps.
>
> We apologize for not clearly presenting this section due to limited space. We have revised this section in our updated draft and will continue polishing our paper.

---

> ### Author Response · Authors · 2025-11-20
> **Response to Reviewer 4v71 (Part 2)**
>
> (Continued)
>
> > The benchmark result scores are hard to interpret without success rates or reference score. Adding some human or optimal scores here could be helpful to see what the gap is. Also, it would be nice to have some more discussion and analysis of failures modes.
>
> Good idea! Please see the table below for the comparison between human and LLM designs for car and catapults. We have also included a catapult design by an average human designer (for which we collected a dataset of player-built machines from the Internet and randomly sampled 20 out of valid machines for each task) that achieves the mean catapult reward in the table.
>
> ### **"Catapult" Task**
>
> | **Models**                  | **Mean** | **Max**   | **Std**   |
> |-----------------------------|----------|-----------|-----------|
> | Gemini 2.5 Pro              | 9.83     | 18.19     | 8.35      |
> | OpenAI o3                   | 2.00     | 11.11     | 3.98      |
> | Qwen3-Coder-480B-A35B       | 3.90     | 6.52      | 2.54      |
> | Doubao Seed 1.6             | 1.73     | 4.76      | 2.39      |
> | Claude Opus 4               | 2.27     | 9.32      | 4.22      |
> | DeepSeek-V3                 | 2.41     | 4.93      | 2.58      |
> | Kimi K2                     | 5.39     | 12.02     | 5.16      |
> | Llama 4 Scout 17B 16E       | 3.59     | 11.83     | 4.15      |
> | **Human Catapult**          | **153.90** | **977.10** | 291.83 |
>
> ---
>
> ### **"Car" Task**
>
> | **Models**                  | **Mean** | **Max**   | **Std**   |
> |-----------------------------|----------|-----------|-----------|
> | Gemini 2.5 Pro              | 29.96    | 41.52     | 7.78      |
> | OpenAI o3                   | 28.39    | 36.18     | 11.01     |
> | Qwen3-Coder-480B-A35B       | 12.59    | 34.05     | 10.78     |
> | Doubao Seed 1.6             | 18.75    | 26.02     | 4.38      |
> | Claude Opus 4               | 14.56    | 38.67     | 20.69     |
> | DeepSeek-V3                 | 17.92    | 31.94     | 12.85     |
> | Kimi K2                     | 9.47     | 14.99     | 5.48      |
> | Llama 4 Scout 17B 16E       | 5.23     | 2.00      | 0.32      |
> | **Human Car**               | **177.23** | **386.58** | 103.29 |
>
> ---
>
> On failure modes: We present failure cases in Appendix E.1 and categorize the dominant failure modes into four types: (a) Flawed high-level reasoning: the LLM generates incorrect spatial reasonings that do not advance task completion. (b) Incorrect parents: the LLM attaches or modifies blocks under the wrong parent node. (c) Incorrect part orientations, the LLM misaligns powered components, causing the mechanism to operate in the wrong direction. (d) Instruction following failures, the LLM produces syntactically correct addition or modification commands, yet the resulting tree or code fails to realize the intended design.

---

> > ### Comment · Reviewer_4v71 · 2025-11-25
> >
> > Thank you for the response as well as the additions to the paper and added baselines.  This cleared up my concerns about a human evaluation baseline and the missing related work.  I also think the newly added tasks would make this paper a stronger contribution.  However, this represents a significant addition to the originally submitted paper, tripling the number of benchmark tasks, which makes it difficult to review comprehensively in context of the entire paper.  Seeing that this concern is repeated for multiple reviewers and may "make or break" the paper, this could be a much stronger future submission with fully fleshed out benchmark tasks.  I'll maintain my score.

---

> ### Author Response · Authors · 2025-12-03
> **Following-up Response**
>
> We deeply thank the reviewer for their acknowledgement on our contributions, and more importantly, their invaluable contribution to create a healthy and constructive academic environment -- for which we feel compelled to explicitly express our gratitude due to this unexpected situation.
>
> ---
>
> Responding to your comments:
>
> First, we apologize for presenting relatively unorganized results due to limited time. We have done a more comprehensive study for agentic design in the new tasks, as in the main experiments for car and catapult tasks. The detailed description is in the updated draft (Appendix B.4, Line 1225-1237 and Line 1308; also see Table 8 & 9 in the Appendix)
>
> | Models | Flying (Height) | Flying (Height) | Flying (Height) | Jumping (Nums/10) | Jumping (Nums/10) | Jumping (Nums/10) | Delivery (Distance) | Delivery (Distance) | Delivery (Distance) | Lifting (Nums/10) | Lifting (Nums/10) | Lifting (Nums/10) | Gearing (Pass/50) |
> |--------|------------------|------------------|------------------|-------------------|-------------------|-------------------|----------------------|----------------------|----------------------|--------------------|--------------------|--------------------|--------------------|
> |        | Mean | Max | Std | Mean | Max | Std | Mean | Max | Std | Mean | Max | Std | PassRate |
> | Gemini 2.5 Pro | 30.51 | 38.50 | 10.68 | 5.50 | **10** | 3.54 | **32.57** | 45.80 | 12.07 | 9.8 | **10** | 0.4 | **45** |
> | OpenAI o3 | **31.09** | **39.20** | 8.95 | **5.88** | 10 | 4.13 | 29.33 | 42.70 | 14.24 | 5.4 | 10 | 4.45 | 38 |
> | Qwen3-Coder-480B-A35B | 3.73 | 6.96 | 3.22 | 1.5 | 2 | 0.5 | 0 | 0 | 0 | 5.5 | 10 | 4.5 | 0 |
> | Doubao Seed 1.6 | 15.61 | 31.19 | 12.56 | 1.5 | 3 | 0.76 | 11.42 | 27.30 | 8.83 | 5.67 | 10 | 4.35 | 36 |
> | Claude Opus 4 | 28.76 | 35.74 | 6.98 | 2.86 | 9 | 3.91 | 3.28 | 11.8 | 4.62 | 1.5 | 2 | 0.5 | 4 |
> | DeepSeek-V3 | 0.31 | 0.51 | 0.2 | 0 | 0 | 0 | 0 | 0 | 0 | 0.75 | 1 | 0.43 | 17 |
> | Kimi K2 | 1.02 | 1.52 | 0.51 | 3 | 3 | 0 | 0 | 0 | 0 | 0.33 | 1 | 0.47 | 5 |
> | Llama 4 Scout 17B 16E | 0.26 | 0.51 | 0.25 | 1 | 1 | 0 | 0 | 0 | 0 | 0 | 0 | 0 | 15 |
> | Human | **256.19** | **568.09** | 181.42 | **10** | **10** | 0 | **88.80** | 214.84 | 49.30 | **10** | **10** | 0 | 5 (Random Guess) |
>
>
> Regarding the amount of new materials: the introduced new materials, while looking bulky, fall in the framework we have laid down: to study how LLMs behave in exemplary design tasks. The new tasks have emphasis on some dimensions (e.g., gearing on pure static and compositional design, closer to combinatorial optimization) but never introduce new dimensions — car and catapults together cover 1) spatial reasoning on positions, poses and spatial constraints, 2) physical reasoning on future frames, 3) instruction following, 4) CoT reasoning, 5) input/output code formatting understanding, 6) ability in decomposing tasks into sub-tasks / high-level task planning, and many more.
>
> The major difference in our opinion is that they require understanding of different physical laws, like flying involves some aspects of aerodynamics. But we would like to point out that the key challenges are the ones on general LLMs capabilities shown above, instead of individual challenges for individual tasks.
>
> In summary: we believe that our additional results provide evidences to support our claim that our pipeline is generalizable, in a way that
> - does not introduce new ideas
> - can be easily evaluated in the same manner as car and catapult tasks
> - do not deviate from our investigation into the reasoning capabilities of LLMs (our major focus; instead of solving specific and individual tasks)

---

### Official Review · Reviewer_Xu8F · 2025-10-29

**Soundness:** 3
**Presentation:** 1
**Contribution:** 2
**Rating:** 4
**Confidence:** 4

**Summary:**

This paper introduces and formalizes the task of compositional machine design for Large Language Models (LLMs). To facilitate this research, the authors present BesiegeField, a new interactive testbed built on the physics-based game Besiege. This environment allows LLM agents to construct machines from parts, test them in a physical simulation, and receive reward-driven feedback.

**Strengths:**

- A Novel and Well-Scoped Benchmark: The "narrow" focus of BesiegeField can also be seen as a primary strength. The authors motivate it as a "minimalist, component-level setting"  that thoughtfully balances complexity. It is more sophisticated than simple block-stacking environments (like LEGO or Minecraft) because it incorporates realistic physics, part semantics, and machine destruction. Simultaneously, it is more tractable and abstract than full-scale CAD modeling, which has prohibitively complex rules. This calibrated environment allows for the specific "spatial and behavioral reasoning challenges"  of design to be isolated and studied effectively.
- Detailed Analysis of Agentic Workflows and Failure Modes: The paper provides more than just performance tables. It offers a qualitative and empirical analysis of why LLMs fail. A key finding is the "CoT-machine correspondence" failure , where agents generate machines that deviate from their own high-level chain-of-thought plans. The experiment where Gemini 2.5 Pro's high-quality CoT is fed to other LLMs, resulting in improved performance, is a strong piece of evidence that effectively decouples the high-level planning capability from the low-level geometric execution.

**Weaknesses:**

- Overly Specialized Task and Limited Generalization. This paper is very ambitious, but its implementation is too "narrow." Frankly, it's somewhat "misleading." Because "the performance of large models assembling tools on BesiegeFields" is not equal to, or even approximates, "the performance of large models creating tools," as the BesiegeField task is too specialized and narrow. The performance of large models on this task also depends on whether their pre-training corpus contains relevant data for this task. Therefore, it's difficult to attribute the performance of large models entirely to their "tool-making ability."
- Limited Insight and Overly Simplified Experimental Design. Aside from the example diagrams used for ease of understanding, the experimental section of the paper contains only two tables, and the information in the tables is very simple: 1. Closed-source models have their own strengths and weaknesses; 2. RLVR can improve model performance. This was an expected result, but the lack of in-depth data presentation and careful design resulted in a rather thin insight analysis in the experimental section.
- Confusing paper writing and data presentation. Admittedly, the authors provided some experimental analyses in the text, such as the experiment of feeding the high-quality CoT of the Gemini 2.5-Pro to other LLMs. Frankly, I liked this experiment, but many figures, including this one, were placed in the appendix, with no links or explanations found in the main text. Although I could access the corresponding figures in the appendix, I believe this was due to possible space constraints. However, I think that as an academic paper, authors should consider the length and formatting during the writing process, ensuring that every figure and citation in the main text corresponds to the original; this is a basic requirement.

**Questions:**

The authors state that the performance of open-source models "fall short". Was any investigation conducted to determine how much of this poor performance is due to a lack of "tool-making ability" versus a simple lack of Besiege-related data in their pre-training corpora? How can we be sure the high performance of a model like Gemini 2.5 Pro isn't simply a result of it having seen more Besiege gameplay examples?

---

> ### Author Response · Authors · 2025-11-20
> **Response to Reviewer Xu8F (Part 1)**
>
> We sincerely thank the reviewer for investing their time and efforts in reviewing our paper and providing valuable suggestions.
>
> > Overly Specialized Task and Limited Generalization. This paper is very ambitious, but its implementation is too "narrow." Frankly, it's somewhat "misleading." Because "the performance of large models assembling tools on BesiegeFields" is not equal to, or even approximates, "the performance of large models creating tools," as the BesiegeField task is too specialized and narrow. The performance of large models on this task also depends on whether their pre-training corpus contains relevant data for this task. Therefore, it's difficult to attribute the performance of large models entirely to their "tool-making ability."
>
> We thank the reviewer for raising their concern.
>
> First, we would also like to clarify a potentially misleading point: our quote on Franklin is only to make a rhetorical statement that being able to build machines for some physical-functionality purpose is an important aspect of intelligence. We do notice that there are subtle differences between tools and machines, and we apologize for any confusion due to this quote.
>
> On "assembling" does not even approximate "creating": we respectfully disagree with the reviewer on this point. If we look into real machine design (like mechanical watches and car engines), one of the core challenges is to compose different standardized components purchased from the market: where should we place a metal lever to transmit forces? What should the layout of the gears be like? Should we use pistons from Supplier A or Supplier B? Even if the whole design process is a pure assembly one, the level of difficulty is not greatly reduced. Indeed, chip layout design (esp. with LLMs/VLMs) [1], a problem that is still not solved and widely studied these days, is only about “assembly” – the types of electric parts on the chip are already decided. We thus argue that the assembly nature of our proposed task of compositional machine design is a hard question worth investigating and is the one of the core components of machine creation.
>
> [1] Liu, Mingjie, et al. "Chipnemo: Domain-adapted LLMs for chip design." arXiv preprint arXiv:2311.00176 (2023).

---

> ### Author Response · Authors · 2025-11-20
> **Response to Reviewer Xu8F (Part 2)**
>
> (Continued)
>
> > Limited Insight and Overly Simplified Experimental Design. Aside from the example diagrams used for ease of understanding, the experimental section of the paper contains only two tables, and the information in the tables is very simple: 1. Closed-source models have their own strengths and weaknesses; 2. RLVR can improve model performance. This was an expected result, but the lack of in-depth data presentation and careful design resulted in a rather thin insight analysis in the experimental section.
>
> > Confusing paper writing and data presentation. Admittedly, the authors provided some experimental analyses in the text, such as the experiment of feeding the high-quality CoT of the Gemini 2.5-Pro to other LLMs. Frankly, I liked this experiment, but many figures, including this one, were placed in the appendix, with no links or explanations found in the main text. Although I could access the corresponding figures in the appendix, I believe this was due to possible space constraints. However, I think that as an academic paper, authors should consider the length and formatting during the writing process, ensuring that every figure and citation in the main text corresponds to the original; this is a basic requirement.
>
> We apologize for not clearly explaining our experimental results: due to limited space, we had to put most of the experiments and ablation studies to the appendix, as the amount of content is indeed a lot, including introducing the task, explaining the environment, presenting both agentic workflow and RL experiments, etc. The main paper refers to the tables and figures in the appendix through clickable links, though we should have kept the appendix in the main paper instead of having it deleted but only in the supplementary materials. To address the reviewer’s concerns, we have updated the draft (Sec 4.3, Effect of CoT reasoning, line 352-354) and hopefully it is easier to navigate through the text. And we will continue polishing our draft to make it easier to read and understand.
>
> In the appendix, we present ablation results on whether including one agent is beneficial, difference in performance if different amounts of budgets are allowed for agentic workflows, importance of environment feedbacks, RL training dynamics, etc; we also include qualitative results such as failure modes. In addition, we have done new experiments to evaluate performance on new tasks and to investigate the spatial reasoning capabilities of LLMs with spatial QA datasets, which you may find in our responses to other reviewers.
>
> While many of the observations in our paper are found in other papers, we believe many of our findings/designs are unique or not widely known as a consensus: 1) poor high-level design and low-level implementation alignment due to poor spatial understanding; 2) tree representation is better than position-based representation; 3) using an agent to selectively query from the full record of environment feedbacks helps; etc.
>
> We further argue that, besides the observations, our main contributions are to introduce the task, provide a minimalist environment that captures the core but is not too hard, and provide a reference point that future works may build upon.

---

> ### Author Response · Authors · 2025-11-20
> **Response to Reviewer Xu8F (Part 3)**
>
> (Continued)
>
> > The authors state that the performance of open-source models "fall short". Was any investigation conducted to determine how much of this poor performance is due to a lack of "tool-making ability" versus a simple lack of Besiege-related data in their pre-training corpora? How can we be sure the high performance of a model like Gemini 2.5 Pro isn't simply a result of it having seen more Besiege gameplay examples?
>
> We thank the reviewer for pointing out this issue—it is indeed a tricky confounder that we need to carefully isolate so that we understand more about the LLMs’ intrinsic capabilities on reasoning. Though it is impossible to completely rule out this possibility, we modified our prompts so that the prompts do not have direct correspondence with Besiege-related descriptions and found that the performance of Gemini is still the top one.
>
> Furthermore, we conducted a direct evaluation on models’ 3D spatial reasoning capability with a spatial QA dataset. This dataset is constructed by collecting the failure designs of other LLMs and labeling where two blocks are spatially overlapping with each other; the task for LLMs is thus to identify which blocks are in conflict given the input invalid tree of blocks. Since this task only involves a structure with block positions, orientations and sizes, the task is nearly independent of any knowledge on Besiege. We found that Gemini, among all tested LLMs, achieves the best performance on this QA task. This partially explains why Gemini is great on the design task and lowers the risk that it succeeds only because it knows Besiege.
>
> | **Models**                 | **Accuracy (20 machines, 77 overlaps)** |
> |---------------------------|-------------------------------------------|
> | Gemini 2.5 Pro            | **0.416 (32/77)**                         |
> | OpenAI o3                 | 0.143 (11/77)                             |
> | Qwen3-Coder-480B-A35B     | 0.208 (16/77)                             |
> | Doubao Seed 1.6           | 0.208 (16/77)                             |
> | Claude Opus 4             | 0.039 (3/77)                              |
> | DeepSeek-V3               | 0.065 (5/77)                              |
> | Kimi K2                   | 0.026 (2/77)                              |
> | Llama 4 Scout 17B 16E     | 0.026 (2/77)                              |

---

### Official Review · Reviewer_T6fA · 2025-11-01

**Soundness:** 3
**Presentation:** 3
**Contribution:** 3
**Rating:** 6
**Confidence:** 3

**Summary:**

This paper explores whether large language models can design functional machines, not just describe them. The authors introduce BesiegeField, a new physics-based testbed built on the game Besiege, where an AI assembles machines from parts like wheels, gears, and beams to achieve goals such as driving or throwing a projectile.

The setup treats the model as an agentic designer: a planner drafts blueprints, builder agents assemble them, and a simulator provides feedback on how the machine performs. The process even uses Monte Carlo Tree Search to refine designs. Two tasks — building a car and a catapult - test both structural stability and dynamic control.

They benchmark several LLMs in this environment and also explore reinforcement learning with simulation-based rewards to improve their performance. The results show that while current models can produce simple working machines, they still struggle with precise spatial reasoning and complex coordination.

Overall, the paper introduces a creative new way to test AI’s ability to reason about physics, engineering, and design, marking a step toward language models that can actually build things.

**Strengths:**

This paper tackles a really fresh and gutsy problem getting AI to design actual working machines and does it with a custom-built setup called BesiegeField. The environment hits a nice balance: it’s realistic enough to feel meaningful, but not so complicated that it’s impossible to study. And by focusing on machines made of parts instead of single monolithic shapes, the work opens a pretty exciting new direction for generative AI research.

The evaluation is super thorough. The authors didn’t just run one model they benchmarked a whole lineup of state-of-the-art LLMs under consistent conditions, and even tried different agent designs and prompting styles. It’s not just “look what we built,” but a deep analysis of why certain strategies help or fail. That level of detail really adds credibility and makes the findings practical for others. Technically, the paper’s also impressive. Integrating a physics engine, multiple LLM agents, and RL fine-tuning is not easy but they made it work. They deal with long contexts, valid XML outputs, structured plans, and even throw in MCTS and LoRA fine-tuning. It’s clear they actually built and ran a working system, not just a concept.

Some of the insights are honestly quite interesting. For example, reasoning-heavy models didn’t outperform simpler ones — suggesting the real limitation is spatial reasoning, not chain-of-thought. They also did smart ablations, like giving a weaker model a strong model’s “blueprint,” which improved results. That shows that design knowledge transfer might be a key piece here.

Overall, the paper’s both ambitious and grounded. Seeing an AI design a small car or catapult that actually works is kinda amazing. The writing’s clear, the claims feel honest, and the contribution is genuinely new. It’s the kind of work that makes you think — okay, maybe AI-assisted engineering isn’t that far off after all.

**Weaknesses:**

The paper’s scope is a bit narrow, it only tests two machine types (cars and catapults) in one environment. Those are good proof-of-concept tasks, but it’s unclear whether the approach would generalize to new goals like lifting or jumping. Each model was trained separately per task, suggesting the method isn’t yet a general “design anything” system. A short discussion or test on multi-task generalization would’ve helped.

Performance-wise, results are modest. Many designs fail outright catapults especially, where only about a quarter of outputs even worked. Cars did better, but still far from optimal. The paper is honest about this, but some of the framing feels a bit more optimistic than the data justifies. In reality, the LLMs produce workable but far-from-expert designs, so the contribution is an early step, not a solved problem. The system is also very complex, with multiple agents, MCTS, RL loops, and structured prompts. It works, but reproducibility could be hard — it’s not obvious which component drives what improvement. Simpler baselines would make the contribution easier to isolate.

BesiegeField itself is a strong testbed, but like any simulator, it simplifies a lot. The machines use fixed control scripts, so the LLM doesn’t design control policies, just structures. There’s also no notion of cost, materials, or nois: a design that works here might not in reality. That’s fine for research, but it means the findings are more conceptual than practical for now.

Also, while the authors talk about diversity in design, their RL training tends to collapse solutions into a few high-reward patterns. The model becomes more of a reward maximizer than a creative generator. Some explicit diversity mechanisms could have helped. And while the quantitative metrics are solid, we don’t get much qualitative evaluation — it’d be nice to see whether the machines actually look well-designed or just barely functional.

**Questions:**

How well does the method generalize beyond the two tested tasks? If asked to design something new, say, a bridge or a multi-launch catapult would the current agent adapt with prompting, or does it need full retraining? It’d be useful to know whether the authors see this evolving toward a single generalist design agent or a set of specialized ones.

On design diversity, the RL phase seems to cause convergence toward a few dominant strategies. Did the authors track how diversity changed over training? Could rewarding novelty or using multiple agents help avoid mode collapse and encourage more creative designs?

Regarding scaling, are the improvements mainly due to model size (Gemini being massive), or do current LLMs fundamentally lack strong spatial priors? Would adding explicit 3D reasoning modules or geometry-aware representations help more than just bigger transformers?

Since BesiegeField fixes control policies, do the authors see future extensions where the LLM also designs controllers? Real machines couple structure and control tightly, so co-design could be a major next step.

About reproducibility: the RL setup required heavy compute (8×A100). Will code and pretrained models be released, and can smaller setups reproduce the main findings? Also, is the simulator deterministic enough for stable reward signals?

Gemini 2.5 Pro clearly dominates. Is that simply due to its scale, or might it have specialized training (e.g., multimodal or code-based) that gives it an edge in physical reasoning?

---

> ### Author Response · Authors · 2025-11-20
> **Response to Reviewer T6fA (Part 1)**
>
> We thank the reviewer for their time and efforts in reviewing our paper and their acknowledgement on our contributions.
>
> > The paper’s scope is a bit narrow, it only tests two machine types (cars and catapults) in one environment. Those are good proof-of-concept tasks, but it’s unclear whether the approach would generalize to new goals like lifting or jumping. Each model was trained separately per task, suggesting the method isn’t yet a general “design anything” system. A short discussion or test on multi-task generalization would’ve helped.
>
> Thank you for raising your concern over our task set. While it seems simple and narrow to create just car and catapults, they capture and cover the key aspects in machine design: 1) static stability, whether one machine stays stable once placed; 2) structural understanding, what it entails to have something done; 3) spatial coordination between components, if parts are aligned to achieve synergy; 4) temporal understanding, whether LLMs understand how one mechanism (like throwing something) works, how forces are applied and how the trajectory is like.
>
> We agree with the reviewer that we should include more tasks and show that the LLMs generalize to some other machine designs. For this purpose, we now include lifting, jumping and flying tasks in the updated draft (Appendix B.4, Fig.11), as suggested by the reviewer. We also include another task to design a gear box, in which the LLM must place two more gears in the middle so that the rotating gear can transmit power to the terminal gear. Our experiments show that our agentic design pipeline generalizes to other settings. In addition, we have shown some other possible tasks in Appendix B.4, Fig.12 which we aim to investigate more in the future.
>
> Regarding our RL setting: it is indeed an interesting direction to investigate if one model learns high-level design knowledge that can be applied to a diverse set of tasks. However, we observed that single-task RL finetuning is already a challenging task since open-source models themselves do not have very good spatial reasoning capabilities (shown in the results on the spatial QA below). While we strongly believe that this is the right path to pursue, without significant improvement on the base model capabilities, emergence of good multi-task design capabilities is less likely to happen.
>
>
> On spatial QA: This dataset is constructed by collecting the failure designs of other LLMs and labeling where two blocks are spatially overlapping with each other; the task for LLMs is thus to identify which blocks are in conflict given the input invalid tree of blocks: the LLMs are asked, given the provided construction tree of invalid machine designs, which two blocks are spatially overlapping with each other. We found that Gemini 2.5 Pro, the best performing model, possesses the strongest spatial reasoning capability. For QA details, please refer to Appendix I in the updated draft.
>
> | **Models**                 | **Accuracy (20 machines, 77 overlaps)** |
> |---------------------------|-------------------------------------------|
> | Gemini 2.5 Pro            | **0.416 (32/77)**                         |
> | OpenAI o3                 | 0.143 (11/77)                             |
> | Qwen3-Coder-480B-A35B     | 0.208 (16/77)                             |
> | Doubao Seed 1.6           | 0.208 (16/77)                             |
> | Claude Opus 4             | 0.039 (3/77)                              |
> | DeepSeek-V3               | 0.065 (5/77)                              |
> | Kimi K2                   | 0.026 (2/77)                              |
> | Llama 4 Scout 17B 16E     | 0.026 (2/77)                              |

---

> ### Author Response · Authors · 2025-11-20
> **Response to Reviewer T6fA (Part 2)**
>
> (Continued)
>
> > Performance-wise, results are modest. Many designs fail outright catapults especially, where only about a quarter of outputs even worked. Cars did better, but still far from optimal. The paper is honest about this, but some of the framing feels a bit more optimistic than the data justifies. In reality, the LLMs produce workable but far-from-expert designs, so the contribution is an early step, not a solved problem. The system is also very complex, with multiple agents, MCTS, RL loops, and structured prompts. It works, but reproducibility could be hard — it’s not obvious which component drives what improvement. Simpler baselines would make the contribution easier to isolate.
>
> Yes, we totally agree with the reviewer that we didn’t aim to solve the mechanical design problem. Our major purposes are that 1) we introduce the task of compositional machine design, a meaningful and simplified one that LLMs now have potential to solve, with a benchmark on the performance with simple pipelines and 2) we build an environment that one can test both agentic workflow and RLVR on compositional machine design. We would also like to emphasize that our environment provides a new dimension for the study of RLVR for spatial and physical reasoning of LLMs: LLMs receive feedback from physical simulation and improve the structural machine designs accordingly.
>
> The reasons why we include multiple agents and RL loops are to benchmark and show the potential upper bound of machine design and to demonstrate the minimalist design one can try, so that other researchers have a reference if they decide to continue on the path or to work on something similar. In the meantime, we have done ablation studies to compare these more complex workflows with simple-agent settings (e.g., Sec 4.3, Table 1 compares total agentic design. Appendix C, Table 8 compares MCTS with simple search strategies; Appendix D, Table 11,12,13 discusses the necessity of meta-designer, inspector, and environment querier) to identify which component is important or not.
>
> For reproducibility: please check the anonymous link here for code: https://github.com/Anonymous4Researchhh/ICLR2026SubmissionNumber3510. The datasets and pretrained models will also be released.
>
> > BesiegeField itself is a strong testbed, but like any simulator, it simplifies a lot. The machines use fixed control scripts, so the LLM doesn’t design control policies, just structures. There’s also no notion of cost, materials, or nois: a design that works here might not in reality. That’s fine for research, but it means the findings are more conceptual than practical for now.
>
> We fully agree with the reviewer that we do not consider many factors that are important for practical machine design, but we would like to emphasize that we aim to provide a viable path that one may study from the simplest setting and gradually approach the harder part. We view our attempt as similar to building an MNIST or CIFAR dataset to motivate people to collect suitable datasets, build better base models, develop better algorithms for both better spatial/physical understanding of LLMs and the task of machine design. This simplest setting is still hard at this point – the “benchmark” is not even saturated by the most powerful LLMs, just like the status of GSM8k dataset for LLM math reasoning two years ago. Therefore, we argue that it is a bit early to work on practical settings but exploring what we miss and how we can do better in toy environments (like our BesiegeField).

---

> ### Author Response · Authors · 2025-11-20
> **Response to Reviewer T6fA (Part 3)**
>
> (Continued)
>
> > Also, while the authors talk about diversity in design, their RL training tends to collapse solutions into a few high-reward patterns. The model becomes more of a reward maximizer than a creative generator. Some explicit diversity mechanisms could have helped.
>
> > On design diversity, the RL phase seems to cause convergence toward a few dominant strategies. Did the authors track how diversity changed over training? Could rewarding novelty or using multiple agents help avoid mode collapse and encourage more creative designs?
>
>
> Thank you for your suggestion! We did not include diversity as we observe that even the best performing machine is not quite comparable to what an average human would design, and it is probably a more critical issue at this point to figure out how LLMs may find one design that is better than human designs. But we fully agree that this dimension of diversity should nevertheless be quantified. We therefore present the evaluation of machine design diversity below for both our agentic workflow and RL experiments, we also updated our draft by adding diversity discussion in Appendix E4. Qualitative results for the catapult machines during RL training are provided in Appendix H.1, Fig. 33.
>
>
> Specifically, we consider the following metrics for machine diversity:
>
> - Shape diversity, which is measured by the distances between Chamfer Distance (CD), Hausdorff Distance (HD), and Intersection-over-Union (IoU). CD gives the bidirectional average point-to-nearest-point distance, yielding a smooth global similarity score; HD reports the largest such distance, indicating the most-diverse deviation; IoU voxelizes the 3-D space and quantifies the overlap ratio of occupied volumes. Each machine is first converted to a point cloud, evaluated with these metrics against every other machine created by the same LLM, and the per-metric averages are taken as the diversity measure.
> - Topology diversity is assessed by Graph Embedding Similarity and Tree Edit Distance (TED). All machines are stored as trees (human-authored designs are first parsed into tree form). For Graph Embedding Similarity, each tree is treated as a weighted graph and embedded with Node2Vec [1]; the cosine similarity between every pair of graphs produced by the same LLM is computed and then averaged. TED complements this by computing the minimum-cost sequence of node insertions, deletions and substitutions needed to turn one tree into another.
>
> To obtain the table below, we first run catapult task in our agentic pipeline, and sampled 20 valid machines for each LLM, we also randomly sampled 20 human-made catapult machines (for which we collected a dataset of player-built machines from the Internet), and evaluated the metrics for machine diversity.
>
> ---
> ### **Diversity for agentic experiments**
>
> | **Model Name**  | **Chamfer Distance (↑)** | **Hausdorff Distance (↑)** | **IoU (↓)** | **Graph Embedding Similarity (↓)** | **Tree Edit Distance (↑)** |
> |-----------------|---------------------------|------------------------------|-------------|-------------------------------------|------------------------------|
> | Gemini 2.5 Pro  | 0.287                     | 0.604                        | 0.247       | 0.520                               | 9.534                        |
> | OpenAI o3              | **0.311**                 | **0.612**                    | 0.235       | 0.542                               | 9.796                        |
> | Kimi-K2          | 0.230                     | 0.480                        | 0.350       | **0.392**                           | 13.512                       |
> | Human           | 0.275                     | 0.538                        | **0.206**   | 0.505                               | **15.712**                   |

---

> ### Author Response · Authors · 2025-11-20
> **Response to Reviewer T6fA (Part 4)**
>
> (Continued)
>
> ---
> ### **Diversity for RL experiments**
>
> | **RL steps** | **Chamfer Distance (↑)** | **Hausdorff Distance (↑)** | **IoU (↓)** | **Graph Embedding Similarity (↓)** | **Tree Edit Distance (↑)** |
> |--------------|---------------------------|------------------------------|-------------|-------------------------------------|------------------------------|
> | 40           | 0.370                     | 0.646                        | 0.264       | 0.660                               | 10.876                       |
> | 80           | 0.320                     | **0.661**                    | 0.292       | 0.725                               | 6.142                        |
> | 120          | 0.360                     | 0.577                        | 0.307       | 0.646                               | 4.858                        |
> | 160          | 0.196                     | 0.484                        | 0.387       | 0.722                               | 4.130                        |
> | 200          | 0.326                     | 0.586                        | 0.322       | 0.726                               | 4.923                        |
> | 240          | **0.377**                 | 0.641                        | 0.214       | 0.672                               | 5.701                        |
> | 280          | 0.110                     | 0.401                        | 0.621       | 0.760                               | 2.884                        |
> | 320          | 0.095                     | 0.338                        | 0.742       | 0.767                               | 1.630                        |
> | 360          | 0.276                     | 0.525                        | 0.390       | 0.729                               | 2.381                        |
> | 400          | 0.226                     | 0.460                        | 0.473       | 0.737                               | 2.356                        |
> | Human        | 0.275                     | 0.538                        | **0.206**   | **0.505**                           | **15.712**                   |
>
>
> We also plot the loss of entropy during RL finetuning in Appendix E, Figure 23.
>
> On preserving diversity: we have tried strategies like KL regularization and entropy-encouraging terms (standard in verl framework for RLVR) but did not find them quite useful. Using multiple agents trained with different random seeds in principle should help; yet, since achieving a good design with RLVR on Qwen is hard enough, it is not a very rewarding path to pursue at this point. It is also possible to use multiple agents during training, but due to limited time and resources (e.g., GPU memory to run multiple agents with 7B/13B models), we leave the investigation to future work.
>
> [1] Grover, Aditya, and Jure Leskovec. "node2vec: Scalable feature learning for networks." Proc. 22nd ACM SIGKDD Int. Conf. Knowledge Discovery and Data Mining. ACM, 2016.
>
>
> > And while the quantitative metrics are solid, we don’t get much qualitative evaluation — it’d be nice to see whether the machines actually look well-designed or just barely functional.
>
> Good question! Indeed both the appearance and quantitative figures can be quite misleading without proper comparison. For this reason, we have included some videos in the anonymous link:
> https://github.com/Anonymous4Researchhh/ICLR2026SubmissionNumber3510/blob/main/assets/Compositional_Machine_Task_Explain.gif
> https://github.com/Anonymous4Researchhh/ICLR2026SubmissionNumber3510/blob/main/assets/throwing.gif
>
> We also present below a table comparing the performance of LLM-designed catapults and human-design ones. It is clear that LLMs are still far from human in mechanical design. A human-designed catapult that can throw far is shown in Appendix E.4 Fig.22.

---

> ### Author Response · Authors · 2025-11-20
> **Response to Reviewer T6fA (Part 5)**
>
> (Continued)
>
> > How well does the method generalize beyond the two tested tasks? If asked to design something new, say, a bridge or a multi-launch catapult would the current agent adapt with prompting, or does it need full retraining? It’d be useful to know whether the authors see this evolving toward a single generalist design agent or a set of specialized ones.
>
> Thank you for pointing out this issue. To show that our method generalizes to some other tasks, we implemented a few more: jumping, flying, lifting and gearing, for which we show the qualitative results in Appendix B.4 Fig.11. We also present the quantitative results for the task flying below:
>
>
> | **Models**      | **Reward (Mean)** | **Reward (Max)** | **Reward (Std)** |
> |-----------------|-------------------|------------------|------------------|
> | Gemini 2.5 Pro  | 30.51             | 38.50            | 10.68            |
> | OpenAI o3       | 31.09             | 39.20            | 8.95             |
> | Kimi K2         | 1.02              | 1.52             | 0.51             |
>
> While the results show that LLMs can build machines for other tasks, machines like multi-launch catapults are still beyond the capabilities of existing LLMs without sophisticated workflow designs. It is possible that with better agentic systems LLMs can manage to accomplish these tasks, though we believe the investigation into that is beyond the scope of this paper. Moreover, the empirical results on 3D spatial reasoning (with the spatial QA dataset) imply that there must be finetuned or even better pretraining to fix these fundamentals issues.
>
> > Regarding scaling, are the improvements mainly due to model size (Gemini being massive), or do current LLMs fundamentally lack strong spatial priors? Would adding explicit 3D reasoning modules or geometry-aware representations help more than just bigger transformers?
>
> Good question! Yes, we qualitatively found that even the strongest LLMs at this point lack strong spatial priors (e.g., Appendix E2, Fig. 21). To quantify their lack of spatial priors, we created a simple spatial QA dataset: the LLMs are asked, given the provided construction tree of invalid machine designs, which two blocks are spatially overlapping with each other.  We found that pipeline works no difference without Besiege words; Gemini 2.5 Pro, the best performing model, possesses the strongest spatial reasoning capability.
>
> Regarding explicit 3D reasoning modules: in principle we believe these modules help, but for LLMs it seems that the most popular approach is to train or finetune the base model with some 3D-knowledge dataset. In our agentic workflow experiments, since we are not retraining the models (and for many of them we cannot) it is hard to verify if this is the case. For our RL experiments, our cold start example partially verifies this setting since the cold-start dataset includes the CoT reasoning paths from Gemini 2.5 Pro.
>
> | **Models**                 | **Accuracy (20 machines, 77 overlaps)** |
> |---------------------------|-------------------------------------------|
> | Gemini 2.5 Pro            | **0.416 (32/77)**                         |
> | OpenAI o3                 | 0.143 (11/77)                             |
> | Qwen3-Coder-480B-A35B     | 0.208 (16/77)                             |
> | Doubao Seed 1.6           | 0.208 (16/77)                             |
> | Claude Opus 4             | 0.039 (3/77)                              |
> | DeepSeek-V3               | 0.065 (5/77)                              |
> | Kimi K2                   | 0.026 (2/77)                              |
> | Llama 4 Scout 17B 16E     | 0.026 (2/77)                              |
>
> > Since BesiegeField fixes control policies, do the authors see future extensions where the LLM also designs controllers? Real machines couple structure and control tightly, so co-design could be a major next step.
>
> Yes, we do plan to investigate this aspect since our environment provides full access to control commands and environment feedbacks that allow for proper control policies.
>
> > About reproducibility: the RL setup required heavy compute (8×A100). Will code and pretrained models be released, and can smaller setups reproduce the main findings? Also, is the simulator deterministic enough for stable reward signals?
>
> Yes, please check the anonymous link here for code: https://github.com/Anonymous4Researchhh/ICLR2026SubmissionNumber3510. The datasets and pretrained models will be released once the final decision of the paper is out.

---

> ### Author Response · Authors · 2025-11-20
> **Response to Reviewer T6fA (Part 6)**
>
> (Continued)
>
> > Gemini 2.5 Pro clearly dominates. Is that simply due to its scale, or might it have specialized training (e.g., multimodal or code-based) that gives it an edge in physical reasoning?
>
> Good question! While we do not have any information on how Gemini is exactly trained, we believe that their multi-modal training is important as it injects spatial reasoning capabilities into the model. Indeed, the superior performance of Gemini 2.5 Pro on our spatial QA dataset implies that this is one of the key aspects that lead to good machine designs.
>
> In addition, we hypothesize that the pretraining dataset of Gemini is diverse enough, as Gemini 2.5 Pro also generates more reasonable reasoning paths: in Appendix I, Fig. 35, we show an experiment where other LLMs continue from the Gemini-generated CoT to build machines, which leads to better performance than using their own CoT reasoning.

---

### Official Review · Reviewer_i7Vh · 2025-11-01

**Soundness:** 2
**Presentation:** 3
**Contribution:** 2
**Rating:** 4
**Confidence:** 4

**Summary:**

This paper introduces the problem of compositional machine design to test LLMs' capabilities in creating machines to complete a given task. To achieve this, the authors propose BesiegeField, a new simulation testbed built on the physics-based game Besiege. The authors design two compositional machine design tasks. The paper then provides comparisons of different state-of-the-art LLMs using three different agentic workflows. Additionally, they also fine-tuned Qwen2.5-14B with RL and showed improvements compared to the base Qwen model.

**Strengths:**

- The paper studies an interesting question – whether the current state-of-the-art LLMs are able to create tools to complete a desired task. The proposed simulation testbed, BesiegeField, provides the potential to answer this question.
- Another key strength is the paper's systematic evaluation. The authors test a wide range of state-of-the-art LLMs across three distinct agentic workflows (Table 1). This is complemented by fine-tuning an LLMs with rewards from the designed tasks. These experiments provide some interesting observations discussed in Section 4.3.

**Weaknesses:**

- The coverage of the tasks is limited. There are only two target machines (car and catapult). Also, the tasks for these two target machines are very similar (driving distance or boulder throwing distance). A meaningful benchmark would need to include a much wider variety of target machines and tasks.
- The paper's metrics are limited to validity and performance scores , which contradicts the authors' own claim that "generating a diverse set of candidate solutions" is an essential requirement for this task (in line 128 - 132).
- The related works are not discussed enough. There are recent works that leverage LLMs to robot morphology, which is also an instance of creating machines (see [1,2]).

[1] Ringel, Ryan P., et al. "Text2robot: Evolutionary robot design from text descriptions." 2025 IEEE International Conference on Robotics and Automation (ICRA). IEEE, 2025..
[2] Qiu, Kevin, et al. "Robomorph: Evolving robot morphology using large language models." arXiv preprint arXiv:2407.08626 (2024).

**Questions:**

- The targeted machines and the tasks chosen seem to test a very narrow slice of the exponentially large design space of the compositional machine design problem. Why would the authors think these two tasks are representative?
- Also, the authors justify the tasks as simple enough to fit into LLMs context window (Line 197, 198).
- What are the related works for hardware or embodiment design in LLMs? How is the proposed work different from them?
- The paper identifies that solution diversity is a major distinguishing factor for this task compared to math benchmarks, stating a model "should function more like a generative model than a simple reward maximizer". Given this, why do the current evaluation metrics, which focus only on validity and performance scores, not include any measure of solution diversity?
- How are the powered blocks controlled? In line 152, the authors mentioned that the power blocks can receive control commands. Are the control commands also generated by LLMs?

---

> ### Author Response · Authors · 2025-11-20
> **Response to Reviewer i7vh (Part 1)**
>
> We appreciate the time and efforts of the reviewer on our paper.
>
> > The coverage of the tasks is limited. There are only two target machines (car and catapult). Also, the tasks for these two target machines are very similar (driving distance or boulder throwing distance). A meaningful benchmark would need to include a much wider variety of target machines and tasks.
>
> > The targeted machines and the tasks chosen seem to test a very narrow slice of the exponentially large design space of the compositional machine design problem. Why would the authors think these two tasks are representative?
>
>
> Thank you for raising your concern over our task set. While it seems simple and narrow to create just car and catapults, they capture and cover the key aspects in machine design: 1) static stability, whether one machine stays stable once placed; 2) structural understanding, what it entails to have something done; 3) spatial coordination between components, if parts are aligned to achieve synergy; 4) temporal understanding, whether LLMs understand how one mechanism (like throwing something) works, how forces are applied and how the trajectory is like.
>
> The task of building catapults indeed is significantly different from that of building cars. For cars, the structure stays static (though wheels are rotating, its location and orientation never changes). For catapults, there can be complex coordination between dynamic moving parts; with a good mechanism that increases the length of the effective lever arm without any spatial conflict, one can throw the stone very far (see the video in the anonymous GitHub https://github.com/Anonymous4Researchhh/ICLR2026SubmissionNumber3510/blob/main/assets/Compositional_Machine_Task_Explain.gif for example). This requires the LLM to understand physical laws, plan in the 3D space, predict the future (i.e., as an implicit physical simulator), etc.
>
> We agree with the reviewer that we should include more tasks and show that the LLMs generalize to some other machine designs. To better address the reviewer’s concerns, we now include lifting, jumping, flying tasks in the updated draft (Appendix B.4, Fig.11) The result for the flying task is shown in the table below.
>
> We also include another task to design a gear box, in which the LLM must place two more gears in the middle so that the rotating gear can transmit power to the terminal gear. Our experiments show that our agentic design pipeline generalizes to other settings. In addition, we have shown some other possible tasks in Appendix B.4, Fig.13-18 which we aim to investigate in the future.
>
> ---
> ### **Experiment for task "Flying"**
>
> | **Models**      | **Reward (Mean)** | **Reward (Max)** | **Reward (Std)** |
> |-----------------|-------------------|------------------|------------------|
> | Gemini 2.5 Pro  | 30.51             | 38.50            | 10.68            |
> | OpenAI o3       | 31.09             | 39.20            | 8.95             |
> | Kimi K2         | 1.02              | 1.52             | 0.51             |

---

> ### Author Response · Authors · 2025-11-20
> **Response to Reviewer i7vh (Part 2)**
>
> (Continued)
>
> > The paper's metrics are limited to validity and performance scores , which contradicts the authors' own claim that "generating a diverse set of candidate solutions" is an essential requirement for this task (in line 128 - 132).
>
> > The paper identifies that solution diversity is a major distinguishing factor for this task compared to math benchmarks, stating a model "should function more like a generative model than a simple reward maximizer". Given this, why do the current evaluation metrics, which focus only on validity and performance scores, not include any measure of solution diversity?
>
> We thank the reviewer for pointing out this issue. We did not include diversity as we observe that even the best performing machine is not quite comparable to what an average human would design, and it is probably a more critical issue at this point to figure out how LLMs may find one design that is better than human designs. But we fully agree that this dimension of diversity should nevertheless be quantified. We therefore present the evaluation of machine design diversity below for both our agentic workflow and RL experiments, we also updated our draft by adding diversity discussion in Appendix E4. Qualitative results for the catapult machines during RL training are provided in Appendix H.1, Fig. 33.
>
>
> Specifically, we consider the following metrics for machine diversity:
>
> - Shape diversity, which is measured by the distances between Chamfer Distance (CD), Hausdorff Distance (HD), and Intersection-over-Union (IoU). CD gives the bidirectional average point-to-nearest-point distance, yielding a smooth global similarity score; HD reports the largest such distance, indicating the most-diverse deviation; IoU voxelizes the 3-D space and quantifies the overlap ratio of occupied volumes. Each machine is first converted to a point cloud, evaluated with these metrics against every other machine created by the same LLM, and the per-metric averages are taken as the diversity measure.
> - Topology diversity is assessed by Graph Embedding Similarity and Tree Edit Distance (TED). All machines are stored as trees (human-authored designs are first parsed into tree form). For Graph Embedding Similarity, each tree is treated as a weighted graph and embedded with Node2Vec [1]; the cosine similarity between every pair of graphs produced by the same LLM is computed and then averaged. TED complements this by computing the minimum-cost sequence of node insertions, deletions and substitutions needed to turn one tree into another.
>
> To obtain the table below, we first run catapult task in our agentic pipeline, and sampled 20 valid machines for each LLM, we also randomly sampled 20 human-made catapult machines (for which we collected a dataset of player-built machines from the Internet), and evaluated the metrics for machine diversity.
>
> ---
> ### **Diversity for agentic experiments**
>
> | **Model Name**  | **Chamfer Distance (↑)** | **Hausdorff Distance (↑)** | **IoU (↓)** | **Graph Embedding Similarity (↓)** | **Tree Edit Distance (↑)** |
> |-----------------|---------------------------|------------------------------|-------------|-------------------------------------|------------------------------|
> | Gemini 2.5 Pro  | 0.287                     | 0.604                        | 0.247       | 0.520                               | 9.534                        |
> | OpenAI o3              | **0.311**                 | **0.612**                    | 0.235       | 0.542                               | 9.796                        |
> | Kimi-K2          | 0.230                     | 0.480                        | 0.350       | **0.392**                           | 13.512                       |
> | Human           | 0.275                     | 0.538                        | **0.206**   | 0.505                               | **15.712**                   |

---

> ### Author Response · Authors · 2025-11-20
> **Response to Reviewer i7vh (Part 3)**
>
> (Continued)
>
> ---
> ### **Diversity for RL experiments**
>
> | **RL steps** | **Chamfer Distance (↑)** | **Hausdorff Distance (↑)** | **IoU (↓)** | **Graph Embedding Similarity (↓)** | **Tree Edit Distance (↑)** |
> |--------------|---------------------------|------------------------------|-------------|-------------------------------------|------------------------------|
> | 40           | 0.370                     | 0.646                        | 0.264       | 0.660                               | 10.876                       |
> | 80           | 0.320                     | **0.661**                    | 0.292       | 0.725                               | 6.142                        |
> | 120          | 0.360                     | 0.577                        | 0.307       | 0.646                               | 4.858                        |
> | 160          | 0.196                     | 0.484                        | 0.387       | 0.722                               | 4.130                        |
> | 200          | 0.326                     | 0.586                        | 0.322       | 0.726                               | 4.923                        |
> | 240          | **0.377**                 | 0.641                        | 0.214       | 0.672                               | 5.701                        |
> | 280          | 0.110                     | 0.401                        | 0.621       | 0.760                               | 2.884                        |
> | 320          | 0.095                     | 0.338                        | 0.742       | 0.767                               | 1.630                        |
> | 360          | 0.276                     | 0.525                        | 0.390       | 0.729                               | 2.381                        |
> | 400          | 0.226                     | 0.460                        | 0.473       | 0.737                               | 2.356                        |
> | Human        | 0.275                     | 0.538                        | **0.206**   | **0.505**                           | **15.712**                   |
>
>
> We also plot the loss of entropy during RL finetuning in Appendix E, Figure 23.
>
>
> [1] Grover, Aditya, and Jure Leskovec. "node2vec: Scalable feature learning for networks." Proc. 22nd ACM SIGKDD Int. Conf. Knowledge Discovery and Data Mining. ACM, 2016.
>
> > The related works are not discussed enough. There are recent works that leverage LLMs to robot morphology, which is also an instance of creating machines (see [1,2]).
>
> > What are the related works for hardware or embodiment design in LLMs? How is the proposed work different from them?
>
> We sincerely thank the reviewer for pointing our overlook on the related work on robot design and we have added a paragraph in the related work section to discuss the commons and differences between our task and robot morphology optimization in our updated draft. In the meantime, we would like to emphasize several major differences we observed:
>
> - Robots typically operate with higher degrees of freedom in the action space with less complex topology, compared to the design of mechanical machines
> - While both compositional machine design and robot morphology design involve topological structure optimization, compositional machine design concerns the assembly of heterogeneous mechanical components with intrinsic physical semantics, such as wheels, gears, springs, and propellers.
> - This heterogeneous modularity produces a more hierarchical design space, where complex machines can be viewed as compositions of functional subsystems such as drivetrains or suspension systems.
> - Moreover, although many machines require a controller to operate, there exist important classes of machines that do not, including engines and gear trains, which distinguishes compositional machine design from robot design in which morphology and control are almost always jointly considered.
>
> > Also, the authors justify the tasks as simple enough to fit into LLMs context window (Line 197, 198).
>
> Thank you for raising this issue and we agree with the reviewer that we should have it better discussed. To answer your question: the context window for many of the recent LLMs are quite long (e.g., 100k for Gemini Pro 2.5). Since 1)  the absolute value of context window length does not mean that the performance will not deteriorate with longer context and 2) we need to incorporate past attempts and environment feedbacks for machine editing, we designed the agentic workflow so that each agent only processes about 5k tokens, a number significantly smaller than the max context window size. We also performed an ablation study in which we discarded parsed 3D Information (Appendix A, Table 5) and modification history (Appendix A, Table 6) for a more compact system prompt. The results show that with this shorter prompt the performance gets worse and partially verify that the relatively long context in our experiments is not a big issue.

---

> ### Author Response · Authors · 2025-11-20
> **Response to Reviewer i7vh (Part 4)**
>
> (Continued)
>
> > How are the powered blocks controlled? In line 152, the authors mentioned that the power blocks can receive control commands. Are the control commands also generated by LLMs?
>
> Since the coupling between control policy and machine structure design can be very complicated (indeed, getting a good control policy of a humanoid with fixed structure is hard enough), we consider the simplistic scenario where all powered blocks (that receive control commands) is turned on with the max power starting from the starting time to the end of simulation. Cars and catapults in our experiments, due to their relatively straightforward mechanisms, can be well studied with this simple setting. The more complex case of co-design of control policy and machine structure is left for future work.

---

> ### Comment · Reviewer_i7Vh · 2025-11-26
> **Post-rebuttal**
>
> Thank you for the detailed response and for the considerable effort invested in expanding the analysis. Several of my original concerns, however, remain insufficiently addressed or resurface in a different way after the rebuttal.
>
> 1. Scope and focus of the new material: The rebuttal introduces a very large amount of new content (additional tasks such as lifting, flying, jumping, gearboxes; extensive diversity metrics; qualitative examples; multiple tables). While valuable, this volume of new content creates a mismatch wth the framing and scope of the original submission. It becomes difficult to assess what the actual contribution is: the paper presents a limited benchmark but the rebuttal suddenly expands the problem space and evaluation pipeline by a very large margin. As my fellow reviewer 4v71 mentioned, this makes the original submission feel underdeveloped relative to the scope of the rebuttal, which now feels like reading a second, more complete set of experiments without a proper, deep analysis.
>
> 2. Robots vs. machines distinction: I'm not convinced about the justification provided for why robot morphology optimization is fundamentally different from compositional machine design. Several statements in the rebuttal are technically inaccurate: 1) robots are mechanical machines, and the claim that robots have "less complex topology" is not correct, at the kinematic level and at the lower level (each joint/motor is a set of mechanisms on its own). Just consider the variability of robot hands, even for single active DoF!; 2) robot morphology work often uses heterogeneous components (different joints, links, actuators, sensors), sometimes with explicit functional semantics; 3) morphology and control are not always co-designed, many works fix the controller or treat morphology search independently. In fact, you can run different types of control on most robots. The differentiation therefore still feels hand-wavy, and the related-work gap is not clearly resoved: there is a large corpus of robotics work on morphology design.
>
> 3. Diversity metrics: The new diversity analysis is extensive, but it is not clearly connected back to the problem definition or to any specific research question posed by the paper. The metrics used are technically interesting, but without a clear argument for how they reflect meaningful diversity for machine design, the evaluation remains difficult to interpret. Why are these metrics the best to measure "originality" and "completeness" in designs? How do they balance between just creating random different things vs. meaningful different designs? This reinforces my original concern that the benchmark's goals and evaluation criteria are not yet clearly articulated.
>
> 4. Task representativeness: The explanation that cars and catapults cover essential aspects of "static stability, structural understanding, spatial coordination, and temporal reasoning" is reasonable in the abstract, but remains speculative. These capabilities are neither explicitly isolated nor measured. Catapults and cars may indeed be different, but they still represent a narrow slice of the overall design space and do not strongly support general claims about "compositional machine design". The additional tasks shown in the rebuttal highlight that a much broader set of behaviors is important for this domain, which in turn strengthens the concern about limited task coverage.
>
> The rebuttal contains substantial and interesting extensions, but many of the core conceptual questions remain open. The newly added material reinforces the impression that this domain is promising but the manuscript was incomplete and the bulk of the evaluation needs to be contextualized.
> Since I appreciate the novelty of the method and the additional experiments post-submission, I will slightly raise my score but I believe a proper rewriting and contextualizing of the (many) new experiments and metrics, with in-depth analysis and discussion would make this paper a better contribution.

---

> ### Author Response · Authors · 2025-12-03
> **Follow-up response (Part 1)**
>
> We deeply thank the reviewer for their acknowledgement on our contributions, and more importantly, their invaluable contribution to create a healthy and constructive academic environment -- for which we feel compelled to explicitly express our gratitude due to this unexpected situation.
>
> ---
>
> Responding to your following-up comments:
>
> > Scope and focus of the new material
>
> > Task representativeness
>
> We apologize for presenting relatively unorganized results due to limited time. We have done a more comprehensive study for agentic design in the new tasks, as in the main experiments for car and catapult tasks. The detailed description is in the updated draft (Appendix B.4, Line 1225-1237 and Line 1308; also see Table 8 & 9 in the Appendix).
>
> | Models | Flying (Height) | Flying (Height) | Flying (Height) | Jumping (Nums/10) | Jumping (Nums/10) | Jumping (Nums/10) | Delivery (Distance) | Delivery (Distance) | Delivery (Distance) | Lifting (Nums/10) | Lifting (Nums/10) | Lifting (Nums/10) | Gearing (Pass/50) |
> |--------|------------------|------------------|------------------|-------------------|-------------------|-------------------|----------------------|----------------------|----------------------|--------------------|--------------------|--------------------|--------------------|
> |        | Mean | Max | Std | Mean | Max | Std | Mean | Max | Std | Mean | Max | Std | PassRate |
> | Gemini 2.5 Pro | 30.51 | 38.50 | 10.68 | 5.50 | **10** | 3.54 | **32.57** | 45.80 | 12.07 | 9.8 | **10** | 0.4 | **45** |
> | OpenAI o3 | **31.09** | **39.20** | 8.95 | **5.88** | 10 | 4.13 | 29.33 | 42.70 | 14.24 | 5.4 | 10 | 4.45 | 38 |
> | Qwen3-Coder-480B-A35B | 3.73 | 6.96 | 3.22 | 1.5 | 2 | 0.5 | 0 | 0 | 0 | 5.5 | 10 | 4.5 | 0 |
> | Doubao Seed 1.6 | 15.61 | 31.19 | 12.56 | 1.5 | 3 | 0.76 | 11.42 | 27.30 | 8.83 | 5.67 | 10 | 4.35 | 36 |
> | Claude Opus 4 | 28.76 | 35.74 | 6.98 | 2.86 | 9 | 3.91 | 3.28 | 11.8 | 4.62 | 1.5 | 2 | 0.5 | 4 |
> | DeepSeek-V3 | 0.31 | 0.51 | 0.2 | 0 | 0 | 0 | 0 | 0 | 0 | 0.75 | 1 | 0.43 | 17 |
> | Kimi K2 | 1.02 | 1.52 | 0.51 | 3 | 3 | 0 | 0 | 0 | 0 | 0.33 | 1 | 0.47 | 5 |
> | Llama 4 Scout 17B 16E | 0.26 | 0.51 | 0.25 | 1 | 1 | 0 | 0 | 0 | 0 | 0 | 0 | 0 | 15 |
> | Human | **256.19** | **568.09** | 181.42 | **10** | **10** | 0 | **88.80** | 214.84 | 49.30 | **10** | **10** | 0 | 5 (Random Guess) |
>
>
> Regarding the focus of the new material: the introduced new materials, while looking bulky, fall in the framework we have laid down: to study how LLMs behave in exemplary design tasks. The new tasks have emphasis on some dimensions (e.g., gearing on pure static and compositional design, closer to combinatorial optimization) but never introduce new dimensions — car and catapults together cover 1) spatial reasoning on positions, poses and spatial constraints, 2) physical reasoning on future frames, 3) instruction following, 4) CoT reasoning, 5) input/output code formatting understanding, 6) ability in decomposing tasks into sub-tasks / high-level task planning, and many more.
>
> The major difference in our opinion is that they require understanding of different physical laws, like flying involves some aspects of aerodynamics. But we would like to point out that the key challenges are the ones on general LLMs capabilities shown above, instead of individual challenges for individual tasks.

---

> > ### Author Response · Authors · 2025-12-03
> > **Follow-up response (Part 2)**
> >
> > > Robots vs. machines distinction
> >
> > We thank the reviewer for their comments on robotic morphology. We totally agree that robotic design and machine design share methodological similarities, and we appreciate the reviewer’s clarification regarding the scope of robot design. However, we emphasize that our work targets the broader problem of machine design, which is arguably one of the central challenges in mechanical engineering and of which robot morphology constitutes a subdomain.
> >
> > In particular, our focus is on complex spatial reasoning tasks that are not typically addressed in mainstream robotics morphology research. For example, the gearing task we present involves spatial composition and functional arrangement challenges that differ significantly from those encountered in common robotic morphology benchmarks. Although gear configuration is not a core topic within robot morphology, it is a fundamental problem in machine design with wide practical relevance, from educational mechanical kits to industrial machinery.
> >
> > Our goal is to explore the generative and interpretative capabilities of LLMs within this broader machine design context. We will continue to carefully review related work to ensure that no relevant research from robotics or adjacent fields is overlooked.
> >
> >
> > > Diversity metrics
> >
> > We thank the reviewer i7Vh for pointing that defining diversity is challenging. In machine design, this could refer to multiple aspects:
> >
> > - shape and appearance (particularly for consumer products),
> > - topology/articulation patterns,
> > - specific parts used,
> > - underlying mechanisms and semantics (e.g., lever vs. pulley systems, or the key principle behind a suspension)
> > - physical properties affecting stability.
> >
> > Given that our tasks are relatively simple and LLMs currently produce only simple mechanisms, and considering that building proxies for semantic diversity is non-trivial, we focused on measuring shape and topology diversity due to time constraints. Fully addressing this question could itself be a separate paper.
> >
> > Also, we would also like to point out that for our simple parts and articulation graphs, these measured dimensions correlate highly with semantic and physical property diversity.

---

### Author Response · Authors · 2025-12-03
**Comprehensive benchmark results on new tasks**

We have benchmarked the performance of agentic pipelines with different LLMs on the new tasks. The detailed description is in the updated draft (Appendix B.4, Line 1225-1237 and Line 1308; also see Table 8 & 9 in the Appendix)

| Models | Flying (Height) | Flying (Height) | Flying (Height) | Jumping (Nums/10) | Jumping (Nums/10) | Jumping (Nums/10) | Delivery (Distance) | Delivery (Distance) | Delivery (Distance) | Lifting (Nums/10) | Lifting (Nums/10) | Lifting (Nums/10) | Gearing (Pass/50) |
|--------|------------------|------------------|------------------|-------------------|-------------------|-------------------|----------------------|----------------------|----------------------|--------------------|--------------------|--------------------|--------------------|
|        | Mean | Max | Std | Mean | Max | Std | Mean | Max | Std | Mean | Max | Std | PassRate |
| Gemini 2.5 Pro | 30.51 | 38.50 | 10.68 | 5.50 | **10** | 3.54 | **32.57** | 45.80 | 12.07 | 9.8 | **10** | 0.4 | **45** |
| OpenAI o3 | **31.09** | **39.20** | 8.95 | **5.88** | 10 | 4.13 | 29.33 | 42.70 | 14.24 | 5.4 | 10 | 4.45 | 38 |
| Qwen3-Coder-480B-A35B | 3.73 | 6.96 | 3.22 | 1.5 | 2 | 0.5 | 0 | 0 | 0 | 5.5 | 10 | 4.5 | 0 |
| Doubao Seed 1.6 | 15.61 | 31.19 | 12.56 | 1.5 | 3 | 0.76 | 11.42 | 27.30 | 8.83 | 5.67 | 10 | 4.35 | 36 |
| Claude Opus 4 | 28.76 | 35.74 | 6.98 | 2.86 | 9 | 3.91 | 3.28 | 11.8 | 4.62 | 1.5 | 2 | 0.5 | 4 |
| DeepSeek-V3 | 0.31 | 0.51 | 0.2 | 0 | 0 | 0 | 0 | 0 | 0 | 0.75 | 1 | 0.43 | 17 |
| Kimi K2 | 1.02 | 1.52 | 0.51 | 3 | 3 | 0 | 0 | 0 | 0 | 0.33 | 1 | 0.47 | 5 |
| Llama 4 Scout 17B 16E | 0.26 | 0.51 | 0.25 | 1 | 1 | 0 | 0 | 0 | 0 | 0 | 0 | 0 | 15 |
| Human | **256.19** | **568.09** | 181.42 | **10** | **10** | 0 | **88.80** | 214.84 | 49.30 | **10** | **10** | 0 | 5 (Random Guess) |

---

### Meta-Review · Area_Chair_Cnyr · 2025-12-31

**Summary:**

This paper introduces BesiegeField, a new simulation testbed to test how well LLMs are able to design complex machines. The authors design and propose six tasks (although only two were present in the initial submission) that test LLMs along various different axes. The authors benchmark various different closed source models and an RL-trained open source model. Results show that while there is a gradient of progress (i.e. models that are generally accepted as "better" are better at this task), all models still lag behind human performance.

The major concerned shared by all reviewers is the breadth of the proposed tasks. In the initial submission there were only two tasks and this doesn't provide the breadth needed to support the paper's analysis. In the rebuttal the authors introduced additional tasks. While the all reviewers and the AC appreciate these tasks and the AC believes the task set is likely now broad enough, there needs to be additional analysis and these tasks need to fully integrated into the paper.

The AC commends the effort by the authors to add these tasks and encourages them to continue working on this paper, but, unfortunately, this paper cannot be recommended for publication at ICLR at this time.

**Reviewer Concerns:**

# Reviewer i7Vh

- Breadth of task coverage. I believe this concern is still outstanding.
- Alignment of the metrics and the paper's goals. I believe this concern has been largely addressed.
- Related work. I believe this concern has been largely addressed.

# Reviewer T6fA

- Breadth of task coverage. I believe this concern is still outstanding.
- Diversity of solutions. I believe this concern is still outstanding, however I also don't think this paper has to deal with this issue to be worthy of publication.
- Performance of models. I believe this concern is still outstanding, however I also don't think this paper has to deal with this issue to be worthy of publication.
- Simplifications in the simulator. I believe this concern is still outstanding, however I also don't think this paper has to deal with this issue to be worthy of publication.

# Reviewer Xu8F

- Breadth of task coverage. I believe this concern is still outstanding.
- Clarity and novelty of claims. I believe this was partially addressed.

# Reviewer 4v71

- Breadth of task coverage. I believe this concern is still outstanding.
- Lack of human baselines. I believe this was addressed.

**Reviewer Scores:**

I don't believe any reviewers would have meaningfully modified their scores. The major concerned shared by all reviewers, the breadth of task coverage, could likely be addressed by the addition of knew tasks, but, the timeframe of a rebuttal is unfortunately not enough to integrate these new results and give them the analysis that they need to fully address this concern.

---

### Decision · Program_Chairs · 2026-01-26

Reject